# Gradient-Based Feature Learning under Structured Data

**Alireza Mousavi-Hosseini**[1], **Denny Wu**[2], **Taiji Suzuki**[3], **Murat A. Erdogdu**[1]

[1]University of Toronto and Vector Institute,
[2]New York University and Flatiron Institute,
[3]University of Tokyo and RIKEN AIP

{mousavi,erdogdu}@cs.toronto.edu, dennywu@nyu.edu,
taiji@mist.i.u-tokyo.ac.jp

## Abstract

Recent works have demonstrated that the sample complexity of gradient-based learning of single index models, i.e. functions that depend on a 1-dimensional projection of the input data, is governed by their *information exponent*. However, these results are only concerned with isotropic data, while in practice the input often contains additional structure which can implicitly guide the algorithm. In this work, we investigate the effect of a *spiked covariance* structure and reveal several interesting phenomena. First, we show that in the anisotropic setting, the commonly used spherical gradient dynamics may fail to recover the true direction, even when the spike is perfectly aligned with the target direction. Next, we show that appropriate weight normalization that is reminiscent of *batch normalization* can alleviate this issue. Further, by exploiting the alignment between the (spiked) input covariance and the target, we obtain improved sample complexity compared to the isotropic case. In particular, under the spiked model with a suitably large spike, the sample complexity of gradient-based training can be made independent of the information exponent while also outperforming lower bounds for rotationally invariant kernel methods.

## 1 Introduction

A fundamental feature of neural networks is their *adaptivity* to learn unknown statistical models. For instance, when the learning problem exhibits certain low-dimensional structure or sparsity, it is expected that neural networks optimized by gradient-based algorithms can efficiently adapt to such structure via feature/representation learning. A considerable amount of research has been dedicated to understanding this phenomenon under various assumptions and to demonstrate the superiority of neural networks over non-adaptive methods such as kernel models [GMMM19, WLLM19, BES⁺19, LMZ20, AAM22, BES⁺22, DLS22, Tel23, MHPG⁺23].

A particular relevant problem setting for feature learning is the estimation of single index models, where the response $y \in \mathbb{R}$ depends on the input $\boldsymbol{x} \in \mathbb{R}^d$ via $y = g(\langle \boldsymbol{u}, \boldsymbol{x} \rangle) + \epsilon$, where $g : \mathbb{R} \to \mathbb{R}$ is the nonlinear link function and $\boldsymbol{u}$ is the unit target direction. Here, learning corresponds to recovering the unknowns $\boldsymbol{u}$ and $g$, which requires the model to extract and adapt to the low-dimensional target direction. Recent works have shown that the sample complexity is determined by certain properties of the link function $g$. In particular, the complexity of gradient-based optimization is captured by the *information exponent* of $g$ introduced by [BAGJ21]. Intuitively, a larger information exponent $s$ corresponds to a more complex $g$ (for gradient-based learning), and it has been proven that when the input is isotropic $\boldsymbol{x} \sim \mathcal{N}(0, \mathbf{I}_d)$, gradient flow can learn the single index model with $\tilde{\mathcal{O}}(d^s)$ sample complexity [BBSS22].

37th Conference on Neural Information Processing Systems (NeurIPS 2023).

In practice, however, real data always exhibits certain structures such as low intrinsic dimensionality, and isotropic data assumptions fail to capture this fact. In statistics methodology, it is known that the directions along which the input $x$ has high variance are often good predictors of the target $y$ [HTFF09]; indeed, this is the main reason principal component analysis is used in pre-training [JWHT13]. A fundamental model that captures such a structure is the *spiked matrix model* in which $x \sim \mathcal{N}(0, \mathbf{I}_d + \kappa \boldsymbol{\theta}\boldsymbol{\theta}^\top)$ for some unit direction $\boldsymbol{\theta} \in \mathbb{R}^d$ and $\kappa > 0$ [Joh01]. Along the direction $\boldsymbol{\theta}$, data has higher variability and predictive power. In single index models, such predictive power translates to a non-trivial alignment between the vectors $u$ and $\boldsymbol{\theta}$ — our focus is to investigate the effect of such alignment on the sample complexity of gradient-based training.

## 1.1 Contributions: learning single index models under spiked covariance

In this paper, we study the sample complexity of learning a single index model using a two-layer neural network and show that it is determined by an interplay between

- **spike-target alignment**: $\langle u, \boldsymbol{\theta} \rangle \asymp d^{-r_1}$, $r_1 \in [0, 1/2]$,
- **spike magnitude**: $\kappa \asymp d^{r_2}$, for $r_2 \in [0, 1]$.

Our contributions can be summarized as follows.

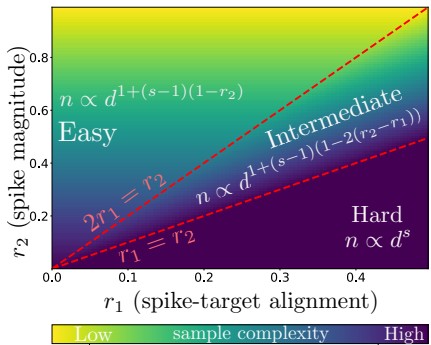

Figure 1: Sample complexity to learn $u$ and $g$ under the spiked model. Smaller $r_1$ denotes a better spike-target alignment, while larger $r_2$ denotes a larger spike magnitude. The sample complexities are based on Corollary 8.

1. We show that even in the case of perfect spike-target alignment ($r_1 = 0$), the spherical gradient flow commonly employed in recent literature (see e.g. [BAGJ21, BBSS22]) cannot recover the target direction for moderate spike magnitudes in the population limit. The failure of this covariance-agnostic procedure under anisotropic structure insinuates the necessity of an appropriate covariance-aware normalization to effectively learn the single index model.

2. We show that a covariance-aware normalization that resembles *batch normalization* resolves this issue. Indeed, the resulting gradient flow can successfully recover the target direction $u$ in this case, and depending on the amount of spike-target alignment, the sample complexity can significantly improve compared to the isotropic case.

3. Under the spiked covariance model, we prove a three-stage phase transition for the sample complexity depending on the quantities $r_1$ and $r_2$. For a suitable direction and magnitude of the spike, the sample complexity can be made $\tilde{\mathcal{O}}(d^{3+\nu})$ for any $\nu > 0$ which is independent of the information exponent $s$. This should be compared against the known complexity of $\tilde{\mathcal{O}}(d^s)$ under isotropic data.

4. We finally show that preconditioning the training dynamics with the inverse covariance improves the sample complexity. This is particularly significant for the spiked covariance model where $\tilde{\mathcal{O}}(d^{3+\nu})$ samples can be reduced to $\tilde{\mathcal{O}}(d^{1+\nu})$ for any $\nu > 0$, i.e. almost linear in $d$. The three-stage phase transition also emerges, as illustrated in Figure 1: in the "hard" regime, the complexity remains $\tilde{\mathcal{O}}(d^s)$ regardless of the magnitude and direction of the spike, while in the "easy" regime the complexity only depends on the spike magnitude and not its direction. The "intermediate" regime interpolates between these two; smaller $r_1$ and larger $r_2$ improve the sample complexity.

The rest of the paper is organized as follows. We discuss the notation and the related work in the remainder of this section. We provide preliminaries on the statistical model and the training procedure in Section 2, and provide a negative result on the covariance-agnostic gradient flow in Section 2.1. Our main sample complexity result on a single neuron is presented in Section 3.2. We provide our results on multi-neuron neural networks in Section 4 and also discuss extensions such as preconditioning and its implications. We provide a technical summary in Section 5 and conclude in Section 6.

**Notation.** We use $\langle \cdot, \cdot \rangle$ and $\|\cdot\|$ to denote Euclidean inner product and norm. For matrices, $\|\cdot\|$ denotes the usual operator norm, and $\lambda_{\max}(\cdot)$ and $\lambda_{\min}(\cdot)$ denote the largest and smallest eigenvalues respectively. We reserve $\gamma$ for the standard Gaussian distribution on $\mathbb{R}$, and let $\|\cdot\|_\gamma$ denote the $L^2(\gamma)$ norm. $\mathbb{S}^{d-1}$ is the unit $d$-dimensional sphere. For quantities $a$ and $b$, we will use $a \lesssim b$ to convey there exists a constant $C$ (a universal constant unless stated otherwise, in which case may depend on polylogarithmic factors of $d$) such that $a \leq Cb$, and $a \asymp b$ signifies that $a \lesssim b$ and $b \lesssim a$.

## 1.2 Further related work

**Non-Linear Feature Learning with Neural Networks.** Recently, two popular scaling regimes of neural networks have emerged for theoretical studies. A large initialization variance leads to the *lazy training* regime, where the weights do not move significantly, and the training dynamics is captured by the neural tangent kernel (NTK) [JGH18, COB19]. However, there are many instances of function classes that are efficiently learnable by neural networks and not efficiently learnable by the NTK [YS19, GMMM19]. Under a smaller initialization scaling, gradient descent on infinite-width neural networks becomes equivalent to Wasserstein gradient flow on the space of measures, known as the mean-field limit [CB18, RVE18, MMN18, MMM19, NWS22, Chi22], which can learn certain low-dimensional target functions efficiently [WLLM19, AAM22, HC22, ASKL23].

As for neural networks with smaller width, recent works showed that a two-stage feature learning procedure can outperform the NTK when the data is sampled uniformly from the hypercube [BEG+22] or isotropic Gaussian [DLS22, BES+22, BBSS22, MHPG+23, ABAM23]. However, these results do not take into account the additional structure that might be present in the covariance matrix of the input data. Two notable exceptions are [GMMM20, RGKZ21], where the authors analyzed a spiked covariance and Gaussian mixture data, respectively. Our setting is closer to [GMMM20], however, they do not provide optimization guarantees through gradient-based training. Furthermore, in a companion work [BES+23], we zoom into the setting where the spike and target are perfectly aligned ($r_1 = 0$), and prove learnability in the $n \asymp d$ regime for both kernel regression and two-layer neural network. Finally, we go over some results concurrent to our work in Appendix A.

**Learning Single Index Models.** The problem of estimating the relevant direction in a single index model is classical in statistics [LD89], with efficient dedicated algorithms ([KKSK11, CM20] among others). However, these algorithms are non-standard and instead, we are concerned with standard iterative algorithms like training neural networks with gradient descent. Recently, [DH18] considered an iterative optimization procedure for learning such models with a polynomial sample complexity that is controlled by the smoothness of the link function. [BES+22] considered the effect of taking a single gradient step on the ability of a two-layer neural network to learn a single index model, and [BBSS22, MHPG+23] considered training a special two-layer neural network architecture where all neurons share the same weight with gradient flow or online SGD. However, these works only consider the isotropic Gaussian input, and the effect of anisotropy in the covariance matrix when training a neural network to learn a single index model has remained unclear.

**Training a Single Neuron with Gradient Descent.** When training the first layer, we consider a setting where there is only one effective neuron. A large body of works exists on training a single neuron using variants of gradient descent. In the realizable setting (i.e. identical link and activation), the typical assumptions on the activation correspond to information exponent 1 as the activations are required to be monotone or have similar properties, see e.g. [Sol17, YO20, DKTZ22]. In the agnostic setting, [FCG20] considered initializing from the origin which is a saddle point for information exponent larger than 1. [ATV22] also considered the agnostic learning of a ReLU activation, albeit their sample complexity is not explicit other than being polynomial in dimension.

## 2 Preliminaries: Statistical Model and Training Procedure

For a $d$-dimensional input $\boldsymbol{x}$ and a link function $g \in L^2(\gamma)$, consider the single index model

$$y = g\left(\frac{\langle \boldsymbol{u}, \boldsymbol{x} \rangle}{\|\boldsymbol{\Sigma}^{1/2} \boldsymbol{u}\|}\right) + \epsilon \quad \text{with} \quad \boldsymbol{x} \sim \mathcal{N}(0, \boldsymbol{\Sigma}), \tag{2.1}$$

where $\epsilon$ is a zero-mean noise with $\mathcal{O}(1)$ sub-Gaussian norm and $\boldsymbol{u} \in \mathbb{S}^{d-1}$. Learning the model (2.1) corresponds to approximately recovering the unknown link $g$ and the unknown direction $\boldsymbol{u}$. Note that a normalization is needed to make this problem well-defined; without loss of generality, we write $\langle \boldsymbol{u}, \boldsymbol{x} \rangle / \|\boldsymbol{\Sigma}^{1/2} \boldsymbol{u}\|$ to ensure that the input variance and the scaling of $g$ both remain independent of the conditioning of $\boldsymbol{\Sigma}$. For this learning task, we will use a two-layer neural network of the form

$$\hat{y}(\boldsymbol{x}; \boldsymbol{W}, \boldsymbol{a}, \boldsymbol{b}) := \sum_{i=1}^{m} a_i \phi(\langle \boldsymbol{w}_i, \boldsymbol{x} \rangle + b_i), \tag{2.2}$$

where $\boldsymbol{W} = \{\boldsymbol{w}_i\}_{i=1}^{m}$ is the $m \times d$ matrix whose rows corresponds to first-layer weights $\boldsymbol{w}_i$, $\boldsymbol{a} = \{a_i\}_{i=1}^{m}$ denote the second-layer weights, $\boldsymbol{b} = \{b_i\}_{i=1}^{m}$ denote the biases, and $\phi$ is the non-linear

activation function. We assume $g$ and $\phi$ are weakly differentiable with weak derivatives $g'$ and $\phi'$ respectively, and $g, g', \phi, \phi' \in L^2(\gamma)$. We are interested in the high-dimensional regime; thus, $d$ is assumed to be sufficiently large throughout the paper. Our ultimate goal is to learn both unknowns $g$ and $\boldsymbol{u}$ by minimizing the population risk

$$R(\boldsymbol{W}, \boldsymbol{a}, \boldsymbol{b}) := \frac{1}{2} \mathbb{E}\big[(\hat{y}(\boldsymbol{x}; \boldsymbol{W}, \boldsymbol{a}, \boldsymbol{b}) - y)^2\big], \tag{2.3}$$

using a gradient-based training method such as gradient flow.

We follow the two-step training procedure employed in recent works [BES$^+$22, MHPG$^+$23, BBSS22, DLS22]: First, we train the first-layer weights $\boldsymbol{W}$ to learn the unknown direction $\boldsymbol{u}$; at the end of this stage, the neurons $\boldsymbol{w}_i$ align with $\boldsymbol{u}$. Here, the goal is to recover only the direction. Next, using random biases and training the second-layer weights, we obtain a good approximation for the unknown link function $g$. In the majority of this work, we focus on the first part of this two-stage procedure as the alignment between $\boldsymbol{w}_i$'s and $\boldsymbol{u}$ essentially determines the sample complexity of the overall procedure. This problem is somewhat equivalent to the simplified problem of minimizing (2.3) with $m = 1$, $a_1 = 1$, $b_1 = 0$, i.e., $\hat{y}(\boldsymbol{x}; \boldsymbol{W}, \boldsymbol{a}, \boldsymbol{b})$ is replaced with $\hat{y}(\boldsymbol{x}; \boldsymbol{w}) := \phi(\langle \boldsymbol{w}, \boldsymbol{x} \rangle)$ and we write $R(\boldsymbol{w}) := R(\boldsymbol{W}, \boldsymbol{a}, \boldsymbol{b})$ for simplicity. We emphasize that unless $\phi = g$ (i.e. the link function is known), the first stage of training only recovers the relevant direction $\boldsymbol{u}$ and is not able to approximate $g$. Indeed, $m > 1$ is often needed to learn the non-linear link function; this is the focus of Section 4.2 where we derive a complete learnability result for a two-layer neural network with $m > 1$.

Characteristics of the link function play an important role in the complexity of learning the model. As such, a central part of our analysis will rely on a particular property based on the Hermite expansion of functions in a basis defined by the normalized Hermite polynomials $\{h_j\}_{j \geq 0}$ given as

$$h_j(z) = \frac{(-1)^j e^{z^2/2}}{\sqrt{j!}} \frac{\mathrm{d}^j}{\mathrm{d}z^j} e^{-z^2/2}. \tag{2.4}$$

These polynomials form an orthonormal basis in the space $L^2(\gamma)$, and the resulting expansion yields the following measure of complexity for $g$, which is termed as *the information exponent*.

**Definition 1** (Information exponent)**.** *Let* $g = \sum_{j \geq 0} \alpha_j h_j$ *be the Hermite expansion of* $g$. *The information exponent of* $g$ *is defined to be* $s := \inf\{j > 0 : \alpha_j \neq 0\}$.

This concept was introduced in [BAGJ21] in a more general framework, and our definition is more in line with the setting in [BBSS22]. We remark that the definition of [BAGJ21] can be modified to handle anisotropy in which case one arrives at Definition 1. We provide a detailed discussion on this concept together with some properties of the Hermite expansion in Appendix B. Throughout the paper, we assume that the information exponent does not grow with dimension.

In the case where the $d$-dimensional input data is isotropic, [BBSS22] showed that learning a single index target with full-batch gradient flow requires a sample complexity of $\tilde{\mathcal{O}}(d^s)$ for $s \geq 3$ where $s$ is the information exponent of $g$. We will show that this sample complexity can be improved under anisotropy. More specifically, if the input covariance $\boldsymbol{\Sigma}$ has non-trivial alignment with the unknown direction $\boldsymbol{u}$, we prove in Section 3 that the resulting sample complexity can be even made independent of the information exponent if we use a certain normalization in the training. In what follows, we prove that such a normalization in training procedure is indeed necessary.

## 2.1 The spiked model and limitations of covariance-agnostic training

In practice, data often exhibit a certain structure which may have a profound impact on the statistical procedure. A well-known model that captures such a structure is the *spiked model* [Joh01] for which one or several large eigenvalues of the input covariance matrix $\boldsymbol{\Sigma}$ are separated from the bulk of the spectrum (see also [BBAP05, BS06]). Although our results hold for generic covariance matrices, they reveal interesting phenomena under the following spiked model assumption.

**Assumption 1.** *The covariance* $\boldsymbol{\Sigma}$ *follows the* $(\kappa, \boldsymbol{\theta})$-*spiked model if* $\boldsymbol{\Sigma} = \frac{\mathbf{I}_d + \kappa \boldsymbol{\theta}\boldsymbol{\theta}^\top}{1+\kappa}$ *where* $\|\boldsymbol{\theta}\| = 1$.

In pursuit of the target (unit) direction $\boldsymbol{u}$, the magnitude of the neuron $\boldsymbol{w}$ is immaterial; thus, recent works take advantage of this and simplify the optimization trajectory by projecting $\boldsymbol{w}$ onto unit sphere $\mathbb{S}^{d-1}$ throughout the training process [BAGJ21, BBSS22]. In the sequel, we study the same dynamics

which is agnostic to the input covariance in order to motivate our investigation of normalized gradient flow in Section 3. More specifically, we consider the spherical population gradient flow

$$\frac{\mathrm{d}\boldsymbol{w}^t}{\mathrm{d}t} = -\nabla^S R(\boldsymbol{w}^t) \text{ where } \nabla^S R(\boldsymbol{w}) = \nabla R(\boldsymbol{w}) - \langle \nabla R(\boldsymbol{w}), \boldsymbol{w}\rangle \boldsymbol{w}. \tag{2.5}$$

where $\nabla^S$ is the spherical gradient at the current iterate. It is straightforward to see that when the initialization $\boldsymbol{w}^0$ is on the unit sphere, the entire flow will remain on the unit sphere, i.e. $\boldsymbol{w}^t \in \mathbb{S}^{d-1}$ for all $t \geq 0$. The flow (2.5) has been proven useful for learning the direction $\boldsymbol{u}$ [BBSS22] in the isotropic case $\boldsymbol{\Sigma} = \mathbf{I}_d$ when the activation $\phi$ is ReLU. In contrast, when $\boldsymbol{\Sigma}$ follows a spiked model, we show that it can get stuck at stationary points that are almost orthogonal to $\boldsymbol{u}$. Indeed, when the input covariance $\boldsymbol{\Sigma}$ has a spike in the target direction $\boldsymbol{u}$, i.e. $\boldsymbol{\theta} = \boldsymbol{u}$, one expects that the training procedure benefits from this as the input $\boldsymbol{x}$ contains information about the sought unknown $\boldsymbol{u}$ without even querying the response $y$. The following result proves the contrary; for moderate spike magnitudes, the alignment between the first-layer weights and target $\langle \boldsymbol{w}^t, \boldsymbol{u}\rangle$ will be insignificant for all $t$.

**Theorem 2.** *Let $s > 2$ be the information exponent of $g$ with $\mathbb{E}[g] = 0$, and assume $\boldsymbol{\Sigma}$ follows the $(\kappa, \boldsymbol{u})$-spiked model with $\Omega(1) \leq \kappa \leq \mathcal{O}(d^{\frac{s-2}{s-1}})$. For ReLU activation, let $\boldsymbol{w}^t$ denote the solution to (2.5) initialized uniformly at random over $\mathbb{S}^{d-1}$, then with probability at least $0.99$,*

$$\sup_{t \geq 0}|\langle \boldsymbol{w}^t, \boldsymbol{u}\rangle| \lesssim 1/\sqrt{d}, \tag{2.6}$$

A non-trivial alignment between the first-layer weights $\boldsymbol{w}^t$ and the target direction $\boldsymbol{u}$ is required to learn the single index model (2.1). However, the above result implies that in high dimensions when $d \gg 1$, the alignment is negligible in the population limit (when the number of samples goes to infinity). We remark that when the spike magnitude is large, i.e. $\kappa \geq \Omega(d)$, the flow (2.5) can achieve alignment as the problem essentially becomes one-dimensional, as we demonstrate in Appendix C.

To see why the flow (2.5) gets stuck at saddle points and fails to recover the true direction, notice that

$$R(\boldsymbol{w}) = \tfrac{1}{2}\mathbb{E}\Big[(\phi(\langle \boldsymbol{w}, \boldsymbol{x}\rangle) - y)^2\Big] = \tfrac{1}{2}\mathbb{E}\big[\phi(\langle \boldsymbol{w}, \boldsymbol{x}\rangle)^2\big] - \mathbb{E}[\phi(\langle \boldsymbol{w}, \boldsymbol{x}\rangle)y] + \tfrac{1}{2}\mathbb{E}\big[y^2\big]. \tag{2.7}$$

If the input was isotropic, i.e. $\boldsymbol{x} \sim \mathcal{N}(0, \mathbf{I}_d)$, the first term in (2.7) would be equal to $\|\phi\|_\gamma^2$, which is independent of $\boldsymbol{w}$. Thus, minimizing $R(\boldsymbol{w})$ in this case is equivalent to maximizing the "correlation" term $\mathbb{E}[\phi(\langle \boldsymbol{w}, \boldsymbol{x}\rangle)y]$. However, under the spiked model, the alignment between $\boldsymbol{w}$ and $\boldsymbol{u}$ breaks the symmetry; consequently, the first term in the decomposition grows with $\langle \boldsymbol{w}, \boldsymbol{u}\rangle$, creating a repulsive force that traps the dynamics around the equator where $\boldsymbol{w}$ is almost orthogonal to $\boldsymbol{u}$.

## 3 Main Results: Alignment via Normalized Dynamics

Having established that the covariance-agnostic training dynamics (2.5) is likely to fail, we consider a covariance-aware normalized flow in this section and show that it can achieve alignment with the unknown target and enjoy better sample complexity compared to the existing results [BAGJ21, BBSS22] in the isotropic case. We start with the population dynamics.

### 3.1 Warm-up: Population dynamics

To simplify the exposition, we define $\boldsymbol{z} := \boldsymbol{\Sigma}^{-1/2}\boldsymbol{x}$, $\overline{\boldsymbol{w}} := \boldsymbol{\Sigma}^{1/2}\boldsymbol{w}/\|\boldsymbol{\Sigma}^{1/2}\boldsymbol{w}\|$ and similarly define $\overline{\boldsymbol{u}}$, and consider the prediction function $\hat{y}(\boldsymbol{x}; \overline{\boldsymbol{w}}) := \phi(\langle \overline{\boldsymbol{w}}, \boldsymbol{z}\rangle)$. Due to symmetry, the second moment of the prediction is $\mathbb{E}\big[\hat{y}(\boldsymbol{x}; \overline{\boldsymbol{w}})^2\big] = \|\phi\|_\gamma^2$ which is independent of $\boldsymbol{w}$; thus, the population risk reads

$$\mathcal{R}(\boldsymbol{w}) := \tfrac{1}{2}\mathbb{E}\Big[(\hat{y}(\boldsymbol{x}; \overline{\boldsymbol{w}}) - y)^2\Big] = \tfrac{1}{2}\|\phi\|_\gamma^2 + \tfrac{1}{2}\mathbb{E}\big[y^2\big] - \mathbb{E}[\phi(\langle \overline{\boldsymbol{w}}, \boldsymbol{z}\rangle)y]. \tag{3.1}$$

In (3.1), the only term that depends on the weights $\boldsymbol{w}$ is the correlation term and the source of the repulsive force in (2.7) is eliminated; we have $\nabla_{\boldsymbol{w}}\mathcal{R}(\boldsymbol{w}) = -\nabla_{\boldsymbol{w}}\mathbb{E}[\phi(\langle \overline{\boldsymbol{w}}, \boldsymbol{z}\rangle)y]$. Based on this, we use the following normalized gradient flow for training

$$\frac{\mathrm{d}\boldsymbol{w}^t}{\mathrm{d}t} = -\eta(\boldsymbol{w}^t)\nabla_{\boldsymbol{w}}\mathcal{R}(\boldsymbol{w}^t) \text{ where } \eta(\boldsymbol{w}) = \|\boldsymbol{\Sigma}^{1/2}\boldsymbol{w}\|^2. \tag{3.2}$$

We remark that, though not identical, this normalization is closely related to batch normalization which is commonly employed in practice [IS15]. Under the invariance provided by the current normalization, minimizing $\mathcal{R}(\boldsymbol{w})$ corresponds to maximizing $\mathbb{E}[\phi(\langle \overline{\boldsymbol{w}}, \boldsymbol{z} \rangle)y]$. Thus, instead of $\boldsymbol{w}$, it will be more useful to track the dynamics of its normalized counterpart $\overline{\boldsymbol{w}}$, which is made possible by the following intermediary result that follows from Stein's lemma; also see e.g. [EDB16, MHPG$^+$23].

**Lemma 3.** *Suppose we train $\boldsymbol{w}^t$ using the gradient flow (3.2). Then $\overline{\boldsymbol{w}}^t$ solves the following ODE*

$$\frac{\mathrm{d}\overline{\boldsymbol{w}}^t}{\mathrm{d}t} = -\zeta_{\phi,g}(\langle \overline{\boldsymbol{w}}^t, \overline{\boldsymbol{u}} \rangle)(\mathbf{I}_d - \overline{\boldsymbol{w}}^t \overline{\boldsymbol{w}}^{t\top})\boldsymbol{\Sigma}(\mathbf{I}_d - \overline{\boldsymbol{w}}^t \overline{\boldsymbol{w}}^{t\top})\overline{\boldsymbol{u}}, \tag{3.3}$$

*where* $\zeta_{\phi,g}(\langle \overline{\boldsymbol{w}}, \overline{\boldsymbol{u}} \rangle) \coloneqq -\mathbb{E}[\phi'(\langle \overline{\boldsymbol{w}}, \boldsymbol{z} \rangle)g'(\langle \overline{\boldsymbol{u}}, \boldsymbol{z} \rangle)]$.

We will investigate if the modified flow (3.3) achieves alignment; in this context, alignment corresponds to $\langle \overline{\boldsymbol{w}}^t, \overline{\boldsymbol{u}} \rangle \approx 1$. Towards that end, we make the following assumption.

**Assumption 2.** *Let $g = \sum_{j \geq 0} \alpha_j h_j$ and $\phi = \sum_{j \geq 0} \beta_j h_j$ be the Hermite decomposition of $g$ and $\phi$ respectively. Let $s$ be the information exponent of $g$. For some universal constant $c > 0$, we assume*

$$\zeta_{\phi,g}(\omega) = -\sum_{j > 0} j \alpha_j \beta_j \, \omega^{j-1} \leq -c\,\omega^{s-1}, \qquad \forall \omega \in (0, 1).$$

There are several important examples that readily satisfy Assumption 2. The obvious example is when the link function is known as in [BAGJ21], i.e. $\phi = g$. A more interesting example is when $\phi$ is an activation with degree $s$ non-zero Hermite coefficient (e.g. ReLU when $s$ is even, see [GKK19, Claim 1]) and $g$ is a degree $s$ Hermite polynomial, which for $s = 2$ corresponds to the phase retrieval problem. In this case, the assumption is satisfied if $\alpha_s$ and $\beta_s$ have the same sign, which occurs with probability $0.5$ if we randomly choose the sign of the second layer.

Under this condition, the following result shows that the population flow (3.3) can achieve alignment.

**Proposition 4.** *Suppose Assumption 2 holds and consider the gradient flow given by (3.3) with initialization satisfying $\langle \overline{\boldsymbol{w}}^0, \overline{\boldsymbol{u}} \rangle > 0$. Then, we have $\langle \overline{\boldsymbol{w}}^T, \overline{\boldsymbol{u}} \rangle \geq 1 - \varepsilon$ as soon as*

$$T \asymp \frac{\tau_s(\langle \overline{\boldsymbol{w}}^0, \overline{\boldsymbol{u}} \rangle) + \ln(1/\varepsilon)}{\lambda_{\min}(\boldsymbol{\Sigma})} \quad \textit{where} \quad \tau_s(z) \coloneqq \begin{cases} 1 & s = 1 \\ \ln(1/z) & s = 2 \\ (1/z)^{s-2} & s > 2 \end{cases}. \tag{3.4}$$

We remark that the information exponent enters the rate in (3.4) through the function $\tau_s$, and time needed to achieve $\varepsilon$ alignment gets worse with larger information exponent. Indeed, it is understood that this quantity serves as a measure of complexity for the target function being learned.

### 3.2 Empirical dynamics and sample complexity

Given $n$ i.i.d. samples $\{(\boldsymbol{x}^{(i)}, y^{(i)})\}_{i=1}^n$ from the single index model (2.1), we consider the flow

$$\frac{\mathrm{d}\boldsymbol{w}^t}{\mathrm{d}t} = -\eta(\boldsymbol{w}^t)\nabla \hat{\mathcal{R}}(\boldsymbol{w}^t) \quad \text{with} \quad \nabla \hat{\mathcal{R}}(\boldsymbol{w}) \coloneqq -\nabla_{\boldsymbol{w}}\left\{ \frac{1}{n}\sum_{i=1}^n \phi\left( \frac{\langle \boldsymbol{w}, \boldsymbol{x}^{(i)} \rangle}{\|\hat{\boldsymbol{\Sigma}}^{1/2}\boldsymbol{w}\|} \right)y^{(i)} \right\}, \tag{3.5}$$

where we estimate the covariance matrix $\boldsymbol{\Sigma}$ using the sample mean $\hat{\boldsymbol{\Sigma}} \coloneqq \frac{1}{n'}\sum_{i=1}^{n'} \boldsymbol{x}^{(i)}\boldsymbol{x}^{(i)\top}$ over $n'$ i.i.d. samples; the above dynamics defines an empirical gradient flow. Notice that we ignored the gradient associated with the term $\phi^2$ since the population dynamics ensures that its gradient will concentrate around zero; thus, it is redundant to estimate this term. Below, we will use $n' = n$ for smooth activations, i.e. the same dataset can be used for covariance estimation; For ReLU, we require a more accurate covariance estimator, thus, we use $n' \gtrsim n^2$ by assuming access to an additional $n' - n$ unlabeled data points. Similar to the previous section, we track the dynamics of normalized $\boldsymbol{w}$ by defining $\overline{\boldsymbol{w}} \coloneqq \hat{\boldsymbol{\Sigma}}^{1/2}\boldsymbol{w}/\|\hat{\boldsymbol{\Sigma}}^{1/2}\boldsymbol{w}\|$ (and leave $\overline{\boldsymbol{u}}$ unchanged from Section 3.1). The same arguments as in Lemma 3 allow us to track the evolution of $\overline{\boldsymbol{w}}$, which ultimately yields the following alignment result under general covariance structure.

**Theorem 5.** *Let $s$ be the information exponent of $g$, and assume it satisfies $|g(\cdot)| \lesssim 1 + |\cdot|^p$ for some $p > 0$. For $\phi$ denoting either the ReLU activation or a smooth activation satisfying $|\phi'| \vee |\phi''| \lesssim 1$, suppose Assumption 2 holds. For any $\varepsilon > 0$, suppose we run the finite sample gradient flow (3.5) with $\eta(\boldsymbol{w}) = \|\hat{\boldsymbol{\Sigma}}^{1/2}\boldsymbol{w}\|^2$, initialized such that $\langle \overline{\boldsymbol{w}}^0, \overline{\boldsymbol{u}} \rangle > 0$, and with number of samples*

$$n \gtrsim d\varkappa(\boldsymbol{\Sigma})^2\left\{ \langle \overline{\boldsymbol{w}}^0, \overline{\boldsymbol{u}} \rangle^{2(1-s)} \vee \varepsilon^{-2} \right\},$$

where $\varkappa(\mathbf{\Sigma})$ is the condition number of $\mathbf{\Sigma}$. Then, for $T \asymp \frac{\tau_s\left(\left\langle \overline{\boldsymbol{w}}^0, \overline{\boldsymbol{u}}\right\rangle\right)+\ln(1/\varepsilon)}{\lambda_{\min}(\mathbf{\Sigma})}$, we have

$$\left\langle \overline{\boldsymbol{w}}^T, \overline{\boldsymbol{u}}\right\rangle \geq 1 - \varepsilon, \tag{3.6}$$

with probability at least $1 - c_1 d^{-c_2}$ for some universal constants $c_1, c_2 > 0$ over the randomness of the dataset. Here, $\tau_s$ is defined in (3.4) and $\gtrsim$ hides poly-logarithmic factors.

**Remark.** We make the following remarks on the above theorem.

- The initial condition $\left\langle \overline{\boldsymbol{w}}^0, \overline{\boldsymbol{u}}\right\rangle > 0$ is required when we have odd information exponent. When $\boldsymbol{w}^0$ is initialized uniformly over $\mathbb{S}^{d-1}$, the condition holds with probability $0.5$ over the initialization. See [BAGJ21, Remark 1.8] for further discussion on this condition.

- Although $\overline{\boldsymbol{w}}$ is defined using the empirical covariance unlike $\overline{\boldsymbol{u}}$ which is defined by population covariance, this definition is the suitable choice to approximate the target function $g$ (c.f. Theorem 9), since it ensures the arguments of $\phi$ and $g$ are sufficiently close when $\overline{\boldsymbol{w}}$ recovers $\overline{\boldsymbol{u}}$.

The intuition behind the proof of Theorem 5 is presented in Section 5 with the complete proof in the appendix. We highlight that the improvement in the sample complexity compared to the isotropic setting occurs whenever the covariance structure induces a stronger initial alignment and consequently stronger signal. The following corollary demonstrates a concrete example of such improvement by specializing Theorem 5 for a spiked covariance model.

**Corollary 6.** *Consider the setting of Theorem 5 with $\mathbf{\Sigma}$ following the $(\kappa, \boldsymbol{\theta})$-spiked model, where $\langle \boldsymbol{u}, \boldsymbol{\theta}\rangle \asymp d^{-r_1}$ and $\kappa \asymp d^{r_2}$ with $r_1 \in [0, 1/2]$ and $r_2 \in [0, 1]$. Suppose $\boldsymbol{w}^0$ is sampled uniformly from $\mathbb{S}^{d-1}$. Then, when conditioned on $\left\langle \overline{\boldsymbol{w}}^0, \overline{\boldsymbol{u}}\right\rangle > 0$, the sample complexity in Theorem 5 reads*

$$n \gtrsim \begin{cases} d^{1+2r_2}\left(d^{s-1} \vee \varepsilon^{-2}\right) & 0 < r_2 < r_1 \\ d^{1+2r_2}\left(d^{(s-1)(1-2(r_2-r_1))} \vee \varepsilon^{-2}\right) & r_1 < r_2 < 2r_1, \\ d^{1+2r_2}\left(d^{(s-1)(1-r_2)} \vee \varepsilon^{-2}\right) & 2r_1 < r_2 < 1 \end{cases} \tag{3.7}$$

*where $\gtrsim$ hides poly-logarithmic factors of $d$.*

**Remark.** We have the following observations on the above sample complexity.

- Corollary 6 demonstrates that structured data can lead to better sample complexity when the right normalization is used during training. This complements Theorem 2 where we recall that spherical training dynamics ignores the structure in data and the target direction cannot be recovered.

- When $g$ is a polynomial of degree $p$, the lower bound for rotationally invariant kernels (including the neural tangent kernel at initialization) implies a complexity of at least $d^{\Omega((1-r_2)p)}$ [DWY21]. Thus the sample complexity of Corollary 6 can always outperform the kernel lower bound when $p$ is sufficiently large and $s$ remains constant.

**Three-step phase transition.** Recall that in the isotropic setting $\mathbf{\Sigma} = \mathbf{I}_d$, the sample complexity of learning $g$ with information exponent $s$ using full-batch gradient flow is $\tilde{\mathcal{O}}(d^s)$ for $s \geq 3$ [BBSS22]. The sample complexity in Corollary 6 is strictly smaller than $\tilde{\mathcal{O}}(d^s)$ as soon as $(s-1)r_1/(s-2) < r_2$. Furthermore, for any $\nu > 0$ it is at most $\tilde{\mathcal{O}}(d^{3+\nu})$ as soon as $r_2 \geq 1 - \nu/(s-3)$ and $2r_1 < r_2$, in which case the sample complexity becomes independent of the information exponent. Interestingly, the complexity becomes independent of $r_1$ when $r_2 > 2r_1$ or $r_2 < r_1$, i.e. the direction of the spike becomes irrelevant when the spike magnitude is sufficiently large or small.

The three-stage phase transition of Corollary 6 is due to the different behaviour of the inner product $\left\langle \overline{\boldsymbol{w}}^0, \overline{\boldsymbol{u}}\right\rangle$ in different regimes of $r_1$ and $r_2$. When $r_2 < r_1$, we have $\left\langle \overline{\boldsymbol{w}}^0, \overline{\boldsymbol{u}}\right\rangle \asymp \left\langle \boldsymbol{w}^0, \boldsymbol{u}\right\rangle$, thus the initial alignment is just as uninformative as the isotropic case providing no improvement. Moreover, a potentially large condition number may hurt the sample complexity in this case. On the other hand, when $r_1 < r_2 < 2r_1$ we have $\left\langle \overline{\boldsymbol{w}}^0, \overline{\boldsymbol{u}}\right\rangle \asymp \kappa \langle \boldsymbol{u}, \boldsymbol{\theta}\rangle \langle \boldsymbol{w}^0, \boldsymbol{\theta}\rangle$, and $r_2 > 2r_1$ leads to $\left\langle \overline{\boldsymbol{w}}^0, \overline{\boldsymbol{u}}\right\rangle \asymp \sqrt{\kappa}\langle \boldsymbol{w}^0, \boldsymbol{\theta}\rangle$, thus large $\kappa$ or $\langle \boldsymbol{u}, \boldsymbol{\theta}\rangle$ in this regime may improve the sample complexity.

# 4 Implications to Neural Networks and Further Improvements

## 4.1 Improving Sample Complexity via Preconditioning

We now demonstrate that preconditioning the training dynamics with $\hat{\boldsymbol{\Sigma}}^{-1}$ can remove the dependency on $\varkappa(\boldsymbol{\Sigma})$, ultimately improving the sample complexity. Consider the preconditioned gradient flow

$$\frac{\mathrm{d}\boldsymbol{w}^t}{\mathrm{d}t} = -\eta(\boldsymbol{w}^t)\hat{\boldsymbol{\Sigma}}^{-1}\nabla\hat{\mathcal{R}}(\boldsymbol{w}^t) \quad \text{with} \quad \eta(\boldsymbol{w}) = \|\hat{\boldsymbol{\Sigma}}^{1/2}\boldsymbol{w}\|^2. \tag{4.1}$$

We have the following alignment result.

**Theorem 7.** *Consider the same setting as Theorem 5, and assume we run the preconditioned empirical gradient flow* (4.1) *with number of samples*

$$n \gtrsim d\Big\{\langle\overline{\boldsymbol{w}}^0,\overline{\boldsymbol{u}}\rangle^{2(1-s)} \vee \varepsilon^{-2}\Big\},$$

*where $\gtrsim$ hides poly-logarithmic factors of d. Then, for $T \asymp \tau_s(\langle\overline{\boldsymbol{w}}^0,\overline{\boldsymbol{u}}\rangle) + \ln(1/\varepsilon)$, we have*

$$\langle\overline{\boldsymbol{w}}^T,\overline{\boldsymbol{u}}\rangle \geq 1 - \varepsilon,$$

*with probability at least $1 - c_1 d^{-c_2}$ for some universal constants $c_1, c_2 > 0$.*

Preconditioning removes the condition number dependence, which is particularly important in the spiked model case where this quantity can be large.

**Corollary 8.** *Consider the setting of Theorem 7, and assume we run the preconditioned empirical gradient flow* (4.1) *for the $(\kappa, \boldsymbol{\theta})$-spiked model where $\langle\boldsymbol{u},\boldsymbol{\theta}\rangle \asymp d^{-r_1}$ and $\kappa \asymp d^{r_2}$ with $r_1 \in [0, 1/2]$ and $r_2 \in [0, 1]$. Suppose $\boldsymbol{w}^0$ is sampled uniformly from $\mathbb{S}^{d-1}$. Then, when conditioned on $\langle\overline{\boldsymbol{w}}^0,\overline{\boldsymbol{u}}\rangle > 0$, the sample complexity of Theorem 7 reads*

$$n \gtrsim \begin{cases} d\big(d^{s-1} \vee \varepsilon^{-2}\big) & 0 < r_2 < r_1 \\ d\big(d^{(s-1)(1-2(r_2-r_1))} \vee \varepsilon^{-2}\big) & r_1 < r_2 < 2r_1, \\ d\big(d^{(s-1)(1-r_2)} \vee \varepsilon^{-2}\big) & 2r_1 < r_2 < 1 \end{cases} \tag{4.2}$$

*where $\gtrsim$ hides poly-logarithmic factors of d.*

The above result improves upon Corollary 6; thus, making a case for preconditioning in practice. The complexity results also strictly improve upon the $\tilde{\mathcal{O}}(d^s)$ complexity in the isotropic case [BBSS22] when $r_2 > r_1$. Further, for any $\nu > 0$, we can obtain the complexity of $\tilde{\mathcal{O}}(d^{1+\nu})$ (nearly linear in dimension) when $r_2 > 1 - \nu/(s-1)$ and $r_2 > 2r_1$ or $r_1 + 1/2(1 - \nu/(s-1)) < r_2 < 2r_1$. In addition to the remarks of Corollary 6, we note that the complexity is independent of both $r_1$ and $r_2$ when $r_2 < r_1$ (cf. Figure 1 hard regime), i.e. the spike magnitude and the spike-target alignment have no effect on the complexity unless $r_2 \geq r_1$.

Under the spiked covariance model, one could improve the above results by instead using spectral initialization, i.e. initializing at $\boldsymbol{\theta}$, which can be estimated from unlabeled data. Assuming perfect access to $\boldsymbol{\theta}$, using the statement of Theorems 5 and 7, this initialization would imply a sample complexity of $\tilde{\mathcal{O}}(d^{1+2r_2+((s-1)(2r_1-r_2)\vee 0)})$ without and $\tilde{\mathcal{O}}(d^{1+((s-1)(2r_1-r_2)\vee 0)})$ with preconditioning.

## 4.2 Two-layer neural networks and learning the link function

Our main focus so far was learning the target direction $\boldsymbol{u}$. Next, we consider learning the unknown link function with a neural network, providing a complete learnability result for single index models.

We use Algorithm 1 and train the first-layer of the neural network with either the empirical gradient flow (3.5) or the preconditioned version (4.1). Then, we randomly choose the bias units and minimize the second layer weights using another gradient flow. Our goal is to track the sample complexity $n$ needed to learn the single index target which we compare against the results of [BBSS22]. We highlight that layer-wise training in Algorithm 1 is frequently employed in the literature [BES$^+$22, BBSS22, DLS22, MHPG$^+$23] and in particular [BBSS22] also used gradient flow for training.

**Algorithm 1** Layer-wise training of a two-layer ReLU network with gradient flow (GF).

---

**Input:** $\boldsymbol{w}^0 \in \mathbb{R}^d, T, T', \Delta, \lambda \in \mathbb{R}_+$ and data $\{(\boldsymbol{x}^{(i)}, y^{(i)})\}_{i=1}^n$.

1: Train the first layer weights $\boldsymbol{W}_j^T$ using the GF (3.5) or the preconditioned GF (4.1).

2: Normalize the weights $\boldsymbol{W}_j^T := \boldsymbol{W}_j^T / \|\hat{\boldsymbol{\Sigma}}^{1/2} \boldsymbol{W}_j^T\|$ for every $1 \le j \le m$.

3: Let $b_j \overset{i.i.d.}{\sim} \text{Unif}(-\Delta, \Delta)$ and $a_j^0 = 1/m$ for $1 \le j \le m$.

4: Train the second layer weights $\boldsymbol{a}^{T'}$ via the gradient flow

$$\frac{\mathrm{d}\boldsymbol{a}^t}{\mathrm{d}t} = -\nabla_{\boldsymbol{a}} \left\{ \frac{1}{2n} \sum_{i=1}^n (\hat{y}(\boldsymbol{x}^{(i)}; \boldsymbol{W}^T, \boldsymbol{a}^t, \boldsymbol{b}) - y^{(i)})^2 + \frac{\lambda \|\boldsymbol{a}^t\|^2}{2} \right\}.$$

5: **return** $(\boldsymbol{W}^T, \boldsymbol{a}^{T'}, \boldsymbol{b})$.

---

**Theorem 9.** *Let $g$ be twice weakly differentiable with information exponent $s$ and assume $g''$ has at most polynomial growth. Suppose $\phi$ is the ReLU activation, Assumption 2 holds and we run Algorithm 1 with $\boldsymbol{w}^0$ initialized uniformly over $\mathbb{S}^{d-1}$. For any $\varepsilon > 0$, let $n$ and $T$ be chosen according to Theorem 5 when we run the gradient flow (3.5) and Theorem 7 when we run the preconditioned gradient flow (4.1). Then, for $\Delta \asymp \sqrt{\ln(nd)}$, some regime of $\lambda$ given by (E.3) and sufficiently large $T'$ given by (E.4), we have*

$$\mathbb{E}_{(\boldsymbol{x},y)} \left[ \left( \hat{y}(\boldsymbol{x}; \boldsymbol{W}^T, \boldsymbol{a}^{T'}, \boldsymbol{b}) - y \right)^2 \right] \le C_1 \, \mathbb{E}[\epsilon^2] + C_2(\varepsilon + 1/m), \tag{4.3}$$

*conditioned on $\langle \overline{\boldsymbol{w}}^0, \overline{\boldsymbol{u}} \rangle > 0$ with probability at least $0.99$ over the randomness of the dataset, biases, and initialization, where $C_1$ is a universal constant and $C_2$ hides $\text{polylog}(m, n, d)$ factors.*

The next result immediately follows from the previous theorem together with Corollaries 6 & 8.

**Corollary 10.** *In the setting of Theorem 9, if $\boldsymbol{\Sigma}$ follows the $(\kappa, \boldsymbol{\theta})$-spiked model, the sample complexity $n$ is given by (3.7) if we use the empirical gradient flow and (4.2) if we use the preconditioned version.*

We remark that for fixed $\varepsilon$, the sample complexity to learn $g$ in the isotropic case is $\tilde{\mathcal{O}}(d^s)$ [BBSS22]. Under the spiked model, if we assume that $r_2$ is sufficiently large and $r_1$ is sufficiently small as discussed in the previous section, Corollary 10 improves this rate to either (3.7) when the empirical gradient flow is used without preconditioning or to (4.2) with preconditioning.

## 5 Technical Overview

In this section, we briefly discuss the key intuitions that lead to the proof of our main results. We first review the case $\boldsymbol{\Sigma} = \mathbf{I}_d$, where we have the following decomposition for population loss

$$R(\boldsymbol{w}) := \tfrac{1}{2} \mathbb{E}\left[(\phi(\langle \boldsymbol{w}, \boldsymbol{x} \rangle) - y)^2\right] = \tfrac{1}{2} \|\phi\|_\gamma^2 + \tfrac{1}{2} \mathbb{E}[y^2] - \mathbb{E}[\phi(\langle \boldsymbol{w}, \boldsymbol{x} \rangle) g(\langle \boldsymbol{u}, \boldsymbol{x} \rangle)]. \tag{5.1}$$

Notice that the only term contributing to the population gradient is the last term which measures the correlation between $\phi$ and $g$. Following the gradient flow and applying Stein's lemma yields

$$\frac{\mathrm{d}\langle \boldsymbol{w}^t, \boldsymbol{u} \rangle}{\mathrm{d}t} = \mathbb{E}\left[\phi'(\langle \boldsymbol{w}^t, \boldsymbol{x} \rangle) g'(\langle \boldsymbol{u}, \boldsymbol{x} \rangle)\right](1 - \langle \boldsymbol{w}^t, \boldsymbol{u} \rangle^2) = (1 - \langle \boldsymbol{w}^t, \boldsymbol{u} \rangle^2) \sum_{j \ge s} j \alpha_j \beta_j \langle \boldsymbol{w}^t, \boldsymbol{u} \rangle^{j-1},$$

where the second identity follows from the Hermite expansion; see also [EDB16, EBD19]. Assume $\alpha_s \beta_s > 0$ to ensure that the population dynamics will move towards $\boldsymbol{u}$ at least near initialization. When replacing the population gradient with a full-batch gradient, we need the estimation noise to be smaller than the signal existing in the gradient. When $\langle \boldsymbol{w}^0, \boldsymbol{u} \rangle \ll 1$, this signal is roughly of the order $\langle \boldsymbol{w}^0, \boldsymbol{u} \rangle^{s-1}$. As the uniform concentration error over $\mathbb{S}^{d-1}$ scales with $\sqrt{d/n}$, we need $n \asymp d \langle \boldsymbol{w}^0, \boldsymbol{u} \rangle^{2(s-1)}$ to ensure the signal remains dominant and $\boldsymbol{w}^t$ moves towards $\boldsymbol{u}$. When $\boldsymbol{w}^0$ is initialized uniformly over $\mathbb{S}^{d-1}$ this translates to a sample complexity of $n \asymp d^s$, which is indeed obtained by [BBSS22] via similar arguments.

However, the behavior of the spherical dynamics entirely changes when we move to the anisotropic case. Suppose $\Sigma$ follows a $(\kappa, \boldsymbol{u})$-spiked model and $\phi$ is ReLU. Using Lemma 12, it is easy to show that with the spherical gradient flow, the alignment obeys the following ODE

$$\frac{\mathrm{d}\langle \boldsymbol{w}^t, \boldsymbol{u}\rangle}{\mathrm{d}t} = \left\{ \mathbb{E}\big[\phi'(\langle \overline{\boldsymbol{w}}^t, \boldsymbol{z}\rangle)g'(\langle \overline{\boldsymbol{u}}, \boldsymbol{z}\rangle)\big] - \frac{\kappa \tilde{\psi}_{\phi,g}(\boldsymbol{w}^t)}{1+\kappa}\langle \boldsymbol{w}^t, \boldsymbol{u}\rangle \right\}(1 - \langle \boldsymbol{w}^t, \boldsymbol{u}\rangle^2),$$

where $\tilde{\psi}_{\phi,g}(\boldsymbol{w}^t)$ is introduced in Lemma 12. The additional $\tilde{\psi}_{\phi,g}(\boldsymbol{w}^t)$ term creates a repulsive force towards the equator $\langle \boldsymbol{w}^t, \boldsymbol{u}\rangle = 0$. The presence of this term is due to the fact that unlike (5.1), the term $\mathbb{E}\big[\phi(\langle \boldsymbol{w}^t, \boldsymbol{u}\rangle)^2\big]$ is no longer independent of $\boldsymbol{w}$ and cannot be replaced by $\|\phi\|_\gamma^2$. When $\boldsymbol{w}^0$ is initialized uniformly over $\mathbb{S}^{d-1}$ and $\Omega(1) \leq \kappa \leq \mathcal{O}(d)$, we have $\langle \overline{\boldsymbol{w}}^0, \overline{\boldsymbol{u}}\rangle \asymp \sqrt{\kappa}\langle \boldsymbol{w}^0, \boldsymbol{u}\rangle$. Furthermore, at this initialization $\tilde{\psi}_{\phi,g}(\boldsymbol{w}^0) \approx 1/2$. Therefore,

$$\frac{\mathrm{d}\langle \boldsymbol{w}^t, \boldsymbol{u}\rangle}{\mathrm{d}t} \approx \left\{ s\alpha_s\beta_s(\sqrt{\kappa}\langle \boldsymbol{w}^0, \boldsymbol{u}\rangle)^{s-1} - \frac{\langle \boldsymbol{w}^0, \boldsymbol{u}\rangle}{2} \right\}.$$

Hence the dynamics is trapped at $|\langle \boldsymbol{w}^t, \boldsymbol{u}\rangle| = \mathcal{O}(1/\sqrt{d})$ for all $t > 0$ as long as $\kappa = \mathcal{O}(d^{1-1/(s-1)})$.

To remove the repulsive force in the spherical dynamics, we can directly normalize the input of $\phi$. As demonstrated by (3.1), once again the only term that varies with $\boldsymbol{w}$ would be the correlation loss. Specifically, using the result of Lemma 3, in the population limit we can track $\langle \overline{\boldsymbol{w}}^t, \overline{\boldsymbol{u}}\rangle$ via

$$\frac{\mathrm{d}\langle \overline{\boldsymbol{w}}^t, \overline{\boldsymbol{u}}\rangle}{\mathrm{d}t} = \mathbb{E}[\phi'(\langle \overline{\boldsymbol{w}}, \boldsymbol{z}\rangle)g'(\langle \overline{\boldsymbol{u}}, \boldsymbol{z}\rangle)]\langle \overline{\boldsymbol{u}}_\perp^t, \Sigma \overline{\boldsymbol{u}}_\perp^t\rangle, \tag{5.2}$$

where $\overline{\boldsymbol{u}}_\perp^t := \overline{\boldsymbol{u}} - \langle \overline{\boldsymbol{w}}^t, \overline{\boldsymbol{u}}\rangle\overline{\boldsymbol{w}}^t$. Thus, the strength of the signal at initialization is of order $\langle \overline{\boldsymbol{w}}^0, \overline{\boldsymbol{u}}\rangle^{s-1}/\varkappa(\Sigma)$, which after controlling the error in the estimate of $\hat{\Sigma}$ and in the estimate of population gradient using finitely many samples, leads to the sample complexity $n \asymp d\varkappa(\Sigma)^2\langle \overline{\boldsymbol{w}}^0, \overline{\boldsymbol{u}}\rangle^{2(1-s)}$. Importantly, $\Sigma$ can incude a much stronger initial alignment $\langle \overline{\boldsymbol{w}}^0, \overline{\boldsymbol{u}}\rangle$ than the isotropic case $\langle \boldsymbol{w}^0, \boldsymbol{u}\rangle$, which is emphasized in Corollary 6. Using preconditioning will further remove the dependency on the condition number of $\Sigma$.

# 6 Conclusion

We studied the dynamics of gradient flow to learn single index models when the input data covariance may contain additional structure. Under a spiked model for the covariance matrix, we showed that using spherical gradient flow, as an example of a covariance-agnostic training mechanism employed in the recent literature, is unable to learn the target direction of the single index model even when the spike and the target directions are identical. In contrast, we showed that an appropriate weight normalization removes this problem and successfully recovers the target direction. Moreover, depending on the alignment between the covariance structure and the target direction, the sample complexity can improve upon the isotropic setting, while also outperforming lower bounds for rotationally-invariant kernels. This phenomenon is due to the additional information about the target direction contained in the covariance matrix which improves the effective alignment at initialization. Additionally, we showed that a simple preconditioning of the gradient flow using the inverse empirical covariance can improve the sample complexity, achieving almost linear rate in certain settings.

We outline a few limitations of our current work and discuss directions for future research.

- While studying single index models provides a pathway to a general understanding of feature learning with structured covariance, considering multi-index models can provide a more complete picture [PSE22], e.g. by establishing incremental learning dynamics [ABAM23]. We leave the problem of learning multi-index models under structured input as an interesting future direction.
- Gradient flow under squared loss can be seen as an example of a Correlational Statistical Query (CSQ) algorithm [BF02, Rey20], i.e. an algorithm that only accesses noisy estimates of expected correlation queries from the model. Understanding the limitations of learning single index models under a structured input through a CSQ lower bound perspective is another important direction that would complement our results in this paper.
- When training the first layer, we considered a somewhat unconventional initialization and relied on the symmetry it induces. It is interesting to consider cases where we train a network with multiple neurons starting from a more standard initialization which can help relax Assumption 2.

## Acknowledgments

The authors thank Alberto Bietti and Zhichao Wang for discussions and feedback on the manuscript. TS was partially supported by JSPS KAKENHI (20H00576) and JST CREST (JPMJCR2015). MAE was partially supported by NSERC Grant [2019-06167], CIFAR AI Chairs program, CIFAR AI Catalyst grant.

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

## A  Concurrent Works

In this paragraph we briefly summarize a few relevant results concurrent to our submission. [BMZ23] provided a precise analysis of the two-timescale dynamics in learning a single index model with information exponent $s = 1$. [MHD+23] considered the learning of a single index target with $s = 2$ using a neural network in the mean-field regime. [BPVZ23] extended the information exponent-based characterization of online SGD to input data beyond Gaussian. [DNGL23] showed that a gradient-smoothed dynamics can improve the sample complexity and match the CSQ lower bound. Finally, beyond the single-index setting, [DKL+23, SWON23, CWPPS23] considered learning low-dimensional target functions supported on $k > 1$ dimensions via gradient-based feature learning.

## B  Background on Hermite Expansion

The normalized Hermite polynomials $\{h_j\}_{j \geq 0}$ given by (2.4) provide an orthonormal basis for $L^2(\gamma)$, thus for every $f \in L^2(\gamma)$ we have

$$f = \sum_{j=0}^{\infty} \langle f, h_j \rangle_\gamma h_j,$$

where $\langle f, h_j \rangle_\gamma := \mathbb{E}_{z \sim \mathcal{N}(0,1)}[f(z)h_j(z)]$. We will commonly invoke the following well-known properties of Hermite polynomials. If $j \geq 1$, then $h_j' = \sqrt{j} h_{j-1}$, where $h_j'$ stands for the derivative of $h_j$. Furthermore, if $z_1$ and $z_2$ are two standard Gaussian random variables with $\mathbb{E}[z_1 z_2] = \rho$, then $\mathbb{E}[h_i(z_1)h_j(z_2)] = \delta_{ij}\rho^j$ where $\delta_{ij}$ is the Kronecker delta. We refer the interested reader to [O'D14, Chapter 11.2] for additional discussions and properties of these polynomials.

We will now discuss how our Definition 1 relates to the original definition of information exponent of [BAGJ21]. In their setting, they assume the true data distribution $\mathbb{P}_{\boldsymbol{u}}$ is parameterized by some unit vector $\boldsymbol{u} \in \mathbb{S}^{d-1}$, and we know the parametric family $\{\mathbb{P}_{\boldsymbol{w}}\}_{\boldsymbol{w} \in \mathbb{S}^{d-1}}$; thus the problem is to estimate the direction $\boldsymbol{u}$. Furthermore, they assume the population loss, which is the expectation of some per-sample loss, has spherical symmetry, i.e. the population loss $R(\boldsymbol{w})$ can be written as $R(\boldsymbol{w}) = \tilde{R}(\langle \boldsymbol{w}, \boldsymbol{u} \rangle)$. Then, [BAGJ21, Definition 1.2] defines the information exponent to be the degree of the first non-zero coefficient of $\tilde{R}$ in its Taylor expansion around the origin. In other words, we say $R$ has information exponent $s$ if

$$\begin{cases} \frac{\mathrm{d}^k \tilde{R}}{\mathrm{d}z^k}(0) = 0 & 1 \leq k < s \\ \frac{\mathrm{d}^k \tilde{R}}{\mathrm{d}z^k}(0) = -c < 0 & k = s \\ \left| \frac{\mathrm{d}^k \tilde{R}}{\mathrm{d}z^k}(z) \right| \leq C & k > s, \forall z \in [-1, 1] \end{cases},$$

where $C, c > 0$ are universal constants. To specialize the above abstract definition to the Gaussian case, consider the setting where the input data is standard Gaussian $\boldsymbol{x} \sim \mathcal{N}(0, \mathbf{I}_d)$ and the problem is to estimate $\boldsymbol{u} \in \mathbb{S}^{d-1}$ given a response variable $y = f(\langle \boldsymbol{u}, \boldsymbol{x} \rangle)$ with known $f$. Via the Hermite expansion of $f$, one can write

$$\tilde{R}(\langle \boldsymbol{w}, \boldsymbol{u} \rangle) := \frac{1}{2} \mathbb{E}\left[ (f(\langle \boldsymbol{w}, \boldsymbol{x} \rangle) - f(\langle \boldsymbol{u}, \boldsymbol{x} \rangle))^2 \right] = -\sum_{j \geq 1} \langle f, h_j \rangle_\gamma^2 \langle \boldsymbol{w}, \boldsymbol{u} \rangle^j + \text{const.}$$

Thus, the information exponent of $\tilde{R}$ is indeed the degree of the first non-zero term in the Hermite expansion of $f$.

Now consider the general case where $\boldsymbol{x} \sim \mathcal{N}(0, \boldsymbol{\Sigma})$. The spherical symmetry assumed in [BAGJ21] no longer holds. However, after proper normalization of weights, if we consider the population loss

$$R(\boldsymbol{w}) := \frac{1}{2} \mathbb{E}\left[ \left( f\left( \frac{\langle \boldsymbol{w}, \boldsymbol{x} \rangle}{\|\boldsymbol{\Sigma}^{1/2}\boldsymbol{w}\|} \right) - f\left( \frac{\langle \boldsymbol{u}, \boldsymbol{x} \rangle}{\|\boldsymbol{\Sigma}^{1/2}\boldsymbol{u}\|} \right) \right)^2 \right],$$

then $R(\boldsymbol{w}) = \tilde{R}\left( \frac{\langle \boldsymbol{w}, \boldsymbol{\Sigma} \boldsymbol{u} \rangle}{\|\boldsymbol{\Sigma}^{1/2}\boldsymbol{w}\| \|\boldsymbol{\Sigma}^{1/2}\boldsymbol{u}\|} \right)$. Indeed, a close examination of the arguments of [BAGJ21] reveals that for their results to hold, the proper symmetry to consider is the ellipsoidal symmetry, and

the proper definition of information exponent is the degree of the first non-zero term in the Hermite expansion of $\tilde{R}$, which reads

$$\tilde{R}(z) = -\sum_{j \geq 1} \langle f, h_j \rangle^2 z^j + \text{const.}$$

Once again, we can consistently define the information exponent to be the degree of the first non-zero term in the Hermite expansion of $f$, as long as the input is Gaussian (potentially anisotropic).

## C   Proofs of Section 2.1

Before beginning our main discussions, we state the following lemma which is a generalization of Stein's lemma (Gaussian integration by parts), and will help obtain a closed-form expression for the population gradient. We refer to [Erd15, MHPG$^+$23] for similar statements.

**Lemma 11.** *Let $f, g : \mathbb{R} \to \mathbb{R}$ with $g$ weakly differentiable. Suppose $z \sim \mathcal{N}(0, \mathbf{I}_d)$. Then, for any $w, u \in \mathbb{S}^{d-1}$, we have*

$$\mathbb{E}[f(\langle w, z \rangle) g(\langle u, z \rangle) z] = \mathbb{E}[f(\langle w, z \rangle) g'(\langle u, z \rangle)] u + \mathbb{E}[f(\langle w, z \rangle) \{g(\langle u, z \rangle) \langle w, z \rangle - g'(\langle u, z \rangle) \langle u, w \rangle\}] w.$$

**Proof.** Consider the conditional distribution $z | \langle w, z \rangle \sim \mathcal{N}(\overline{\mu}, \overline{\Sigma})$, where

$$\overline{\mu} = \langle w, z \rangle w \quad \text{and} \quad \overline{\Sigma} = \mathbf{I}_d - w w^\top.$$

Recall that Stein's lemma (Gaussian integration by parts) states that when $\overline{z} \sim \mathcal{N}(\overline{\mu}, \overline{\Sigma})$, then

$$\mathbb{E}[g(\overline{z}) \overline{z}] = \mathbb{E}[g(\overline{z})] \overline{\mu} + \overline{\Sigma} \, \mathbb{E}[\nabla g(\overline{z})].$$

Hence,

$$\mathbb{E}[g(\langle u, z \rangle) z \mid \langle w, z \rangle] = \mathbb{E}[g(\langle u, z \rangle) \mid \langle w, z \rangle] \langle w, z \rangle w + (\mathbf{I}_d - w w^\top) \, \mathbb{E}[g'(\langle u, z \rangle) \mid \langle w, z \rangle] u.$$

Applying the tower property of conditional expectation and rearranging the terms yields the desired result. $\qquad \square$

We are now ready to state and prove the expression for the population gradient when using the ReLU activation.

**Lemma 12.** *Suppose $\phi$ is the ReLU activation and $g$ is weakly differentiable. Let $z \sim \mathcal{N}(0, \mathbf{I}_d)$. Define $\overline{w} := \frac{\Sigma^{1/2} w}{\|\Sigma^{1/2} w\|}$ and similarly define $\overline{u}$. Then*

$$\nabla R(w) = \Sigma \left\{ \tilde{\psi}_{\phi, g}(w) w + \tilde{\zeta}_{\phi, g}(w) u \right\}$$

*where*

$$\tilde{\psi}_{\phi, g}(w) := \frac{\mathbb{E}[-\phi(\langle \overline{w}, z \rangle) g(\langle \overline{u}, z \rangle) + \phi'(\langle \overline{w}, z \rangle) g'(\langle \overline{u}, z \rangle) \langle \overline{u}, \overline{w} \rangle]}{\|\Sigma^{1/2} w\|} + \frac{1}{2},$$

*and*

$$\tilde{\zeta}_{\phi, g}(w) := -\frac{\mathbb{E}[\phi'(\langle \overline{w}, z \rangle) g'(\langle \overline{u}, z \rangle)]}{\|\Sigma^{1/2} u\|}.$$

**Proof.** Notice that the population risk is given by

$$R(w) = \frac{1}{2} \mathbb{E}\left[ \left( \phi(\langle w, x \rangle) - g\left( \frac{\langle u, x \rangle}{\|\Sigma^{1/2} u\|} \right) \right)^2 \right] + \frac{\mathbb{E}[\epsilon^2]}{2} = \frac{1}{2} \mathbb{E}[\phi(\langle w, x \rangle)^2] - \mathbb{E}[\phi(\langle w, x \rangle) g] + \frac{\mathbb{E}[y^2]}{2},$$

Notice that $x = \Sigma^{1/2} z$. By the homogeneity of ReLU, we can rewrite the first term as

$$\mathbb{E}[\phi(\langle w, x \rangle)^2] = w^\top \Sigma w \|\phi\|_\gamma^2 = \frac{w^\top \Sigma w}{2}.$$

Then we have,

$$\nabla R(\boldsymbol{w}) = \frac{\boldsymbol{\Sigma}\boldsymbol{w}}{2} - \underbrace{\mathbb{E}\left[\phi(\langle\boldsymbol{w},\boldsymbol{x}\rangle)'g\left(\frac{\langle\boldsymbol{u},\boldsymbol{x}\rangle}{\|\boldsymbol{\Sigma}^{1/2}\boldsymbol{u}\|}\right)\boldsymbol{x}\right]}_{=:v(\boldsymbol{w})}$$

Note that $\phi'(\langle\boldsymbol{w},\boldsymbol{x}\rangle) = \phi'(\langle\overline{\boldsymbol{w}},\boldsymbol{z}\rangle)$. Then,

$$\begin{aligned}
v(\boldsymbol{w}) &= \boldsymbol{\Sigma}^{1/2}\,\mathbb{E}[\phi'(\langle\overline{\boldsymbol{w}},\boldsymbol{z}\rangle)'g(\langle\overline{\boldsymbol{u}},\boldsymbol{z}\rangle)\boldsymbol{z}]\\
&= \boldsymbol{\Sigma}^{1/2}\{\mathbb{E}[\phi'(\langle\overline{\boldsymbol{w}},\boldsymbol{z}\rangle)g(\langle\overline{\boldsymbol{u}},\boldsymbol{z}\rangle)]\overline{\boldsymbol{u}} + \mathbb{E}[\phi(\langle\overline{\boldsymbol{w}},\boldsymbol{z}\rangle g(\langle\overline{\boldsymbol{u}},\boldsymbol{z}\rangle) - \phi'(\langle\overline{\boldsymbol{w}},\boldsymbol{z}\rangle)g'(\langle\overline{\boldsymbol{u}},\boldsymbol{z}\rangle)\langle\overline{\boldsymbol{u}},\overline{\boldsymbol{w}}\rangle]\overline{\boldsymbol{w}}\}\\
&= \boldsymbol{\Sigma}\left\{\frac{\mathbb{E}[\phi'(\langle\overline{\boldsymbol{w}},\boldsymbol{z}\rangle)g'(\langle\overline{\boldsymbol{u}},\boldsymbol{z}\rangle)]}{\|\boldsymbol{\Sigma}^{1/2}\boldsymbol{u}\|}\boldsymbol{u} + \frac{\mathbb{E}[\phi(\langle\overline{\boldsymbol{w}},\boldsymbol{z}\rangle g(\langle\overline{\boldsymbol{u}},\boldsymbol{z}\rangle) - \phi'(\langle\overline{\boldsymbol{w}},\boldsymbol{z}\rangle)g'(\langle\overline{\boldsymbol{u}},\boldsymbol{z}\rangle)\langle\overline{\boldsymbol{u}},\overline{\boldsymbol{w}}\rangle]}{\|\boldsymbol{\Sigma}^{1/2}\boldsymbol{w}\|}\boldsymbol{w}\right\}
\end{aligned}$$

where we used Lemma 11 and the fact that $\phi'(z)z = \phi(z)$. Therefore,

$$\nabla R(\boldsymbol{w}) = \boldsymbol{\Sigma}\left\{\tilde{\psi}_{\phi,g}(\boldsymbol{w})\boldsymbol{w} + \tilde{\zeta}_{\phi,g}(\boldsymbol{w})\boldsymbol{u}\right\}$$

which concludes the proof. $\qquad\square$

Particularly, the above lemma yields the following corollary for the spherical dynamics in the population limit.

**Corollary 13.** *Suppose $\{\boldsymbol{w}^t\}_{t\geq 0}$ is a solution to the population spherical gradient flow* (2.5), *$\phi$ is the ReLU activation, and $\boldsymbol{\Sigma}$ follows the $(\kappa,\boldsymbol{\theta})$-spiked model. Then,*

$$\begin{aligned}
\frac{\mathrm{d}\langle\boldsymbol{w}^t,\boldsymbol{u}\rangle}{\mathrm{d}t} &= -\frac{\tilde{\zeta}_{\phi,g}(\boldsymbol{w}^t)(1 - \langle\boldsymbol{w}^t,\boldsymbol{u}\rangle^2)}{1+\kappa}\\
&\quad - \frac{\kappa\left\{\tilde{\psi}_{\phi,g}(\boldsymbol{w}^t)\langle\boldsymbol{w}^t,\boldsymbol{\theta}\rangle + \tilde{\zeta}_{\phi,g}(\boldsymbol{w}^t)\langle\boldsymbol{u},\boldsymbol{\theta}\rangle\right\}}{1+\kappa}(\langle\boldsymbol{\theta},\boldsymbol{u}\rangle - \langle\boldsymbol{w}^t,\boldsymbol{\theta}\rangle\langle\boldsymbol{w}^t,\boldsymbol{u}\rangle).
\end{aligned} \tag{C.1}$$

### C.1 Proof of Theorem 2

Plugging in $\boldsymbol{\theta} = \boldsymbol{u}$ in Corollary 13, we obtain

$$\frac{\mathrm{d}\langle\boldsymbol{w}^t,\boldsymbol{u}\rangle}{\mathrm{d}t} = -\left\{\tilde{\zeta}_{\phi,g}(\boldsymbol{w}^t) + \frac{\kappa\tilde{\psi}_{\phi,g}(\boldsymbol{w}^t)\langle\boldsymbol{w}^t,\boldsymbol{u}\rangle}{1+\kappa}\right\}(1 - \langle\boldsymbol{u},\boldsymbol{w}^t\rangle^2). \tag{C.2}$$

To prove the statement of the theorem, we will show that whenever

$$|\langle\boldsymbol{w}^t,\boldsymbol{u}\rangle| \leq \frac{C}{\sqrt{d}},$$

we will have $\frac{\mathrm{d}\langle\boldsymbol{w}^t,\boldsymbol{u}\rangle^2}{\mathrm{d}t} < 0$, thus when initialized from $\mathcal{O}(1/\sqrt{d})$, $\langle\boldsymbol{w}^t,\boldsymbol{u}\rangle$ can never escape the saddle point near the equator $\langle\boldsymbol{w},\boldsymbol{u}\rangle = 0$.

Recall from the properties of the Hermite expansion in Appendix B that

$$g' = \sum_{j\geq 1}\sqrt{j}\alpha_j h_{j-1} \quad\text{and}\quad \phi' = \sum_{j\geq 1}\sqrt{j}\beta_j h_{j-1}.$$

Since we additionally assume $\mathbb{E}[g] = \alpha_0 = 0$, by the definition of $\tilde{\psi}_{\phi,g}$ in Lemma 12 and the properties of Hermite expansion discussed in Appendix B, we have

$$\tilde{\psi}_{\phi,g}(\boldsymbol{w}^t) = \frac{\sum_{j\geq s}(j-1)\alpha_j\beta_j\langle\overline{\boldsymbol{w}}^t,\boldsymbol{u}\rangle^j}{\|\boldsymbol{\Sigma}^{1/2}\boldsymbol{w}\|} + \frac{1}{2},$$

and similarly

$$\tilde{\zeta}_{\phi,g}(\boldsymbol{w}^t) := -\sum_{j\geq s}j\alpha_j\beta_j\langle\overline{\boldsymbol{w}}^t,\boldsymbol{u}\rangle^{j-1}.$$

Thus we obtain

$$\frac{\mathrm{d}\langle \boldsymbol{w}^t, \boldsymbol{u}\rangle^2}{\mathrm{d}t} = 2F(\boldsymbol{w}^t)(1 - \langle \boldsymbol{w}^t, \boldsymbol{u}\rangle^2),$$

where

$$F(\boldsymbol{w}^t) := \sum_{j\geq s} j\alpha_j\beta_j \langle \overline{\boldsymbol{w}}^t, \boldsymbol{u}\rangle^{j-1}\langle \boldsymbol{w}^t, \boldsymbol{u}\rangle - \frac{\kappa}{1+\kappa}\sum_{j\geq s}(j-1)\alpha_j\beta_j\langle \overline{\boldsymbol{w}}^t, \boldsymbol{u}\rangle^{j+1}\langle \boldsymbol{w}^t, \boldsymbol{u}\rangle - \frac{\kappa\langle \boldsymbol{w}^t, \boldsymbol{u}\rangle^2}{2(1+\kappa)}.$$

We proceed by upper bounding $F$. To do so, first note that

$$\|\boldsymbol{\Sigma}^{1/2}\boldsymbol{w}\| = \sqrt{\frac{1 + \kappa\langle \boldsymbol{w}, \boldsymbol{u}\rangle^2}{1+\kappa}} \geq \sqrt{\frac{1}{1+\kappa}}.$$

To bound the first term of $F$, we have

$$\sum_{j\geq s} j\alpha_j\beta_j\langle \overline{\boldsymbol{w}}^t, \boldsymbol{u}\rangle^{j-1}\langle \boldsymbol{w}^t, \boldsymbol{u}\rangle \leq |\langle \boldsymbol{w}^t, \boldsymbol{u}\rangle||\langle \overline{\boldsymbol{w}}^t, \boldsymbol{u}\rangle|^{s-1}\sum_{j\geq s} j|\alpha_j\beta_j||\langle \overline{\boldsymbol{w}}^t, \boldsymbol{u}\rangle|^{j-s}$$

$$\leq \|\phi'\|_\gamma\|g'\|_\gamma|\langle \overline{\boldsymbol{w}}^t, \boldsymbol{u}\rangle|^{s-1}|\langle \boldsymbol{w}^t, \boldsymbol{u}\rangle|$$

$$\leq \|\phi'\|_\gamma\|g'\|_\gamma(1+\kappa)^{(s-1)/2}|\langle \boldsymbol{w}^t, \boldsymbol{u}\rangle|^s.$$

Similarly,

$$-\frac{\kappa}{1+\kappa}\sum_{j\geq s}(j-1)\alpha_j\beta_j\langle \overline{\boldsymbol{w}}^t, \boldsymbol{u}\rangle^{j+1}\langle \boldsymbol{w}^t, \boldsymbol{u}\rangle \leq \|\phi'\|_\gamma\|g'\|_\gamma(1+\kappa)^{(s+1)/2}|\langle \boldsymbol{w}^t, \boldsymbol{u}\rangle|^{s+2}.$$

Hence, for $\kappa \geq 1$,

$$F(\boldsymbol{w}^t) \leq \|\phi'\|_\gamma\|g'\|_\gamma(1+\kappa)^{(s-1)/2}|\langle \boldsymbol{w}^t, \boldsymbol{u}\rangle|^s\Big(1 + (1+\kappa)\langle \boldsymbol{w}^t, \boldsymbol{u}\rangle^2\Big) - \frac{\langle \boldsymbol{w}^t, \boldsymbol{u}\rangle^2}{4}.$$

Suppose $\kappa < d/C^2 - 1$, then

$$F(\boldsymbol{w}^t) \leq \langle \boldsymbol{w}^t, \boldsymbol{u}\rangle^2\Big(2\|\phi'\|_\gamma\|g'\|_\gamma(1+\kappa)^{(s-1)/2}|\langle \boldsymbol{w}^t, \boldsymbol{u}\rangle|^{s-2} - 1/4\Big)$$

$$\leq \langle \boldsymbol{w}^t, \boldsymbol{u}\rangle^2\Big(2\|\phi'\|_\gamma\|g'\|_\gamma C^{s-2}\sqrt{\frac{(1+\kappa)^{s-1}}{d^{s-2}}} - 1/4\Big).$$

Thus, for any $\kappa$ such that

$$1 \leq \kappa \leq \left\{\frac{d^{\frac{s-2}{s-1}}}{(8C^{s-2}\|\phi'\|_\gamma\|g'\|_\gamma)^{\frac{2}{s-1}}} \wedge \frac{d}{C^2}\right\} - 1$$

and any $\boldsymbol{w}^t$ such that $|\langle \boldsymbol{w}^t, \boldsymbol{u}\rangle| \leq C/\sqrt{d}$, we have $\frac{\mathrm{d}\langle \boldsymbol{w}^t, \boldsymbol{u}\rangle^2}{\mathrm{d}t} \leq 0$, hence $\sup_{t\geq 0}|\langle \boldsymbol{w}^t, \boldsymbol{u}\rangle| \leq C/\sqrt{d}$, as long as the above holds true at initialization.

Finally, we will show $|\langle \boldsymbol{w}^0, \boldsymbol{u}\rangle| \leq C/\sqrt{d}$ with probability at least 0.99 for a suitable choice of constant $C$. Indeed, this is an elementary concentration of measure result on the unit sphere. For simplicity, we avoid performing sharp probability of failure analysis and only remark that $\mathbb{E}\big[\langle \boldsymbol{w}^0, \boldsymbol{u}\rangle^2\big] = 1/d$, thus by the Markov inequality

$$\mathbb{P}\Big(\langle \boldsymbol{w}^0, \boldsymbol{u}\rangle^2 \geq C^2/d\Big) \leq 1/C^2,$$

hence a choice of $C \geq 10$ suffices, and the proof is complete. $\qquad\square$

## C.2 Extremely Large Spike

In this section, we will show that under extremely large spike, the spherical gradient flow (2.5) can potentially recover the true direction. Namely, we will prove the following proposition.

**Proposition 14.** *Suppose we initialize the spherical population gradient flow* (2.5) *from* $\boldsymbol{w}^0$. *Let* $\phi$ *be the ReLU activation and assume*

$$\langle \phi, g \rangle_\gamma := \mathbb{E}_{z \sim \mathcal{N}(0,1)}[\phi(z)g(z)] = \alpha > 1/2,$$

*and* $\kappa \geq \frac{C}{\langle \boldsymbol{w}^0, \boldsymbol{u} \rangle^2}$ *for a sufficiently large constant* $C > 0$ *depending only on g. Then, the gradient flow on the sphere satisfies*

$$\begin{cases} \langle \boldsymbol{w}^T, \boldsymbol{u} \rangle \geq 1 - \varepsilon & \text{if} \quad \langle \boldsymbol{w}^0, \boldsymbol{u} \rangle > 0 \\ \langle \boldsymbol{w}^T, \boldsymbol{u} \rangle \leq -1 + \varepsilon & \text{if} \quad \langle \boldsymbol{w}^0, \boldsymbol{u} \rangle < 0 \end{cases} \tag{C.3}$$

*whenever*

$$T \geq \frac{1}{\alpha - 1/2} \ln(2/\varepsilon). \tag{C.4}$$

Before proceeding to the proof, we notice that if we uniformly initialize $\boldsymbol{w}^0$ over $\mathbb{S}^{d-1}$, then the typical value for $\langle \boldsymbol{w}^0, \boldsymbol{u} \rangle$ is of order $d^{-1/2}$, meaning that the above proposition asks for $\kappa = \Omega(d)$. This is a regime where lower bounds for the sample complexity of kernel methods are $\Omega(1)$ [DWY21], thus no meaningful separation in terms of dimension dependency of the sample complexity between neural networks and kernel methods is possible, as the problem becomes effectively one-dimensional.

**Proof.** The cases where $\langle \boldsymbol{w}^0, \boldsymbol{u} \rangle > 0$ and $\langle \boldsymbol{w}^0, \boldsymbol{u} \rangle < 0$ are symmetric, thus we only present the proof for the former. Using (C.2), we can write the dynamics on the sphere more explicitly as

$$\frac{\mathrm{d}\langle \boldsymbol{w}^t, \boldsymbol{u} \rangle}{\mathrm{d}t} = \left\{ \underbrace{\frac{\mathbb{E}\big[\phi'(\langle \overline{\boldsymbol{w}}^t, \boldsymbol{z} \rangle)g'(\langle \boldsymbol{u}, \boldsymbol{z} \rangle)\big]}{1 + \kappa \langle \boldsymbol{w}^t, \boldsymbol{u} \rangle^2}}_{=:B_1} + \underbrace{\frac{\kappa \langle \boldsymbol{w}^t, \boldsymbol{u} \rangle \, \mathbb{E}\big[\phi(\langle \overline{\boldsymbol{w}}^t, \boldsymbol{z} \rangle)g(\langle \boldsymbol{u}, \boldsymbol{z} \rangle)\big]}{\sqrt{1+\kappa}\sqrt{1+\kappa \langle \boldsymbol{w}^t, \boldsymbol{u} \rangle^2}}}_{=:B_2} - \frac{\kappa \langle \boldsymbol{w}^t, \boldsymbol{u} \rangle}{2(1+\kappa)} \right\} (1 - \langle \boldsymbol{w}^t, \boldsymbol{u} \rangle^2).$$

Our goal is to study the regime of large $\kappa$, therefore we will bound how much $B_1$ and $B_2$ can deviate from their corresponding $\kappa = \infty$ values. In particular, we have

$$B_1 \geq -\frac{\|\phi'\|_\gamma \|g'\|_\gamma}{1 + \kappa \langle \boldsymbol{w}^t, \boldsymbol{u} \rangle^2} = \frac{-\|g'\|_\gamma/2}{1 + \kappa \langle \boldsymbol{w}^t, \boldsymbol{u} \rangle^2}.$$

Furthermore, assuming $\langle \boldsymbol{w}^t, \boldsymbol{u} \rangle > 0$ and $\langle \phi, g \rangle_\gamma > 0$, let $c_\kappa(\boldsymbol{w}^t) := \frac{\kappa \langle \boldsymbol{w}^t, \boldsymbol{u} \rangle}{\sqrt{1+\kappa}\sqrt{1+\kappa \langle \boldsymbol{w}^t, \boldsymbol{u} \rangle^2}}$. Then, by the Lipschitzness of $\phi$,

$$\begin{aligned} B_2 &= c_\kappa(\boldsymbol{w}^t)\langle \phi, g \rangle_\gamma + c_\kappa(\boldsymbol{w}^t) \, \mathbb{E}\big[(\phi(\langle \overline{\boldsymbol{w}}^t, \boldsymbol{z} \rangle) - \phi(\langle \boldsymbol{u}, \boldsymbol{z} \rangle))g(\langle \boldsymbol{u}, \boldsymbol{z} \rangle)\big] \\ &\geq c_\kappa(\boldsymbol{w}^t)\langle \phi, g \rangle_\gamma - |c_\kappa(\boldsymbol{w}^t)|\|g\|_\gamma \|\overline{\boldsymbol{w}}^t - \boldsymbol{u}\| \\ &\geq c_\kappa(\boldsymbol{w}^t)\langle \phi, g \rangle_\gamma - |c_\kappa(\boldsymbol{w}^t)|\|g\|_\gamma \sqrt{2(1 - \langle \overline{\boldsymbol{w}}^t, \boldsymbol{u} \rangle^2)} \\ &\geq c_\kappa(\boldsymbol{w}^t)\langle \phi, g \rangle_\gamma - \frac{\sqrt{2\kappa}\|g\|_\gamma |\langle \boldsymbol{w}^t, \boldsymbol{u} \rangle|}{1 + \kappa \langle \boldsymbol{w}^t, \boldsymbol{u} \rangle^2}. \end{aligned}$$

where we used $\overline{\boldsymbol{w}}^t = \frac{\boldsymbol{\Sigma}^{1/2}\boldsymbol{w}^t}{\|\boldsymbol{\Sigma}^{1/2}\boldsymbol{w}^t\|}$ in the last step. Suppose $\langle \boldsymbol{w}^t, \boldsymbol{u} \rangle > 0$ (which holds at least on a neighborhood around initialization, and as we will see below holds for all $t > 0$), then,

$$c_\kappa(\boldsymbol{w}^t) \geq \frac{\kappa \langle \boldsymbol{w}^t, \boldsymbol{u} \rangle^2}{1 + \kappa \langle \boldsymbol{w}^t, \boldsymbol{u} \rangle^2}.$$

As a result, we obtain

$$B_1 + B_2 - \frac{\kappa \langle \boldsymbol{w}^t, \boldsymbol{u} \rangle}{2(1+\kappa)} \geq \langle \phi, g \rangle_\gamma \left(1 - \frac{1}{\kappa \langle \boldsymbol{w}^t, \boldsymbol{u} \rangle^2}\right) - \frac{\|g'\|_\gamma/2 + \sqrt{2}\|g\|_\gamma}{\sqrt{\kappa \langle \boldsymbol{w}^t, \boldsymbol{u} \rangle^2}} - \frac{1}{2}.$$

Consequently, the lower bound of the time derivative of $\langle \boldsymbol{w}^t, \boldsymbol{u} \rangle$ becomes larger as $\langle \boldsymbol{w}^t, \boldsymbol{u} \rangle$ increases. Therefore, assuming $\langle \boldsymbol{w}^0, \boldsymbol{u} \rangle > 0$, we only need to control this lower bound at initialization. Assume

$$\kappa \geq \frac{4}{\langle \boldsymbol{w}^0, \boldsymbol{u} \rangle^2 (\alpha - 1/2)} \left\{ \alpha \vee \frac{(\|g'\|_\gamma + \sqrt{8}\|g\|_\gamma)^2}{\alpha - 1/2} \right\}.$$

From this, we conclude that when $\langle \boldsymbol{w}^t, \boldsymbol{u} \rangle > 0$ and $\langle \phi, g \rangle_\gamma = \alpha > 1/2$, we have

$$\frac{\mathrm{d}\langle \boldsymbol{w}^t, \boldsymbol{u} \rangle}{\mathrm{d}t} \geq \frac{\alpha - 1/2}{2}(1 - \langle \boldsymbol{w}^t, \boldsymbol{u} \rangle^2),$$

integration yields the desired result. $\qquad\square$

# D   Proofs of Section 3

We begin by stating the closed-form expression for the population gradient, i.e. the counterpart of Lemma 12 in the normalized setting.

**Lemma 15.** *Consider the population risk $\mathcal{R}(\boldsymbol{w})$ defined by (3.1), recall that*

$$\mathcal{R}(\boldsymbol{w}) = \mathbb{E}\left[ -\phi\left( \frac{\langle \boldsymbol{w}, \boldsymbol{x} \rangle}{\|\boldsymbol{\Sigma}^{1/2}\boldsymbol{w}\|} \right) g(\langle \overline{\boldsymbol{u}}, \boldsymbol{z} \rangle) \right] + \frac{1}{2}\|\phi\|_\gamma^2 + \frac{1}{2}\mathbb{E}[y^2].$$

*Then,*

$$\nabla \mathcal{R}(\boldsymbol{w}) = \frac{\boldsymbol{\Sigma}^{1/2}(\mathbf{I}_d - \overline{\boldsymbol{w}}\,\overline{\boldsymbol{w}}^\top)\zeta_{\phi,g}(\langle \overline{\boldsymbol{w}}, \overline{\boldsymbol{u}} \rangle)\overline{\boldsymbol{u}}}{\|\boldsymbol{\Sigma}^{1/2}\boldsymbol{w}\|}, \tag{D.1}$$

*where*

$$\zeta_{\phi,g}(\langle \overline{\boldsymbol{w}}, \overline{\boldsymbol{u}} \rangle) := -\mathbb{E}[\phi'(\langle \overline{\boldsymbol{w}}, \boldsymbol{z} \rangle)g'(\langle \overline{\boldsymbol{u}}, \boldsymbol{z} \rangle)] = -\sum_{j \geq s} j\alpha_j\beta_j\langle \overline{\boldsymbol{w}}, \overline{\boldsymbol{u}} \rangle^{j-1}. \tag{D.2}$$

**Proof.** Recall from (3.1) that

$$\begin{aligned}
\nabla \mathcal{R}(\boldsymbol{w}) &= \nabla_{\boldsymbol{w}} \mathbb{E}\left[ -\phi\left( \frac{\langle \boldsymbol{w}, \boldsymbol{x} \rangle}{\|\boldsymbol{\Sigma}^{1/2}\boldsymbol{w}\|} \right) g(\langle \overline{\boldsymbol{u}}, \boldsymbol{z} \rangle) \right] \\
&= -\frac{1}{\|\boldsymbol{\Sigma}^{1/2}\boldsymbol{w}\|}\left( \mathbf{I}_d - \frac{\boldsymbol{\Sigma}\boldsymbol{w}\boldsymbol{w}^\top}{\|\boldsymbol{\Sigma}^{1/2}\boldsymbol{w}\|^2} \right)\boldsymbol{\Sigma}^{1/2}\,\mathbb{E}[\phi'(\langle \overline{\boldsymbol{w}}, \boldsymbol{z} \rangle)g(\langle \overline{\boldsymbol{u}}, \boldsymbol{z} \rangle)\boldsymbol{z}] \\
&= \frac{\boldsymbol{\Sigma}^{1/2}(\mathbf{I}_d - \overline{\boldsymbol{w}}\,\overline{\boldsymbol{w}}^\top)}{\|\boldsymbol{\Sigma}^{1/2}\boldsymbol{w}\|}\{\zeta_{\phi,g}(\langle \overline{\boldsymbol{w}}, \overline{\boldsymbol{u}} \rangle)\overline{\boldsymbol{u}} + \psi_{\phi,g}(\langle \overline{\boldsymbol{w}}, \overline{\boldsymbol{u}} \rangle)\overline{\boldsymbol{w}}\} \qquad \text{(by Lemma 11)} \\
&= \frac{\boldsymbol{\Sigma}^{1/2}(\mathbf{I}_d - \overline{\boldsymbol{w}}\,\overline{\boldsymbol{w}}^\top)\zeta_{\phi,g}(\langle \overline{\boldsymbol{w}}, \overline{\boldsymbol{u}} \rangle)\overline{\boldsymbol{u}}}{\|\boldsymbol{\Sigma}^{1/2}\boldsymbol{w}\|},
\end{aligned}$$

where

$$\psi_{\phi,g}(\langle \overline{\boldsymbol{w}}, \overline{\boldsymbol{u}} \rangle) := -\mathbb{E}[\phi'(\langle \overline{\boldsymbol{w}}, \boldsymbol{z} \rangle)g(\langle \overline{\boldsymbol{u}}, \boldsymbol{z} \rangle)\langle \overline{\boldsymbol{w}}, \boldsymbol{z} \rangle - \phi'(\langle \overline{\boldsymbol{w}}, \boldsymbol{z} \rangle)g'(\langle \overline{\boldsymbol{u}}, \boldsymbol{z} \rangle)\langle \overline{\boldsymbol{w}}, \overline{\boldsymbol{u}} \rangle] \tag{D.3}$$

(the above is only a function of $\langle \overline{\boldsymbol{w}}, \overline{\boldsymbol{u}} \rangle$ due to the Hermite expansion). $\qquad\square$

Given the closed form of the population gradient, the proof of Lemma 3 is immediate by noticing that

$$\frac{\mathrm{d}\overline{\boldsymbol{w}}^t}{\mathrm{d}t} = \frac{(\mathbf{I}_d - \overline{\boldsymbol{w}}^t\overline{\boldsymbol{w}}^{t\top})}{\|\boldsymbol{\Sigma}^{1/2}\boldsymbol{w}\|}\boldsymbol{\Sigma}^{1/2}\frac{\mathrm{d}\boldsymbol{w}^t}{\mathrm{d}t}. \tag{D.4}$$

Next, we move on to prove Proposition 4.

### D.1 Proof of Proposition 4

From Lemma 3, we have

$$\frac{\mathrm{d}\langle \overline{\boldsymbol{w}}^t, \overline{\boldsymbol{u}}\rangle}{\mathrm{d}t} = -\zeta_{\phi,g}(\langle \overline{\boldsymbol{w}}^t, \overline{\boldsymbol{u}}\rangle)\|\boldsymbol{\Sigma}^{1/2}\overline{\boldsymbol{u}}_{\perp}^t\|^2$$

$$\geq c\lambda_{\min}(\boldsymbol{\Sigma})\langle \overline{\boldsymbol{w}}^t, \overline{\boldsymbol{u}}\rangle^{s-1}(1 - \langle \overline{\boldsymbol{w}}^t, \overline{\boldsymbol{u}}\rangle^2),$$

where $\overline{\boldsymbol{u}}_{\perp}^t := \overline{\boldsymbol{u}} - \langle \overline{\boldsymbol{w}}^t, \overline{\boldsymbol{u}}\rangle\overline{\boldsymbol{w}}^t$. The above inequality and the fact that $\langle \overline{\boldsymbol{w}}^0, \overline{\boldsymbol{u}}\rangle > 0$ imply that $\langle \overline{\boldsymbol{w}}^t, \overline{\boldsymbol{u}}\rangle$ is non-decreasing in time. Let

$$T_1 := \sup\{t > 0 : \langle \overline{\boldsymbol{w}}^t, \overline{\boldsymbol{u}}\rangle < 1/2\}.$$

Then, on $t \in [0, T_1]$, we have

$$\frac{\mathrm{d}\langle \overline{\boldsymbol{w}}^t, \overline{\boldsymbol{u}}\rangle}{\mathrm{d}t} \geq \frac{3c\lambda_{\min}(\boldsymbol{\Sigma})}{4}\langle \overline{\boldsymbol{w}}^t, \overline{\boldsymbol{u}}\rangle^{s-1},$$

and integration yields

$$T_1 \leq 0 \vee \frac{4}{3c\lambda_{\min}(\boldsymbol{\Sigma})} \begin{cases} 1/2 - \langle \overline{\boldsymbol{w}}^0, \overline{\boldsymbol{u}}\rangle & s = 1 \\ \ln(1/(2\langle \overline{\boldsymbol{w}}^0, \overline{\boldsymbol{u}}\rangle)) & s = 2 \\ \frac{1}{s-2}\left((1/\langle \overline{\boldsymbol{w}}^0, \overline{\boldsymbol{u}}\rangle)^{s-2} - 2^{s-2}\right) & s > 2 \end{cases}.$$

Therefore, $T_1 \lesssim \tau_s(\langle \overline{\boldsymbol{w}}^0, \overline{\boldsymbol{u}}\rangle)/\lambda_{\min}(\boldsymbol{\Sigma})$.

For $t > T_1$, we have

$$\frac{\mathrm{d}\langle \overline{\boldsymbol{w}}^t, \overline{\boldsymbol{u}}\rangle}{\mathrm{d}t} \geq \frac{c\lambda_{\min}(\boldsymbol{\Sigma})}{2^{s-1}}(1 - \langle \overline{\boldsymbol{w}}^t, \overline{\boldsymbol{u}}\rangle^2).$$

Let

$$T_2 = \sup\{t > 0 : \langle \overline{\boldsymbol{w}}^t, \overline{\boldsymbol{u}}\rangle < 1 - \varepsilon\}.$$

Once again, integration implies

$$T_2 \leq T_1 + \frac{2^{s-2}}{c\lambda_{\min}(\boldsymbol{\Sigma})}\ln(2/(3\varepsilon)),$$

which completes the proof. $\qquad\square$

### D.2 Preliminary Lemmas for proving Theorem 5

We first introduce a number of concentration (and anti-concentration) lemmas that will be useful for proving Theorem 5.

**Lemma 16.** *Suppose $\{\boldsymbol{z}^{(i)}\}_{i=1}^n \overset{i.i.d.}{\sim} \mathcal{N}(0, \mathbf{I}_d)$, and $g : \mathbb{R} \to \mathbb{R}$ satisfies $|g(\cdot)| \leq C(1 + |\cdot|^p)$ for some $C > 0$ and $p \geq 1$. Additionally, suppose $\{\epsilon^{(i)}\}_{i=1}$ are i.i.d. $\sigma$-sub-Gaussian zero-mean noise independent of $\{\boldsymbol{z}^{(i)}\}_{i=1}^n$. Let $y^{(i)} := g(\langle \overline{\boldsymbol{u}}, \boldsymbol{z}^{(i)}\rangle) + \epsilon^{(i)}$ for some $\overline{\boldsymbol{u}} \in \mathbb{S}^{d-1}$. Then, for any $q > 0$, with probability at least $1 - 4d^{-q}$, we have*

$$\left|y^{(i)}\right| \leq C + C(2\ln(nd^q))^{p/2} + \sigma\sqrt{2\ln(nd^q)} \lesssim \ln(nd^q)^{p/2},$$

*for all $i$.*

**Proof.** Notice that $\langle \overline{\boldsymbol{u}}, \boldsymbol{z}^{(i)}\rangle \sim \mathcal{N}(0, 1)$, thus

$$\mathbb{P}\left(\left|\langle \overline{\boldsymbol{u}}, \boldsymbol{z}^{(i)}\rangle\right| \geq \sqrt{2\ln(nd^q)}\right) \leq 2n^{-1}d^{-q}.$$

Similarly, by the sub-Gaussian and zero-mean property of $\epsilon^{(i)}$,

$$\mathbb{P}\left(\left|\epsilon^{(i)}\right| \geq \sigma\sqrt{2\ln(nd^q)}\right) \leq 2n^{-1}d^{-q}.$$

Thus, by a union bound, we have

$$\left|\langle \overline{\boldsymbol{u}}, \boldsymbol{z}^{(i)}\rangle\right| \leq \sqrt{2\ln(nd^q)} \quad \text{and} \quad \left|\epsilon^{(i)}\right| \leq \sigma\sqrt{2\ln(nd^q)}, \quad \text{for all } 1 \leq i \leq n,$$

with probability at least $1 - 4d^{-q}$. Using the upper bound on $|g|$ finishes the proof. $\qquad\square$

**Lemma 17.** *Suppose* $\{\overline{z}^{(i)}\}_{i=1}^n$ *are i.i.d. samples drawn uniformly from* $\mathbb{S}^{d-1}$. *Then,*

$$\mathbb{P}\left(\sup_{\overline{w}\in\mathbb{S}^{d-1}}\sum_{i=1}^n \mathbf{1}\left(\left|\left\langle\overline{w},\overline{z}^{(i)}\right\rangle\right| \leq \frac{3\sqrt{d}}{8n}\right) \geq 3d\left(2 + \ln(8n/\sqrt{d})\right)\right) \leq e^{-d}.$$

**Proof.** Fix some $\epsilon \in (0,1)$. Let $N_\epsilon$ be a minimal $\epsilon$-covering of $\mathbb{S}^{d-1}$. Let $\hat{w}$ be the projection of $\overline{w}$ onto $N_\epsilon$. Notice that by the triangle inequality and the union bound

$$\mathbb{P}\left(\sup_{\overline{w}\in\mathbb{S}^{d-1}}\sum_{i=1}^n \mathbf{1}\left(\left|\left\langle\overline{w},\overline{z}^{(i)}\right\rangle\right| \leq \epsilon\right) \geq \alpha\right) \leq \mathbb{P}\left(\sup_{\hat{w}\in N_\epsilon}\sum_{i=1}^n \mathbf{1}\left(\left|\left\langle\hat{w},\overline{z}^{(i)}\right\rangle\right| \leq 2\epsilon\right) \geq \alpha\right)$$

$$\leq |N_\epsilon|\mathbb{P}\left(\sum_{i=1}^n \mathbf{1}\left(\left|\left\langle\hat{w},\overline{z}^{(i)}\right\rangle\right| \leq 2\epsilon\right) \geq \alpha\right)$$

$$\leq (3/\epsilon)^d \mathbb{P}\left(\sum_{i=1}^n \mathbf{1}\left(\left|\left\langle\hat{w},\overline{z}^{(i)}\right\rangle\right| \leq 2\epsilon\right) \geq \alpha\right).$$

Moreover, due to [BBSS22, Lemma A.7],

$$\mathbb{E}[\mathbf{1}(|\langle\hat{w},\overline{z}\rangle| \leq 2\epsilon)] = \mathbb{P}(\langle\hat{w},\overline{z}\rangle \leq 2\epsilon) \leq 8\sqrt{d}\epsilon.$$

Choose $\epsilon = \frac{3\sqrt{d}}{8n}$. By Lemma 25

$$\mathbb{P}\left(\sum_{i=1}^n \mathbf{1}\left(\left|\left\langle\hat{w},\overline{z}^{(i)}\right\rangle\right| \leq 2\epsilon\right) \geq 3d\left(2 + \ln(8n/\sqrt{d})\right)\right) \leq (3/\epsilon)^d e^{-d},$$

which completes the proof. $\qquad\square$

We summarize the above statements into a "good event", as characterized by the following lemma.

**Lemma 18.** *Let* $\{z^{(i)}\}_{i=1}^n \overset{i.i.d.}{\sim} \mathcal{N}(0, \mathbf{I}_d)$, *and* $\overline{z}^{(i)} := z^{(i)}/\|z^{(i)}\|$ *for every* $i$. *We say event* $\mathcal{G}$ *occurs whenever:*

1. $|y|^{(i)} \lesssim \ln(nd^q)^{p/2}$ *for all* $1 \leq i \leq n$.

2. $\sup_{\overline{w}\in\mathbb{S}^{d-1}}\sum_{i=1}^n \mathbf{1}\left(\left\langle\overline{w},\overline{z}^{(i)}\right\rangle \lesssim \sqrt{d}/n\right) \lesssim d\ln(n/\sqrt{d})$.

3. $\lambda_{\max}\left(\frac{1}{n}\sum_{i=1}^n z^{(i)}z^{(i)\top}\right) - 1 \lesssim \sqrt{d/n}$.

4. $1 - \lambda_{\min}\left(\frac{1}{n}\sum_{i=1}^n z^{(i)}z^{(i)\top}\right) \lesssim \sqrt{d/n}$.

*For* $n \gtrsim d$, *event* $\mathcal{G}$ *occurs with probability at least* $1 - \mathcal{O}(d^{-q})$.

**Proof.** The first and second statements of the lemma follow from Lemmas 16 and 17 respectively. The third and fourth statements are standard Gaussian covariance concentration bounds (see e.g. [Wai19, Theorem 6.1] where both statements hold with probability at least $1 - 2e^{-d}$). $\qquad\square$

### D.3  Proof of Theorem 5

We begin by recalling the definition $\overline{w} := \frac{\hat{\Sigma}^{1/2}w}{\|\hat{\Sigma}^{1/2}w\|}$ and the finite-samples dynamics (3.5), which we copy here for the reader's convenience,

$$\frac{\mathrm{d}w^t}{\mathrm{d}t} = \eta(w^t)\nabla_{w^t}\left\{\frac{1}{n}\sum_{i=1}^n \phi\left(\frac{\langle w^t, x^{(i)}\rangle}{\|\hat{\Sigma}^{1/2}w^t\|}\right)y^{(i)}\right\},$$

where $\eta(w^t) = \|\hat{\Sigma}^{1/2}w^t\|^2$. Moreover, via chain rule, we obtain

$$\frac{\mathrm{d}\overline{w}^t}{\mathrm{d}t} = \frac{(\mathbf{I}_d - \overline{w}^t\overline{w}^{t\top})\hat{\Sigma}^{1/2}}{\|\hat{\Sigma}^{1/2}w^t\|}\frac{\mathrm{d}w^t}{\mathrm{d}t}. \tag{D.5}$$

Let $\tilde{z}^{(i)} := \hat{\boldsymbol{\Sigma}}^{-1/2} x^{(i)} = \hat{\boldsymbol{\Sigma}}^{-1/2} \boldsymbol{\Sigma}^{1/2} z^{(i)}$. Then,

$$\frac{\mathrm{d}\overline{\boldsymbol{w}}^t}{\mathrm{d}t} = (\mathbf{I}_d - \overline{\boldsymbol{w}}^t\overline{\boldsymbol{w}}^{t\top})\hat{\boldsymbol{\Sigma}}(\mathbf{I}_d - \overline{\boldsymbol{w}}^t\overline{\boldsymbol{w}}^{t\top})\left\{\frac{1}{n}\sum_{i=1}^n \phi'\left(\frac{\langle \boldsymbol{w}^t, \boldsymbol{x}^{(i)}\rangle}{\|\hat{\boldsymbol{\Sigma}}^{1/2}\boldsymbol{w}^t\|}\right)y^{(i)}\tilde{z}^{(i)}\right\}$$

To simplify the notation, define

$$\boldsymbol{\nu}(\overline{\boldsymbol{w}}) := (\mathbf{I}_d - \overline{\boldsymbol{w}}\,\overline{\boldsymbol{w}}^\top)\hat{\boldsymbol{\Sigma}}(\mathbf{I}_d - \overline{\boldsymbol{w}}\,\overline{\boldsymbol{w}}^\top)\overline{\boldsymbol{u}}.$$

Then

$$\frac{\mathrm{d}\langle\overline{\boldsymbol{w}}^t, \overline{\boldsymbol{u}}\rangle}{\mathrm{d}t} = \left\langle \boldsymbol{\nu}(\overline{\boldsymbol{w}}^t), \frac{1}{n}\sum_{i=1}^n \phi'\left(\langle\overline{\boldsymbol{w}}^t, \tilde{z}^{(i)}\rangle\right)y^{(i)}\tilde{z}^{(i)}\right\rangle.$$

We can decompose the above dynamics into a population term and three different error terms in the following manner:

$$\begin{aligned}
\frac{\mathrm{d}\langle\overline{\boldsymbol{w}}^t, \overline{\boldsymbol{u}}\rangle}{\mathrm{d}t} =&\langle\boldsymbol{\nu}(\overline{\boldsymbol{w}}^t), \mathbb{E}_{\boldsymbol{z},y}[\phi'(\langle\overline{\boldsymbol{w}}^t, \boldsymbol{z}\rangle)y\boldsymbol{z}]\rangle \\
&+ \underbrace{\left\langle \boldsymbol{\nu}(\overline{\boldsymbol{w}}^t), \frac{1}{n}\sum_{i=1}^n \phi'\left(\langle\overline{\boldsymbol{w}}^t, \boldsymbol{z}^{(i)}\rangle\right)y^{(i)}\boldsymbol{z}^{(i)} - \mathbb{E}_{\boldsymbol{z},y}[\phi'(\langle\overline{\boldsymbol{w}}^t, \boldsymbol{z}\rangle)y\boldsymbol{z}]\right\rangle}_{=:\mathcal{E}_1} \\
&+ \underbrace{\frac{1}{n}\sum_{i=1}^n\left\{\phi'\left(\langle\overline{\boldsymbol{w}}^t, \tilde{z}^{(i)}\rangle\right) - \phi'\left(\langle\overline{\boldsymbol{w}}^t, \boldsymbol{z}^{(i)}\rangle\right)\right\}y^{(i)}\langle\boldsymbol{z}^{(i)}, \boldsymbol{\nu}(\overline{\boldsymbol{w}}^t)\rangle}_{=:\mathcal{E}_2} \\
&+ \underbrace{\frac{1}{n}\sum_{i=1}^n \phi'\left(\langle\overline{\boldsymbol{w}}, \tilde{z}^{(i)}\rangle\right)y^{(i)}\langle\tilde{z}^{(i)} - \boldsymbol{z}^{(i)}, \boldsymbol{\nu}(\overline{\boldsymbol{w}}^t)\rangle}_{=:\mathcal{E}_3}.
\end{aligned}$$

We will proceed in three steps. In the first, we bound $\mathcal{E}_1$, the concentration error. In the second, we bound $\mathcal{E}_2$ and $\mathcal{E}_3$, the errors due to estimating $\boldsymbol{\Sigma}$ with $\hat{\boldsymbol{\Sigma}}$ (i.e. replacing $\boldsymbol{z}^{(i)}$ with $\tilde{z}^{(i)}$). Finally, we will analyze the convergence time similar to that of Proposition 4. Throughout the proof, we will assume that the event $\mathcal{G}$ of Lemma 18 occurs.

**Step 1. Controlling the concentration error $\mathcal{E}_1$.** Let $K \asymp \ln(nd^q)^{p/2}$, and notice that on event $\mathcal{G}$ we have $|y^{(i)}| \lesssim K$ for all $i$. Let $y_K := y\mathbf{1}(|y| \leq K)$. On the event $\mathcal{G}$, we have $y_K^{(i)} = y^{(i)}$ for all $i$, and

$$\mathcal{E}_1 = \langle\boldsymbol{\nu}(\overline{\boldsymbol{w}}^t), \boldsymbol{\Delta}_n\rangle \geq -\|\boldsymbol{\Delta}_n\|\|\boldsymbol{\nu}(\overline{\boldsymbol{w}}^t)\|,$$

where

$$\boldsymbol{\Delta}_n := \frac{1}{n}\sum_{i=1}^n \phi'\left(\langle\overline{\boldsymbol{w}}, \boldsymbol{z}^{(i)}\rangle\right)y_K^{(i)}\boldsymbol{z}^{(i)} - \mathbb{E}_{\boldsymbol{z},y}[\phi'(\langle\overline{\boldsymbol{w}}, \boldsymbol{z}\rangle)y\boldsymbol{z}].$$

Thus, our objective is to bound $\|\boldsymbol{\Delta}_n\|$ uniformly for all $\overline{\boldsymbol{w}} \in \mathbb{S}^{d-1}$. To that end, we first modify the expectation in the above definition so that the empirical average and expected value match in terms of their random variables. Specifically,

$$\begin{aligned}
\sup_{\overline{\boldsymbol{w}}, \boldsymbol{v}\in\mathbb{S}^{d-1}}\langle\boldsymbol{\Delta}_n, \boldsymbol{v}\rangle = \sup_{\overline{\boldsymbol{w}}, \boldsymbol{v}\in\mathbb{S}^{d-1}}\frac{1}{n}\sum_{i=1}^n &\phi'\left(\langle\overline{\boldsymbol{w}}, \boldsymbol{z}^{(i)}\rangle\right)y_K^{(i)}\langle\boldsymbol{z}^{(i)}, \boldsymbol{v}\rangle - \mathbb{E}_{\boldsymbol{z},y}[\phi'(\langle\overline{\boldsymbol{w}}, \boldsymbol{z}\rangle)y_K\langle\boldsymbol{z}, \boldsymbol{v}\rangle] \\
&- \mathbb{E}_{\boldsymbol{z},y}[\phi'(\langle\overline{\boldsymbol{w}}, \boldsymbol{z}\rangle)y\langle\boldsymbol{z}, \boldsymbol{v}\rangle\mathbf{1}(|y| > K)].
\end{aligned}$$

By the Cauchy-Schwartz inequality,

$$\begin{aligned}
|\mathbb{E}_{\boldsymbol{z},y}[\phi'(\langle\overline{\boldsymbol{w}}, \boldsymbol{z}\rangle)y\langle\boldsymbol{z}, \boldsymbol{v}\rangle\mathbf{1}(|y| > K)]| &\leq \mathbb{E}_{\boldsymbol{z},y}\left[\phi'(\langle\overline{\boldsymbol{w}}, \boldsymbol{z}\rangle)^2 y^2\langle\boldsymbol{z}, \boldsymbol{v}\rangle^2\right]^{1/2}\mathbb{E}[\mathbf{1}(|y| > K)]^{1/2} \\
&\lesssim \mathbb{E}[y^4]^{1/4}\mathbb{E}_{\boldsymbol{z}}[\langle\boldsymbol{z}, \boldsymbol{v}\rangle^4]^{1/4}\mathbb{P}(|y| > K)^{1/2} \\
&\lesssim d^{-q/2},
\end{aligned}$$

where the last inequality follows from Lemma 16. Hence,

$$\sup_{\overline{\boldsymbol{w}}, \boldsymbol{v} \in \mathbb{S}^{d-1}} \langle \boldsymbol{\Delta}_n, \boldsymbol{v} \rangle \le \sup_{\overline{\boldsymbol{w}}, \boldsymbol{v} \in \mathbb{S}^{d-1}} \frac{1}{n} \sum_{i=1}^n \phi'\Big(\Big\langle \overline{\boldsymbol{w}}, \boldsymbol{z}^{(i)} \Big\rangle\Big) y_K^{(i)} \Big\langle \boldsymbol{z}^{(i)}, \boldsymbol{v} \Big\rangle - \mathbb{E}[\phi'(\langle \overline{\boldsymbol{w}}, \boldsymbol{z} \rangle) y_K \langle \boldsymbol{z}, \boldsymbol{v} \rangle]$$
$$+ \mathcal{O}(d^{-q/2}).$$

Next, we need to establish high-probability bounds via a covering argument. To simplify the exposition, define the stochastic process indexed by $\overline{\boldsymbol{w}} \in \mathbb{S}^{d-1}$ and $\boldsymbol{v} \in \mathbb{S}^{d-1}$ via

$$X_{\overline{\boldsymbol{w}}, \boldsymbol{v}}^{(i)} := \phi'\Big(\Big\langle \overline{\boldsymbol{w}}, \boldsymbol{z}^{(i)} \Big\rangle\Big) y_K^{(i)} \Big\langle \boldsymbol{z}^{(i)}, \boldsymbol{v} \Big\rangle.$$

Fix some $\epsilon_w, \epsilon_v > 0$. Let $\Theta_w$ and $\Theta_v$ be $\epsilon_w$ and $\epsilon_v$ coverings of $\mathbb{S}^{d-1}$, and let $\hat{\boldsymbol{w}}$ and $\hat{\boldsymbol{v}}$ denote the projection of $\overline{\boldsymbol{w}}$ onto $\Theta_w$ and of $\boldsymbol{v}$ onto $\Theta_v$ respectively, then

$$\sup_{\overline{\boldsymbol{w}}, \boldsymbol{v} \in \mathbb{S}^{d-1}} \frac{1}{n} \sum_{i=1}^n X_{\overline{\boldsymbol{w}}, \boldsymbol{v}}^{(i)} - \mathbb{E}[X_{\overline{\boldsymbol{w}}, \boldsymbol{v}}] = \sup_{\overline{\boldsymbol{w}}, \boldsymbol{v} \in \mathbb{S}^{d-1}} \frac{1}{n} \sum_{i=1}^n \Big( X_{\overline{\boldsymbol{w}}, \boldsymbol{v}}^{(i)} - X_{\overline{\boldsymbol{w}}, \hat{\boldsymbol{v}}}^{(i)} \Big) + \frac{1}{n} \sum_{i=1}^n \Big( X_{\overline{\boldsymbol{w}}, \hat{\boldsymbol{v}}}^{(i)} - X_{\hat{\boldsymbol{w}}, \hat{\boldsymbol{v}}}^{(i)} \Big)$$
$$+ \mathbb{E}_{\boldsymbol{z}, y}[X_{\overline{\boldsymbol{w}}, \hat{\boldsymbol{v}}} - X_{\overline{\boldsymbol{w}}, \boldsymbol{v}}] + \mathbb{E}_{\boldsymbol{z}, y}[X_{\hat{\boldsymbol{w}}, \hat{\boldsymbol{v}}} - X_{\overline{\boldsymbol{w}}, \hat{\boldsymbol{v}}}]$$
$$+ \frac{1}{n} \sum_{i=1}^n X_{\hat{\boldsymbol{w}}, \hat{\boldsymbol{v}}}^{(i)} - \mathbb{E}_{\boldsymbol{z}, y}[X_{\hat{\boldsymbol{w}}, \hat{\boldsymbol{v}}}].$$

We bound each of the terms using Cauchy-Schwartz. Specifically,

$$\frac{1}{n} \sum_{i=1}^n \Big( X_{\overline{\boldsymbol{w}}, \boldsymbol{v}}^{(i)} - X_{\overline{\boldsymbol{w}}, \hat{\boldsymbol{v}}}^{(i)} \Big) \le \sqrt{\frac{1}{n} \sum_{i=1}^n \phi'(\langle \overline{\boldsymbol{w}}, \boldsymbol{z}^{(i)} \rangle)^2 y_K^{(i)2}} \sqrt{\frac{1}{n} \sum_{i=1}^n \langle \boldsymbol{z}^{(i)}, \boldsymbol{v} - \hat{\boldsymbol{v}} \rangle^2} \lesssim K \epsilon_v,$$

where we used the upper bound on the operator norm of $\frac{1}{n} \sum_{i=1}^n \boldsymbol{z}^{(i)} \boldsymbol{z}^{(i)\top}$ from Lemma 18 together with the fact that $n \gtrsim d$. Similarly,

$$\mathbb{E}_{\boldsymbol{z}, y}[X_{\overline{\boldsymbol{w}}, \hat{\boldsymbol{v}}} - X_{\overline{\boldsymbol{w}}, \boldsymbol{v}}] \le \mathbb{E}_{\boldsymbol{z}, y}\big[\phi'(\langle \overline{\boldsymbol{w}}, \boldsymbol{z} \rangle)^2 y_K^2 \big]^{1/2} \mathbb{E}_{\boldsymbol{z}}\Big[\langle \boldsymbol{z}, \boldsymbol{v} - \hat{\boldsymbol{v}} \rangle^2\Big]^{1/2} \lesssim K \epsilon_v.$$

To bound the differences when we replace $\overline{\boldsymbol{w}}$ with $\hat{\boldsymbol{w}}$, we need to make a distinction between ReLU and smooth activations as the respective arguments are to some extent different. When $\phi'$ is Lipschitz,

$$\frac{1}{n} \sum_{i=1}^n \Big( X_{\overline{\boldsymbol{w}}, \hat{\boldsymbol{v}}}^{(i)} - X_{\hat{\boldsymbol{w}}, \hat{\boldsymbol{v}}}^{(i)} \Big) \le \sqrt{\frac{1}{n} \sum_{i=1}^n (y_K^{(i)})^2 \big(\phi'(\langle \overline{\boldsymbol{w}}, \boldsymbol{z}^{(i)} \rangle) - \phi'(\langle \hat{\boldsymbol{w}}, \boldsymbol{z}^{(i)} \rangle)\big)^2} \sqrt{\frac{1}{n} \sum_{i=1}^n \langle \boldsymbol{z}^{(i)}, \hat{\boldsymbol{v}}^2 \rangle} \lesssim K \epsilon_w,$$

and

$$\mathbb{E}[X_{\hat{\boldsymbol{w}}, \hat{\boldsymbol{v}}} - X_{\overline{\boldsymbol{w}}, \hat{\boldsymbol{v}}}] \le \mathbb{E}\Big[ y_K^2 (\phi'(\langle \overline{\boldsymbol{w}}, \boldsymbol{z} \rangle) - \phi'(\langle \hat{\boldsymbol{w}}, \boldsymbol{z} \rangle))^2 \Big]^{1/2} \mathbb{E}\Big[\langle \boldsymbol{z}, \hat{\boldsymbol{v}} \rangle^2 \Big]^{1/2} \lesssim K \epsilon_w.$$

Therefore, for a smooth activation $\phi$ we choose $\epsilon_v = \epsilon_w = \sqrt{d/n}$, and obtain

$$\sup_{\overline{\boldsymbol{w}}, \boldsymbol{v} \in \mathbb{S}^{d-1}} \frac{1}{n} \sum_{i=1}^n X_{\overline{\boldsymbol{w}}, \boldsymbol{v}}^{(i)} - \mathbb{E}_{\boldsymbol{z}, y}[X_{\overline{\boldsymbol{w}}, \boldsymbol{v}}] \le \sup_{\hat{\boldsymbol{w}}, \hat{\boldsymbol{v}}} \frac{1}{n} \sum_{i=1}^n X_{\hat{\boldsymbol{w}}, \hat{\boldsymbol{v}}}^{(i)} - \mathbb{E}_{\boldsymbol{z}, y}[X_{\hat{\boldsymbol{w}}, \hat{\boldsymbol{v}}}] + \tilde{\mathcal{O}}(\sqrt{d/n}).$$

When $\phi$ is the ReLU activation, we need to show that the sign of the preactivation changes only for a small number of samples when we change the weight $\overline{\boldsymbol{w}}$ to $\hat{\boldsymbol{w}}$. Notice that

$$\text{sign}\Big(\Big\langle \overline{\boldsymbol{w}}, \boldsymbol{z}^{(i)} \Big\rangle\Big) \ne \text{sign}\Big(\Big\langle \hat{\boldsymbol{w}}, \boldsymbol{z}^{(i)} \Big\rangle\Big) \implies \Big| \Big\langle \overline{\boldsymbol{w}}, \boldsymbol{z}^{(i)} \Big\rangle \Big| \le \Big| \Big\langle \hat{\boldsymbol{w}} - \overline{\boldsymbol{w}}, \boldsymbol{z}^{(i)} \Big\rangle \Big|$$
$$\implies \Big| \Big\langle \overline{\boldsymbol{w}}, \overline{\boldsymbol{z}}^{(i)} \Big\rangle \Big| \le \epsilon_w.$$

Recall that $\overline{\boldsymbol{z}}^{(i)} := \boldsymbol{z}^{(i)} / \|\boldsymbol{z}^{(i)}\|$. Choose $\epsilon_w \asymp \sqrt{d/n}$. On event $\mathcal{G}$, we know from Lemma 18 that at most $\mathcal{O}(d \ln(n/\sqrt{d}))$ samples can satisfy the above condition. Therefore,

$$\frac{1}{n} \sum_{i=1}^n \Big( X_{\overline{\boldsymbol{w}}, \hat{\boldsymbol{v}}}^{(i)} - X_{\hat{\boldsymbol{w}}, \hat{\boldsymbol{v}}}^{(i)} \Big) \le \sqrt{\frac{1}{n} \sum_{i=1}^n (y_K^{(i)})^2 \big(\phi'(\langle \overline{\boldsymbol{w}}, \boldsymbol{z}^{(i)} \rangle) - \phi'(\langle \hat{\boldsymbol{w}}, \boldsymbol{z}^{(i)} \rangle)\big)^2} \sqrt{\frac{1}{n} \sum_{i=1}^n \langle \boldsymbol{z}^{(i)}, \hat{\boldsymbol{v}} \rangle^2}$$
$$\lesssim K \sqrt{\frac{d \ln(n/\sqrt{d})}{n}}.$$

and

$$\mathbb{E}[X_{\hat{\boldsymbol{w}},\hat{\boldsymbol{v}}} - X_{\overline{\boldsymbol{w}},\hat{\boldsymbol{v}}}] \leq \mathbb{E}\Big[y_K^2(\phi'(\langle\overline{\boldsymbol{w}},\boldsymbol{z}\rangle) - \phi'(\langle\hat{\boldsymbol{w}},\boldsymbol{z}\rangle))^2\Big]^{1/2} \mathbb{E}\Big[\langle\boldsymbol{z},\hat{\boldsymbol{v}}\rangle^2\Big]^{1/2}$$

$$\leq K\mathbb{P}(\mathrm{sign}(\langle\overline{\boldsymbol{w}},\boldsymbol{z}\rangle) \neq \mathrm{sign}(\langle\hat{\boldsymbol{w}},\boldsymbol{z}\rangle))^{1/2}$$

$$\leq K\mathbb{P}(|\langle\overline{\boldsymbol{w}},\boldsymbol{z}\rangle| \leq \epsilon_w)^{1/2}$$

$$\leq 2K\sqrt{d^{1/2}\epsilon_w},$$

where the last inequality follows from the anti-concentration on the sphere [BBSS22, Lemma A.7]. Thus, for ReLU we choose $\epsilon_v \asymp \sqrt{d/n}$ and $\epsilon_w \asymp \sqrt{d}/n$, and once again obtain

$$\sup_{\overline{\boldsymbol{w}},\boldsymbol{v}\in\mathbb{S}^{d-1}} \frac{1}{n}\sum_{i=1}^n X_{\overline{\boldsymbol{w}},\boldsymbol{v}}^{(i)} - \mathbb{E}_{\boldsymbol{z},y}[X_{\overline{\boldsymbol{w}},\boldsymbol{v}}] \leq \sup_{\hat{\boldsymbol{w}},\hat{\boldsymbol{v}}} \frac{1}{n}\sum_{i=1}^n X_{\hat{\boldsymbol{w}},\hat{\boldsymbol{v}}}^{(i)} - \mathbb{E}_{\boldsymbol{z},y}[X_{\hat{\boldsymbol{w}},\hat{\boldsymbol{v}}}] + \tilde{\mathcal{O}}(\sqrt{d/n}).$$

It remains to bound the term

$$\sup_{\hat{\boldsymbol{w}},\hat{\boldsymbol{v}}} \frac{1}{n}\sum_{i=1}^n X_{\hat{\boldsymbol{w}},\hat{\boldsymbol{v}}}^{(i)} - \mathbb{E}[X_{\hat{\boldsymbol{w}},\hat{\boldsymbol{v}}}].$$

Notice that for fixed $\hat{\boldsymbol{w}},\hat{\boldsymbol{v}}$, $X_{\hat{\boldsymbol{w}},\hat{\boldsymbol{v}}}$ is sub-Gaussian with sub-Gaussian norm $\mathcal{O}(K)$. Thus, via the sub-Gaussian maximal inequality [VH16, Lemma 5.2],

$$\sup_{\hat{\boldsymbol{w}},\hat{\boldsymbol{v}}} \frac{1}{n}\sum_{i=1}^n X_{\hat{\boldsymbol{w}},\hat{\boldsymbol{v}}}^{(i)} - \mathbb{E}[X_{\hat{\boldsymbol{w}},\hat{\boldsymbol{v}}}] \lesssim \sqrt{K^2 d/n \ln(1/(\epsilon_w\epsilon_v))},$$

with probability at least $1 - e^{-d}$. Consequently, we have

$$\sup_{\overline{\boldsymbol{w}}\in\mathbb{S}^{d-1}} \|\boldsymbol{\Delta}_n\| \leq \tilde{\mathcal{O}}(\sqrt{d/n} + d^{-q/2}),$$

with probability at least $1 - \mathcal{O}(d^{-q})$. Assuming that $n$ grows at most polynomially in dimension and choosing a sufficiently large $q$, we have $\sup_{\overline{\boldsymbol{w}}\in\mathbb{S}^{d-1}} \|\boldsymbol{\Delta}_n\| \leq \tilde{\mathcal{O}}(\sqrt{d/n})$ with probability at least $1 - \mathcal{O}(d^{-q})$.

Finally, by Lemma 23,

$$\|\boldsymbol{\nu}(\overline{\boldsymbol{w}}^t)\| \leq \lambda_{\max}\Big(\hat{\boldsymbol{\Sigma}}\Big) \lesssim \lambda_{\max}(\boldsymbol{\Sigma}), \tag{D.6}$$

with probability at least $1 - e^{-n'/2}$. Combining the above with the bound on $\|\boldsymbol{\Delta}_n\|$, we have $\mathcal{E}_1 \geq -\lambda_{\max}(\boldsymbol{\Sigma})\tilde{\mathcal{O}}(\sqrt{d/n})$ with probability at least $1 - \mathcal{O}(d^{-q})$, which concludes the first step of the proof.

**Step 2. Bounding the error due to the estimation of $\boldsymbol{\Sigma}$, i.e. $\mathcal{E}_2$ and $\mathcal{E}_3$.** Recall that we are considering the event $\mathcal{G}$, thus $y^{(i)} = y_K^{(i)}$. We can control each of the error terms separately. We begin by $\mathcal{E}_2$, where by Cauchy-Schwartz

$$\mathcal{E}_2 = \frac{1}{n}\sum_{i=1}^n \Big\{\phi'\Big(\big\langle\overline{\boldsymbol{w}}^t,\tilde{\boldsymbol{z}}^{(i)}\big\rangle\Big) - \phi'\Big(\big\langle\overline{\boldsymbol{w}}^t,\boldsymbol{z}^{(i)}\big\rangle\Big)\Big\}y_K^{(i)}\big\langle\boldsymbol{z}^{(i)},\boldsymbol{\nu}(\overline{\boldsymbol{w}}^t)\big\rangle$$

$$\geq -\sqrt{\frac{1}{n}\sum_{i=1}^n\Big\{\phi'(\langle\overline{\boldsymbol{w}}^t,\boldsymbol{z}^{(i)}\rangle) - \phi'\Big(\big\langle\overline{\boldsymbol{w}}^t,\tilde{\boldsymbol{z}}^{(i)}\big\rangle\Big)\Big\}^2}\sqrt{\frac{1}{n}\sum_{i=1}^n y_K^{(i)^2}\big\langle\boldsymbol{z}^{(i)},\boldsymbol{\nu}(\overline{\boldsymbol{w}}^t)\big\rangle^2}$$

$$\geq -K\|\boldsymbol{\nu}(\overline{\boldsymbol{w}}^t)\|\sqrt{\frac{1}{n}\sum_{i=1}^n\Big\{\phi'(\langle\overline{\boldsymbol{w}}^t,\boldsymbol{z}^{(i)}\rangle) - \phi'\Big(\big\langle\overline{\boldsymbol{w}}^t,\tilde{\boldsymbol{z}}^{(i)}\big\rangle\Big)\Big\}^2},$$

where the last line follows from Lemma 18 and the fact that $n \gtrsim d$. When $\phi'$ is additionally Lipschitz, we have

$$\mathcal{E}_2 \gtrsim -K\|\boldsymbol{\nu}(\overline{\boldsymbol{w}}^t)\|\sqrt{\frac{1}{n}\sum_{i=1}^n\big\langle\overline{\boldsymbol{w}}^t,\boldsymbol{z}^{(i)} - \tilde{\boldsymbol{z}}^{(i)}\big\rangle^2}.$$

Moreover, for any $\overline{\boldsymbol{w}} \in \mathbb{S}^{d-1}$,

$$\frac{1}{n}\sum_{i=1}^{n}\left\langle\overline{\boldsymbol{w}}, \boldsymbol{z}^{(i)} - \tilde{\boldsymbol{z}}^{(i)}\right\rangle^2 \le \|\frac{1}{n}\sum_{i=1}^{n}\boldsymbol{z}^{(i)}\boldsymbol{z}^{(i)\top}\|^2\|(\mathbf{I}_d - \hat{\boldsymbol{\Sigma}}^{-1/2}\boldsymbol{\Sigma}^{1/2})\overline{\boldsymbol{w}}\|^2$$

$$\le \|\frac{1}{n}\sum_{i=1}^{n}\boldsymbol{z}^{(i)}\boldsymbol{z}^{(i)\top}\|^2\|\mathbf{I}_d - \hat{\boldsymbol{\Sigma}}^{-1/2}\boldsymbol{\Sigma}^{1/2}\|^2$$

$$\lesssim \frac{d}{n'},$$

where the last inequality holds with probability at least $1 - 2e^{-d}$ on the event of Lemma 24. Hence for smooth activations we conclude

$$\mathcal{E}_2 \gtrsim -K\|\boldsymbol{\nu}(\overline{\boldsymbol{w}}^t)\|\sqrt{d/n'}.$$

When $\phi$ is the ReLU activation, we need a more involved argument to control $\mathcal{E}_2$. In particular, we will show that for any $\overline{\boldsymbol{w}}$, at most only $\tilde{\mathcal{O}}(d)$ datapoints can have $\operatorname{sign}(\langle\overline{\boldsymbol{w}}, \boldsymbol{z}^{(i)}\rangle) \neq \operatorname{sign}(\langle\overline{\boldsymbol{w}}, \tilde{\boldsymbol{z}}^{(i)}\rangle)$. Notice that

$$\operatorname{sign}\left(\left\langle\overline{\boldsymbol{w}}, \boldsymbol{z}^{(i)}\right\rangle\right) \neq \operatorname{sign}\left(\left\langle\overline{\boldsymbol{w}}, \tilde{\boldsymbol{z}}^{(i)}\right\rangle\right) \implies \left|\left\langle\overline{\boldsymbol{w}}, \boldsymbol{z}^{(i)}\right\rangle\right| \le \left|\left\langle\overline{\boldsymbol{w}}, \boldsymbol{z}^{(i)} - \tilde{\boldsymbol{z}}^{(i)}\right\rangle\right|$$

$$\implies \left|\left\langle\overline{\boldsymbol{w}}, \overline{\boldsymbol{z}}^{(i)}\right\rangle\right| \le \|\mathbf{I}_d - \hat{\boldsymbol{\Sigma}}^{-1/2}\boldsymbol{\Sigma}^{1/2}\| \qquad \text{(D.7)}$$

where $\overline{\boldsymbol{z}}^{(i)} := \frac{\boldsymbol{z}^{(i)}}{\|\boldsymbol{z}^{(i)}\|}$ is distributed uniformly over $\mathbb{S}^{d-1}$. From Lemma 24 we have $\|\mathbf{I}_d - \hat{\boldsymbol{\Sigma}}^{-1/2}\boldsymbol{\Sigma}^{1/2}\| \lesssim \sqrt{\frac{d}{n'}}$ with probability at least $1 - 2e^{-d}$. On the other hand, from Lemma 17 we know with probability at least $1 - e^{-d}$, for any $\overline{\boldsymbol{w}} \in \mathbb{S}^{d-1}$ at most $\tilde{\mathcal{O}}(d)$ of the labeled samples have $|\langle\overline{\boldsymbol{w}}, \boldsymbol{z}^{(i)}\rangle| \lesssim \sqrt{d}/n$. Recall that $n' \gtrsim n^2$ when using the ReLU activation. This is precisely why we make this choice for the ReLU activation, as we need to balance the RHS of (D.7) which is of order $\sqrt{d/n'}$ with the LHS of (D.7) which should at most be of order $\sqrt{d}/n$ if we want to ensure only $\tilde{\mathcal{O}}(d)$ samples satisfy the bound. When $n' = n^2$ we can balance these two terms, thus with probability at least $1 - 3e^{-d}$ the sign change can occur for at most $\tilde{\mathcal{O}}(d)$ many samples, and

$$\frac{1}{n}\sum_{i=1}^{n}\left\{\phi'\left(\left\langle\overline{\boldsymbol{w}}^t, \boldsymbol{z}^{(i)}\right\rangle\right) - \phi'\left(\left\langle\overline{\boldsymbol{w}}^t, \tilde{\boldsymbol{z}}^{(i)}\right\rangle\right)\right\}^2 \le \tilde{\mathcal{O}}\left(\frac{d}{n}\right).$$

In this case, we end up with

$$\mathcal{E}_2 \ge -K\|\boldsymbol{\nu}(\overline{\boldsymbol{w}}^t)\|\tilde{\mathcal{O}}(\sqrt{d/n}).$$

Bounding $\mathcal{E}_3$ for ReLU and Lipschitz $\phi'$ is identical. In both cases, by Cauchy-Schwartz,

$$\mathcal{E}_3 \ge -\sqrt{\frac{1}{n}\sum_{i=1}^{n}\phi'(\langle\overline{\boldsymbol{w}}, \boldsymbol{z}^{(i)}\rangle)^2 y_K^{(i)2}}\sqrt{\frac{1}{n}\sum_{i=1}^{n}\left\langle\boldsymbol{z}^{(i)} - \tilde{\boldsymbol{z}}^{(i)}, \boldsymbol{\nu}(\overline{\boldsymbol{w}}^t)\right\rangle^2}$$

$$\gtrsim -K\|\frac{1}{n}\sum_{i=1}^{n}\boldsymbol{z}^{(i)}\boldsymbol{z}^{(i)\top}\|\|\mathbf{I}_d - \hat{\boldsymbol{\Sigma}}^{-1/2}\boldsymbol{\Sigma}^{1/2}\|\|\boldsymbol{\nu}(\overline{\boldsymbol{w}}^t)\|$$

$$\gtrsim -K\|\boldsymbol{\nu}(\overline{\boldsymbol{w}}^t)\|\sqrt{d/n'},$$

which holds on the intersection of event $\mathcal{G}$ and of Lemma 23. At last, using the bound on $\|\boldsymbol{\nu}(\overline{\boldsymbol{w}}^t)\|$ from (D.6), we obtain

$$\mathcal{E}_2 \wedge \mathcal{E}_3 \ge -\lambda_{\max}(\boldsymbol{\Sigma})\tilde{\mathcal{O}}(\sqrt{d/n}),$$

with probability at least $1 - \mathcal{O}(d^{-q})$.

**Step 3. Analyzing the Convergence.** As a result of the previous steps, we have established

$$\frac{\mathrm{d}\langle\overline{\boldsymbol{w}}^t, \overline{\boldsymbol{u}}\rangle}{\mathrm{d}t} \ge \left\langle\boldsymbol{\nu}(\overline{\boldsymbol{w}}^t), \mathbb{E}\left[\phi'(\langle\overline{\boldsymbol{w}}^t, \boldsymbol{z}\rangle)y\boldsymbol{z}\right]\right\rangle - \lambda_{\max}(\boldsymbol{\Sigma})\tilde{\mathcal{O}}(\sqrt{d/n}).$$

Thanks to Lemma 11, we can write

$$\mathbb{E}[\phi'(\langle \overline{w}, z \rangle) y z] = \mathbb{E}[\phi'(\langle \overline{w}, z \rangle) g(\langle \overline{u}, z \rangle) z]$$
$$= -\zeta_{\phi,g}(\langle \overline{w}, \overline{u} \rangle) \overline{u} - \psi_{\phi,g}(\langle \overline{w}^t, \overline{u} \rangle) \overline{w},$$

where $\zeta_{\phi,g}$ and $\psi_{\phi,g}$ were introduced in (D.2) and (D.3) respectively. Recall the definition of $\boldsymbol{\nu}(\overline{w}^t)$,

$$\boldsymbol{\nu}(\overline{w}^t) := (\mathbf{I}_d - \overline{w}^t \overline{w}^{t\top}) \hat{\boldsymbol{\Sigma}} (\mathbf{I}_d - \overline{w}^t \overline{w}^{t\top}) \overline{u}.$$

Therefore,

$$\frac{\mathrm{d} \langle \overline{w}^t, \overline{u} \rangle}{\mathrm{d}t} \geq -\zeta_{\phi,g}(\langle \overline{w}^t, \overline{u} \rangle) \overline{u}^\top (\mathbf{I}_d - \overline{w}^t \overline{w}^{t\top}) \hat{\boldsymbol{\Sigma}} (\mathbf{I}_d - \overline{w}^t \overline{w}^{tT}) \overline{u} - \lambda_{\max}(\boldsymbol{\Sigma}) \tilde{\mathcal{O}}(\sqrt{d/n})$$

$$\geq c \langle \overline{w}^t, \overline{u} \rangle^{s-1} \langle \overline{u}_\perp^t, \hat{\boldsymbol{\Sigma}} \overline{u}_\perp^t \rangle - \lambda_{\max}(\boldsymbol{\Sigma}) \tilde{\mathcal{O}}(\sqrt{d/n}) \qquad \text{(By Assumption 2)}$$

$$\geq c \lambda_{\min}(\hat{\boldsymbol{\Sigma}}) \langle \overline{w}^t, \overline{u} \rangle^{s-1} (1 - \langle \overline{w}^t, \overline{u} \rangle^2) - \lambda_{\max}(\boldsymbol{\Sigma}) \tilde{\mathcal{O}}(\sqrt{d/n}),$$

where $\overline{u}_\perp^t := \overline{u} - \langle \overline{u}, \overline{w}^t \rangle \overline{w}^t$. Moreover, from Lemma 23, we have

$$\lambda_{\min}(\hat{\boldsymbol{\Sigma}}) \geq \lambda_{\min}(\boldsymbol{\Sigma}) \left( \frac{1}{2} - \sqrt{\frac{\mathrm{tr}(\boldsymbol{\Sigma})}{n' \lambda_{\min}(\boldsymbol{\Sigma})}} \right)$$

$$\geq \lambda_{\min}(\boldsymbol{\Sigma}) \left( \frac{1}{2} - \sqrt{\frac{d \varkappa(\boldsymbol{\Sigma})}{n'}} \right),$$

with probability at least $1 - e^{-n'/8}$. Hence, for $n' \gtrsim d \varkappa(\boldsymbol{\Sigma})$ we have $\lambda_{\min}(\hat{\boldsymbol{\Sigma}}) \gtrsim \lambda_{\min}(\boldsymbol{\Sigma})$, and consequently,

$$\frac{\mathrm{d} \langle \overline{w}^t, \overline{u} \rangle}{\mathrm{d}t} \geq c' \lambda_{\min}(\boldsymbol{\Sigma}) \langle \overline{w}^t, \overline{u} \rangle^{s-1} (1 - \langle \overline{w}^t, \overline{u} \rangle^2) - \lambda_{\max}(\boldsymbol{\Sigma}) \tilde{\mathcal{O}}(\sqrt{d/n}),$$

where $c'$ is a universal constant. Notice that the first term in the RHS above denotes the signal, while the second term denotes the noise. We want to ensure the noise remains smaller than the signal throughout the trajectory, which leads to the convergence of $\overline{w}^t$ to $\overline{u}$. Notice that the signal term is first increasing, then decreasing for $\langle \overline{w}^t, \overline{u} \rangle \in [0, 1]$. Thus, it suffices to ensure the noise is smaller than the signal on the two ends of the interval, i.e. at time $t = 0$ and at time $t = T$ where $\langle \overline{w}^T, \overline{u} \rangle = 1 - \varepsilon$. At initialization, this condition leads to

$$n \gtrsim C d \varkappa(\boldsymbol{\Sigma})^2 \langle \overline{w}^0, \overline{u} \rangle^{2(1-s)},$$

and at time $t = T$, leads to

$$n \gtrsim \frac{C d \varkappa(\boldsymbol{\Sigma})^2}{\varepsilon^2},$$

where $C$ hides constant depending only on $s$ and at most polylogarithmic factors of $d$. Thus, we have established the sample complexity as presented by Theorem 5.

It remains to obtain the convergence time. With the above sample complexity, we have

$$\frac{\mathrm{d} \langle \overline{w}^t, \overline{u} \rangle}{\mathrm{d}t} \geq c'' \lambda_{\min}(\boldsymbol{\Sigma}) \langle \overline{w}^t, \overline{u} \rangle^{s-1} (1 - \langle \overline{w}^t, \overline{u} \rangle^2),$$

where $c''$ is a universal constant. The rest of the proof follows by integration and is identical to the proof of Proposition 4 in Appendix D.1. $\qquad \square$

## D.4 Proof of Corollary 6

The proof follows immediately from Theorem 5 and the following lemma which describes how $\langle \overline{w}^0, \overline{u} \rangle$ behaves under different regimes of $r_1$ and $r_2$.

**Lemma 19.** *Suppose $\boldsymbol{\Sigma}$ follows the $(\kappa, \boldsymbol{\theta})$-spiked model, $\boldsymbol{w}^0$ is sampled uniformly from $\mathbb{S}^{d-1}$, $n' \gtrsim d$, and there exist universal constants $C_2, C_2', C_3, C_3' > 0$ such that*

$$C_2 d^{r_2} \leq \kappa \leq C_2' d^{r_2} \qquad \text{and} \qquad C_3 d^{-r_1} \leq \langle \boldsymbol{u}, \boldsymbol{\theta} \rangle \leq C_3' d^{-r_1}$$

*for $r_1 \in [0, 1/2]$ and $r_2 \in [0, 1]$. Then, conditioned on $\langle \overline{\boldsymbol{w}}^0, \overline{\boldsymbol{u}} \rangle > 0$, with any arbitrarily large constant probability $1 - \delta$, for sufficiently large $d$ (that depends on $\delta$) we have*

$$\langle \overline{\boldsymbol{w}}^0, \overline{\boldsymbol{u}} \rangle \gtrsim \begin{cases} d^{-1/2} & 0 \leq r_2 < r_1 \\ d^{r_2 - r_1 - 1/2} & r_1 < r_2 < 2r_1 \\ d^{(r_2-1)/2} & 2r_1 < r_2 < 1 \end{cases}. \tag{D.8}$$

**Proof.** By definition,

$$\langle \overline{\boldsymbol{w}}^0, \overline{\boldsymbol{u}} \rangle = \frac{\left\langle \hat{\boldsymbol{\Sigma}}^{1/2} \boldsymbol{w}^0, \boldsymbol{\Sigma}^{1/2} \boldsymbol{u} \right\rangle}{\|\hat{\boldsymbol{\Sigma}}^{1/2} \boldsymbol{w}^0\| \|\boldsymbol{\Sigma}^{1/2} \boldsymbol{u}\|}.$$

Recall that we are conditioning our arguments on $\langle \overline{\boldsymbol{w}}^0, \overline{\boldsymbol{u}} \rangle > 0$, hence the numerator of the above fraction is positive. To translate the sample complexities of Theorems 5 and 7 to the spiked model, our goal is to lower bound $\langle \overline{\boldsymbol{w}}^0, \overline{\boldsymbol{u}} \rangle$ in terms of $d$, $r_1$, and $r_2$.

We begin by observing that

$$\|\hat{\boldsymbol{\Sigma}}^{1/2} \boldsymbol{w}\| \leq \|\hat{\boldsymbol{\Sigma}}^{1/2} \boldsymbol{\Sigma}^{-1/2}\| \|\boldsymbol{\Sigma}^{1/2} \boldsymbol{w}\| \lesssim \|\boldsymbol{\Sigma}^{1/2} \boldsymbol{w}\|,$$

where the last inequality holds on the event of Lemma 24, which happens with probability at least $1 - 2e^{-d}$. Consequently,

$$\langle \overline{\boldsymbol{w}}^0, \overline{\boldsymbol{u}} \rangle \gtrsim \frac{\left\langle \hat{\boldsymbol{\Sigma}}^{1/2} \boldsymbol{w}^0, \boldsymbol{\Sigma}^{1/2} \boldsymbol{u} \right\rangle}{\|\boldsymbol{\Sigma}^{1/2} \boldsymbol{w}^0\| \|\boldsymbol{\Sigma}^{1/2} \boldsymbol{u}\|} = \frac{\langle \boldsymbol{w}^0, \boldsymbol{\Sigma} \boldsymbol{u} \rangle + \left\langle \boldsymbol{w}^0, (\hat{\boldsymbol{\Sigma}}^{1/2} - \boldsymbol{\Sigma}^{1/2}) \boldsymbol{\Sigma}^{1/2} \boldsymbol{u} \right\rangle}{\|\boldsymbol{\Sigma}^{1/2} \boldsymbol{w}^0\| \|\boldsymbol{\Sigma}^{1/2} \boldsymbol{u}\|}.$$

Furthermore, due to the Markov inequality,

$$\mathbb{P}\left( \langle \boldsymbol{w}^0, \boldsymbol{\theta} \rangle^2 \geq \frac{C_1}{d} \right) \leq 1/C_1.$$

Similarly (by conditioning on $\hat{\boldsymbol{\Sigma}}$)

$$\mathbb{P}\left( \left\langle \boldsymbol{w}^0, (\hat{\boldsymbol{\Sigma}}^{1/2} - \boldsymbol{\Sigma}^{1/2}) \boldsymbol{\Sigma}^{1/2} \boldsymbol{u} \right\rangle^2 \geq \frac{C_1 \|(\hat{\boldsymbol{\Sigma}}^{1/2} - \boldsymbol{\Sigma}^{1/2}) \boldsymbol{\Sigma}^{1/2} \boldsymbol{u}\|^2}{d} \right) \leq 1/C_1.$$

Additionally, on the event of Lemma 24,

$$\|(\hat{\boldsymbol{\Sigma}}^{1/2} - \boldsymbol{\Sigma}^{1/2}) \boldsymbol{\Sigma}^{1/2} \boldsymbol{u}\| \leq \|\hat{\boldsymbol{\Sigma}}^{1/2} \boldsymbol{\Sigma}^{-1/2} - \mathbf{I}_d\| \|\boldsymbol{\Sigma} \boldsymbol{u}\| \lesssim \sqrt{d/n'} \|\boldsymbol{\Sigma} \boldsymbol{u}\|.$$

Therefore, on the above events, for some absolute constant $C > 0$

$$\langle \overline{\boldsymbol{w}}^0, \overline{\boldsymbol{u}} \rangle \gtrsim \frac{\langle \boldsymbol{w}^0, \boldsymbol{\Sigma} \boldsymbol{u} \rangle - C \|\boldsymbol{\Sigma} \boldsymbol{u}\| \sqrt{1/n'}}{\sqrt{\frac{1 + C_2' C_1}{1 + \kappa}} \|\boldsymbol{\Sigma} \boldsymbol{u}\|}$$

$$\gtrsim \frac{\langle \boldsymbol{w}^0, \boldsymbol{u} \rangle + \kappa \langle \boldsymbol{w}^0, \boldsymbol{\theta} \rangle \langle \boldsymbol{u}, \boldsymbol{\theta} \rangle - C(1 + \kappa |\langle \boldsymbol{u}, \boldsymbol{\theta} \rangle|) \sqrt{1/n'}}{\sqrt{1 + C_2' C_1} \sqrt{1 + \kappa \langle \boldsymbol{u}, \boldsymbol{\theta} \rangle^2}}.$$

Recall that $C_2 d^{r_2} \leq \kappa \leq C_2' d^{r_2}$ and $C_3 d^{-r_1} \leq \langle \boldsymbol{u}, \boldsymbol{\theta} \rangle \leq C_3' d^{-r_1}$ (notice that changing $\boldsymbol{\theta}$ to $-\boldsymbol{\theta}$ does not change the spiked model of Assumption 1, thus we can assume $\langle \boldsymbol{u}, \boldsymbol{\theta} \rangle \geq 0$ without loss of generality). Then,

$$\langle \overline{\boldsymbol{w}}^0, \overline{\boldsymbol{u}} \rangle \gtrsim \frac{\langle \boldsymbol{w}^0, \boldsymbol{u} \rangle + \kappa \langle \boldsymbol{w}^0, \boldsymbol{\theta} \rangle \langle \boldsymbol{u}, \boldsymbol{\theta} \rangle - C(1 + \kappa \langle \boldsymbol{u}, \boldsymbol{\theta} \rangle) \sqrt{1/n'}}{\sqrt{1 + C_2' C_1} \sqrt{1 + C_2' C_3'^2 d^{r_2 - 2r_1}}}. \tag{D.9}$$

The last term in the numerator can be made arbitrarily small by sufficiently large $n'$, hence we focus on other terms for now. Intuitively, when $r_2 < 2r_1$, the denominator is of constant order. If additionally $r_2 < r_1$, the dominant term in the numerator is $\langle w^0, u \rangle$ and $\langle \overline{w}^0, \overline{u} \rangle \asymp 1/\sqrt{d}$, otherwise the dominant term is $\kappa \langle w^0, \theta \rangle \langle u, \theta \rangle$ and $\langle \overline{w}^0, \overline{u} \rangle \asymp d^{r_2 - r_1 - 1/2}$. On the other hand, when $r_2 > 2r_1$, the denominator is of order $d^{r_2/2 - r_1}$, and once again the dominant term of the numerator is $\kappa \langle w^0, \theta \rangle \langle u, \theta \rangle$, therefore $\langle \overline{w}^0, \overline{u} \rangle \asymp d^{(r_2-1)/2}$. Using this intuition, we analyze each of the following regimes separately.

Case 1. $0 < r_2 < r_1$: In this case, by [BBSS22, Lemma A.7] we have $|\langle w^0, u \rangle| \geq c/\sqrt{d}$ with probability at least $1 - 4c$. On the intersection of all considered events with $\langle \overline{w}^0, \overline{u} \rangle > 0$, and for sufficiently large $d$ and $n' \gtrsim d$, we must have $\langle w^0, u \rangle > 0$ (otherwise $\langle \overline{w}^0, \overline{u} \rangle < 0$). Thus by plugging the values in (D.9),

$$\langle \overline{w}^0, \overline{u} \rangle \gtrsim \frac{c - \sqrt{C_1} C_2' C_3' d^{r_2 - r_1} - C(1 + C_2' C_3' d^{r_2 - r_1}) \sqrt{d/n'}}{\sqrt{d} \sqrt{1 + C_2' C_1} \sqrt{1 + C_2' C_3'^2 d^{r_2 - 2r_1}}} \gtrsim \frac{1}{\sqrt{d}}. \tag{D.10}$$

The intersection of all desired events and $\langle \overline{w}^0, \overline{u} \rangle > 0$ happens with probability at least $\frac{1}{2} - 4c - 2/C_1 - 2e^{-d}$, thus conditioned on $\langle \overline{w}^0, \overline{u} \rangle$ the probability is at least $1 - 8c - 4/C_1 - 4e^{-d}$. Choosing sufficiently small $c$, large $C_1$, and respectively large $d$ and $n' \gtrsim d$ with sufficiently large absolute constant, we can arbitrarily increase the (constant) probability of success. Thus the analysis of this regime is complete.

Case 2. $r_1 < r_2 < 2r_1$: This time we use the fact that $|\langle w^0, u \rangle| \leq \sqrt{C_1/d}$ with probability at least $1 - 1/C_1$, and $|\langle w^0, \theta \rangle| \geq c/\sqrt{d}$ with probability at least $1 - 4c$. By an argument similar to the previous case, for sufficiently large $d$ and $n' \gtrsim d$, $\langle \overline{w}^0, \overline{u} \rangle > 0$ implies $\langle w^0, \theta \rangle > 0$, hence by (D.9)

$$\langle \overline{w}^0, \overline{u} \rangle \gtrsim \frac{-\sqrt{C_1} + c C_2 C_3 d^{r_2 - r_1} - C(1 + C_2' C_3' d^{r_2 - r_1}) \sqrt{d/n'}}{\sqrt{d} \sqrt{1 + C_2' C_1} \sqrt{1 + C_2' C_3'^2 d^{r_2 - 2r_1}}} \gtrsim d^{r_2 - r_1 - 1/2}, \tag{D.11}$$

with probability at least $1 - 8c - 4/C_1 - 4e^{-d}$ when conditioned on $\langle \overline{w}^0, \overline{u} \rangle > 0$.

Case 3. $2r_1 < r_2 < 1$: Once again recall (D.9). To bound the numerator, we repeat the exact same argument as in the previous case, thus

$$\langle \overline{w}^0, \overline{u} \rangle \geq \frac{-\sqrt{C_1} + c C_2 C_3 d^{r_2 - r_1} - C(1 + C_2' C_3' d^{r_2 - 1}) \sqrt{d/n'}}{\sqrt{(1 + C_2' C_1) C_2' C_3'^2 d^{\frac{1 + r_2 - 2r_1}{2}}}} \gtrsim d^{(r_2 - 1)/2}, \tag{D.12}$$

which finishes the proof of the lemma. $\qquad\square$

# E   Proofs of Section 4

## E.1   Proof of Theorem 7

We recall from (D.5) that

$$\frac{d\overline{w}^t}{dt} = \frac{(\mathbf{I}_d - \overline{w}^t \overline{w}^{t\top}) \hat{\Sigma}^{1/2}}{\|\Sigma^{1/2} w\|} \frac{dw^t}{dt}.$$

Furthermore, the preconditioned dynamics of $w^t$ given by (4.1) reads

$$\frac{dw^t}{dt} = \frac{\eta(w^t) \hat{\Sigma}^{-1/2} (\mathbf{I}_d - \overline{w}^t \overline{w}^{t\top})}{\|\hat{\Sigma}^{1/2} w\|} \left\{ \frac{1}{n} \sum_{i=1}^n \phi'\left( \langle \overline{w}^t, \tilde{z}^{(i)} \rangle \right) y^{(i)} \tilde{z}^{(i)} \right\},$$

where we recall $\tilde{z}^{(i)} := \hat{\Sigma}^{-1/2} x^{(i)}$. Plugging in $\eta(w^t) = \|\hat{\Sigma}^{1/2} w\|^2$ yields

$$\frac{d\overline{w}^t}{dt} = (\mathbf{I}_d - \overline{w}^t \overline{w}^{t\top})^2 \left\{ \frac{1}{n} \sum_{i=1}^n \phi'\left( \langle \overline{w}^t, \tilde{z}^{(i)} \rangle \right) y^{(i)} \tilde{z}^{(i)} \right\}$$

$$= (\mathbf{I}_d - \overline{w}^t \overline{w}^{t\top}) \left\{ \frac{1}{n} \sum_{i=1}^n \phi'\left( \langle \overline{w}^t, \tilde{z}^{(i)} \rangle \right) y^{(i)} \tilde{z}^{(i)} \right\}$$

The rest of the analysis is identical to that of the proof of Theorem 5 in Appendix D.3. Specifically, by defining

$$\overline{\boldsymbol{u}}_\perp^t := \overline{\boldsymbol{u}} - \langle \overline{\boldsymbol{w}}^t, \overline{\boldsymbol{u}} \rangle \overline{\boldsymbol{w}}^t,$$

we have

$$
\begin{aligned}
\frac{\mathrm{d}\langle \overline{\boldsymbol{w}}^t, \overline{\boldsymbol{u}} \rangle}{\mathrm{d}t} =& \langle \overline{\boldsymbol{u}}_\perp^t, \mathbb{E}_{\boldsymbol{z},y}[\phi'(\langle \overline{\boldsymbol{w}}^t, \boldsymbol{z} \rangle)y\boldsymbol{z}] \rangle \\
&+ \underbrace{\left\langle \overline{\boldsymbol{u}}_\perp^t, \frac{1}{n}\sum_{i=1}^n \phi'\left(\left\langle \overline{\boldsymbol{w}}^t, \boldsymbol{z}^{(i)} \right\rangle\right)y^{(i)}\boldsymbol{z}^{(i)} - \mathbb{E}_{\boldsymbol{z},y}[\phi'(\langle \overline{\boldsymbol{w}}^t, \boldsymbol{z} \rangle)y\boldsymbol{z}] \right\rangle}_{=:\mathcal{E}_1} \\
&+ \underbrace{\frac{1}{n}\sum_{i=1}^n \left\{ \phi'\left(\left\langle \overline{\boldsymbol{w}}^t, \tilde{\boldsymbol{z}}^{(i)} \right\rangle\right) - \phi'\left(\left\langle \overline{\boldsymbol{w}}^t, \boldsymbol{z}^{(i)} \right\rangle\right) \right\} y^{(i)}\left\langle \boldsymbol{z}^{(i)}, \overline{\boldsymbol{u}}_\perp^t \right\rangle}_{=:\mathcal{E}_2} \\
&+ \underbrace{\frac{1}{n}\sum_{i=1}^n \phi'\left(\left\langle \overline{\boldsymbol{w}}, \tilde{\boldsymbol{z}}^{(i)} \right\rangle\right) y^{(i)}\left\langle \tilde{\boldsymbol{z}}^{(i)} - \boldsymbol{z}^{(i)}, \overline{\boldsymbol{u}}_\perp^t \right\rangle}_{=:\mathcal{E}_3}.
\end{aligned}
$$

As long as $n \gtrsim d$, $n' = n$ for the smooth case, and $n' \gtrsim n^2$ for the ReLU case, the first two steps of the proof of Theorem 5 in Appendix D.3 implies that

$$\mathcal{E}_1 \wedge \mathcal{E}_2 \wedge \mathcal{E}_3 \geq -\|\overline{\boldsymbol{u}}_\perp^t\|\tilde{\mathcal{O}}(\sqrt{d/n}) \geq -\tilde{\mathcal{O}}(\sqrt{d/n}).$$

Once again, we apply Lemma 11 to obtain

$$\mathbb{E}[\phi'(\langle \overline{\boldsymbol{w}}, \boldsymbol{z} \rangle)y\boldsymbol{z}] = -\zeta_{\phi,g}(\langle \overline{\boldsymbol{w}}, \overline{\boldsymbol{u}} \rangle)\overline{\boldsymbol{u}} - \psi_{\phi,g}(\langle \overline{\boldsymbol{w}}^t, \overline{\boldsymbol{u}} \rangle)\overline{\boldsymbol{w}},$$

with $\zeta_{\phi,g}$ and $\psi_{\phi,g}$ given in (D.2) and (D.3) respectively. As a result,

$$
\begin{aligned}
\frac{\mathrm{d}\langle \overline{\boldsymbol{w}}^t, \overline{\boldsymbol{u}} \rangle}{\mathrm{d}t} &\geq -\zeta_{\phi,g}(\langle \overline{\boldsymbol{w}}^t, \overline{\boldsymbol{u}} \rangle)\|\overline{\boldsymbol{u}}_\perp^t\|^2 - \tilde{\mathcal{O}}(\sqrt{d/n}) \\
&\geq c\langle \overline{\boldsymbol{w}}^t, \overline{\boldsymbol{u}} \rangle^{s-1}(1 - \langle \overline{\boldsymbol{w}}^t, \overline{\boldsymbol{u}} \rangle^2) - \tilde{\mathcal{O}}(\sqrt{d/n}) \qquad \text{(By Assumption 2).}
\end{aligned}
$$

We need to ensure the noise term, i.e. the second term on the RHS remains smaller than the signal, i.e. the first term. The signal term attains its minimum at either initialization $t = 0$ or at the end of the trajectory $t = T$ where $\langle \overline{\boldsymbol{w}}^T, \overline{\boldsymbol{u}} \rangle = 1 - \varepsilon$, which imposes the following sufficient conditions on $n$. Namely, at initialization we require

$$n \gtrsim Cd\langle \overline{\boldsymbol{w}}^0, \overline{\boldsymbol{u}} \rangle^{2(1-s)},$$

while at $t = T$ we require

$$n \gtrsim Cd/\varepsilon^2,$$

where $C$ hides constant that only depend on $s$ and polylogarithmic factors of $d$. Hence, we obtain

$$\frac{\mathrm{d}\langle \overline{\boldsymbol{w}}^t, \overline{\boldsymbol{u}} \rangle}{\mathrm{d}t} \geq c'\langle \overline{\boldsymbol{w}}^t, \overline{\boldsymbol{u}} \rangle^{s-1}(1 - \langle \overline{\boldsymbol{w}}^t, \overline{\boldsymbol{u}} \rangle^2).$$

for some universal constant $c' > 0$. Via integration (similar to the proof of Proposition 4 in Appendix D.1), for

$$T_1 := \sup\{t > 0 : \langle \overline{\boldsymbol{w}}^t, \overline{\boldsymbol{u}} \rangle < 1/2\}$$

we obtain

$$T_1 \lesssim \tau_s(\langle \overline{\boldsymbol{w}}^0, \overline{\boldsymbol{u}} \rangle),$$

and for

$$T_2 := \sup\{t > 0 : \langle \overline{\boldsymbol{w}}^t, \overline{\boldsymbol{u}} \rangle < 1 - \varepsilon\}$$

we obtain $T_2 - T_1 \lesssim \ln(1/\varepsilon)$, which completes the proof. We conclude by remarking that the proof of Corollary 8 is immediate given Theorem 7 and Lemma 19. □

### E.2 Preliminary Lemmas for Proving Theorem 9

We will adapt the following lemma from [MHPG$^+$23], which provides a non-parametric approximation of $g$ via random biases.

**Lemma 20.** [MHPG$^+$23, Lemma 22] *For any smooth $g : \mathbb{R} \to \mathbb{R}$ and $\Delta > 0$, let $\tilde{g} : \mathbb{R} \to \mathbb{R}$ be a smooth function such that $\tilde{g}(z) = g(z)$ for $|z| \leq \Delta$ and $\tilde{g}(-2\Delta) = \tilde{g}'(-2\Delta) = 0$. Suppose $\{b_j\}_{j=1}^m \overset{i.i.d.}{\sim} \mathrm{Unif}(-2\Delta, 2\Delta)$, and let $\Delta_* := \Delta \sup_{|z| \leq 2\Delta} |\tilde{g}''(z)|$. Then, there exist second layer weights $\{a_j(b_j)\}_{j=1}^m$ with $\|\boldsymbol{a}\| \lesssim \Delta_*/\sqrt{m}$, such that for any fixed $z \in [-\Delta, \Delta]$ and any $\delta > 0$, with probability at least $1 - \delta$ over the random biases,*

$$\left| \sum_{j=1}^m a(b_j)\phi(z + b_j) - g(b_j) \right| \lesssim \Delta\Delta_* \sqrt{\frac{\ln(1/\delta)}{m}},$$

*where $\phi$ is the ReLU activation.*

We use the above lemma to show the existence of a second layer with $\tilde{\mathcal{O}}(1/\sqrt{m})$ norm with training error of order $\tilde{\mathcal{O}}(1/m)$.

**Lemma 21.** *For any $\varepsilon < 1$, suppose $\langle \overline{\boldsymbol{w}}, \overline{\boldsymbol{u}} \rangle \geq 1 - \varepsilon$. Then for any $q > 0$, sufficiently large $d$, $n \gtrsim d/\varepsilon^2$, with probability at least $1 - \mathcal{O}(d^{-q})$ over the random biases and the dataset, there exists a second layer $\boldsymbol{a}$ with $\|\boldsymbol{a}\| \leq \tilde{\mathcal{O}}(1/\sqrt{m})$ described by Lemma 20 such that*

$$\frac{1}{n} \sum_{i=1}^n \left( \sum_{j=1}^m a_j \phi(\langle \overline{\boldsymbol{w}}, \tilde{\boldsymbol{z}}^{(i)} \rangle + b_j) - y^{(i)} \right)^2 \lesssim \mathbb{E}[\epsilon^2] + \tilde{\mathcal{O}}(1/m + \varepsilon),$$

*where $\phi$ is the ReLU activation.*

**Proof.** We begin by replacing $\overline{\boldsymbol{w}}$ and $\tilde{\boldsymbol{z}}^{(i)}$ with $\overline{\boldsymbol{u}}$ and $\boldsymbol{z}^{(i)}$. Specifically, via Jensen's inequality,

$$\frac{1}{n} \sum_{i=1}^n \left\{ \sum_{j=1}^m a_j \phi\left( \langle \overline{\boldsymbol{w}}, \tilde{\boldsymbol{z}}^{(i)} \rangle + b_j \right) - y^{(i)} \right\}^2 \leq \underbrace{\frac{4}{n} \sum_{i=1}^n \left\{ \sum_{j=1}^m a_j \phi\left( \langle \overline{\boldsymbol{u}}, \boldsymbol{z}^{(i)} \rangle + b_j \right) - g\left( \langle \overline{\boldsymbol{u}}, \boldsymbol{z}^{(i)} \rangle \right) \right\}^2}_{=:\mathcal{E}_1}$$

$$\leq \underbrace{\frac{4}{n} \sum_{i=1}^n \left\{ y^{(i)} - g\left( \langle \overline{\boldsymbol{u}}, \boldsymbol{z}^{(i)} \rangle \right) \right\}^2}_{=:\mathcal{E}_2}$$

$$\leq \underbrace{\frac{4}{n} \sum_{i=1}^n \left( \sum_{j=1}^m a_j \left\{ \phi\left( \langle \overline{\boldsymbol{w}}, \tilde{\boldsymbol{z}}^{(i)} \rangle + b_j \right) - \phi\left( \langle \overline{\boldsymbol{w}}, \boldsymbol{z}^{(i)} \rangle + b_j \right) \right\} \right)^2}_{=:\mathcal{E}_3}$$

$$\leq \underbrace{\frac{4}{n} \sum_{i=1}^n \left( \sum_{j=1}^m a_j \left\{ \phi\left( \langle \overline{\boldsymbol{w}}, \boldsymbol{z}^{(i)} \rangle + b_j \right) - \phi\left( \langle \overline{\boldsymbol{u}}, \boldsymbol{z}^{(i)} \rangle + b_j \right) \right\} \right)^2}_{=:\mathcal{E}_4}.$$

We bound each term separately. For $\mathcal{E}_1$, we can invoke Lemma 20 which implies that each term in the sum can be bounded by $\tilde{\mathcal{O}}(1/m)$ with probability at least $1 - 1/(nd^q)$, thus by a union bound, with probability at least $1 - d^{-q}$ over the random biases,

$$\mathcal{E}_1 \leq \tilde{\mathcal{O}}(1/m).$$

By sub-Guassianity of $\epsilon^{(i)}$ (hence sub-exponentiality of $\epsilon^{(i)^2}$), for $n \gtrsim d$ (with a sufficiently large constant) we have

$$\mathcal{E}_2 \lesssim \mathbb{E}[\epsilon^2] + \sqrt{d/n},$$

with probability at least $1 - e^{-d}$.

For $\mathcal{E}_3$, via the Lipschitzness of ReLU and the Cauchy-Schwartz inequality we can write

$$\mathcal{E}_3 \leq \frac{\tilde{\mathcal{O}}(1)}{n} \sum_{i=1}^{n} \left\langle \overline{w}, \tilde{z}^{(i)} - z^{(i)} \right\rangle^2 \leq \tilde{\mathcal{O}}(1) \|\mathbf{I} - \hat{\Sigma}^{-1/2} \Sigma^{1/2}\|^2 \leq \tilde{\mathcal{O}}(d/n'),$$

where we used the event of Lemma 24 which happens with probability at least $1 - 2e^{-d}$, and $\tilde{\mathcal{O}}(1)$ represents a constant that depends at most polylogarithmically on $d$.

Finally, we bound the last term. Once again via the Lipschitzness of the ReLU activation and the Cauchy-Schwartz inequality

$$\mathcal{E}_4 \leq \frac{\tilde{\mathcal{O}}(1)}{n} \sum_{i=1}^{n} \left\langle \overline{w} - \overline{u}, z^{(i)} \right\rangle^2 \leq \tilde{\mathcal{O}}(\|\overline{w} - \overline{u}\|^2) \leq \tilde{\mathcal{O}}(\varepsilon),$$

where once again we used the event of Lemma 24. On the intersection of all desired events, we have

$$\frac{1}{n} \sum_{i=1}^{n} \left( \sum_{j=1}^{m} a_j \phi(\left\langle \overline{w}, \tilde{z}^{(i)} \right\rangle + b_j) - y^{(i)} \right) \lesssim \mathbb{E}\left[\epsilon^2\right] + \tilde{\mathcal{O}}(1/m + \sqrt{d/n} + d/n' + \varepsilon).$$

We conclude the proof by noticing that $n' \gtrsim n^2$ and $n \gtrsim d\varepsilon^{-2}$. $\qquad \square$

Additionally, we will use the following standard Lemma on the Rademacher complexity of two-layer neural networks, which in particular is a restatement of [MHPG$^+$23, Lemma 18] in a way suitable for our analysis.

**Lemma 22.** *Let $\mathcal{F}$ be a class of real-valued functions on $(z, y)$. Given $n$ samples $\{z^{(i)}, y\}_{i=1}^{n}$, define the empirical Rademacher complexity of $\mathcal{F}$ as*

$$\hat{\mathfrak{R}}_n(\mathcal{F}) := \mathbb{E}_{(\varsigma_i)_{i=1}^n} \left[ \sup_{f \in \mathcal{F}} \frac{1}{n} \sum_{i=1}^{n} \varsigma_i f(z^{(i)}, y^{(i)}) \right],$$

*where $(\varsigma_i)$ are i.i.d. Rademacher random variables (i.e. $\pm 1$ with equal probability). Suppose $\mathcal{F}$ is given by*

$$\mathcal{F} := \left\{ (z, y) \mapsto \left( \sum_{j=1}^{m} a_j \phi(\langle \overline{u}, z \rangle + b_j) - y \right)^2 \wedge C : \|a\| \leq r_a/\sqrt{m}, \quad |b_j| \leq r_b, \forall 1 \leq j \leq m \right\},$$

*for some fixed $\overline{u} \in \mathbb{S}^{d-1}$. Suppose $\{z^{(i)}\}_{i=1}^{n} \overset{i.i.d.}{\sim} \mathcal{N}(0, \mathbf{I}_d)$, and suppose $|\phi'| \leq 1$. Then,*

$$\mathbb{E}_{(z^{(i)}, y^{(i)})_{i=1}^n} \left[ \hat{\mathfrak{R}}_n(\mathcal{F}) \right] \leq \frac{2\sqrt{2C}(1 + r_b)r_a}{\sqrt{n}}.$$

**Proof.** See the proof of [MHPG$^+$23, Lemma 18]. $\qquad \square$

### E.3 Proof of Theorem 9

Throughout the proof, we will assume $\langle \overline{w}, \overline{u} \rangle \geq 1 - \varepsilon$ where we recall

$$\overline{w} := \frac{\hat{\Sigma}^{1/2} w}{\|\hat{\Sigma}^{1/2} w\|} \quad \text{and} \quad \overline{u} := \frac{\Sigma^{1/2} u}{\|\Sigma^{1/2} u\|}.$$

From either Theorem 5 or Theorem 7, we can assume $\langle \overline{w}, \overline{u} \rangle \geq 1 - \varepsilon$ with probability at least $1 - \mathcal{O}(d^{-q})$ for any fixed $q > 0$. For simplicity, let

$$\hat{y}(\tilde{z}; \overline{w}) = \sum_{j=1}^{m} a_j \phi(\langle \overline{w}, \tilde{z} \rangle + b_j),$$

and similarly define $\hat{y}(\boldsymbol{z}; \overline{\boldsymbol{u}})$. We define the following quantities,

$$\mathcal{R}(\overline{\boldsymbol{w}}) := \mathbb{E}_{\boldsymbol{z},y}\Big[(\hat{y}(\tilde{\boldsymbol{z}}; \overline{\boldsymbol{w}}) - y)^2\Big] \qquad \text{and} \qquad R(\overline{\boldsymbol{u}}) := \mathbb{E}_{\boldsymbol{z},y}\Big[(\hat{y}(\boldsymbol{z}; \overline{\boldsymbol{u}}) - y)^2\Big], \qquad \text{(E.1)}$$

and similarly define their empirical counterparts,

$$\hat{\mathcal{R}}(\overline{\boldsymbol{w}}) := \frac{1}{n}\sum_{i=1}^{n}\Big(\hat{y}(\tilde{\boldsymbol{z}}^{(i)}; \overline{\boldsymbol{w}}) - y^{(i)}\Big)^2 \quad \text{and} \quad \hat{R}(\overline{\boldsymbol{u}}) := \frac{1}{n}\sum_{i=1}^{n}\Big(\hat{y}(\boldsymbol{z}^{(i)}; \overline{\boldsymbol{u}}) - y^{(i)}\Big)^2. \qquad \text{(E.2)}$$

Notice that ultimately, we are interested in bounding $\mathcal{R}(\overline{\boldsymbol{w}})$. We break down the proof into three steps. In the first step, we show that $\mathcal{R}(\overline{\boldsymbol{w}})$ can be upper bounded by $R(\overline{\boldsymbol{u}})$. Then, via a generalization bound, we show that the $R(\overline{\boldsymbol{u}})$ can be upper bounded by $\hat{R}(\overline{\boldsymbol{u}})$. Finally, we show that $\hat{R}(\overline{\boldsymbol{u}})$ can be upper bounded by the training error, i.e. $\hat{\mathcal{R}}(\overline{\boldsymbol{w}})$, and convex optimization of the last layer can attain the near-optimal value of this training error which is bounded by Lemma 21.

**Step 1. Bounding $\mathcal{R}(\overline{\boldsymbol{w}})$ via $R(\overline{\boldsymbol{u}})$.** By Jensen's inequality,

$$\mathbb{E}_{\boldsymbol{z},y}\Big[(\hat{y}(\tilde{\boldsymbol{z}}; \overline{\boldsymbol{w}}) - y)^2\Big] \leq 3\,\mathbb{E}_{\boldsymbol{z}}\Big[(\hat{y}(\tilde{\boldsymbol{z}}; \overline{\boldsymbol{w}}) - \hat{y}(\boldsymbol{z}; \overline{\boldsymbol{w}}))^2\Big] + 3\,\mathbb{E}_{\boldsymbol{z}}\Big[(\hat{y}(\boldsymbol{z}; \overline{\boldsymbol{w}}) - \hat{y}(\boldsymbol{z}; \overline{\boldsymbol{u}}))^2\Big] + 3\,\mathbb{E}_{\boldsymbol{z},y}\Big[(\hat{y}(\boldsymbol{z}; \overline{\boldsymbol{u}}) - y)^2\Big].$$

Suppose $\|\boldsymbol{a}\| \leq r_a/\sqrt{m}$. For the first term, by Lipschitzness of $\phi$ and the Cauchy-Schwartz inequality

$$\mathbb{E}_{\boldsymbol{z}}\Big[(\hat{y}(\tilde{\boldsymbol{z}}; \overline{\boldsymbol{w}}) - \hat{y}(\boldsymbol{z}; \overline{\boldsymbol{w}}))^2\Big] = \mathbb{E}_{\boldsymbol{z}}\left[\left(\sum_{j=1}^{m} a_j\Big\{\phi\big(\big\langle \overline{\boldsymbol{w}}, \tilde{\boldsymbol{z}}^{(i)}\big\rangle + b_j\big) - \phi\big(\big\langle \overline{\boldsymbol{w}}, \boldsymbol{z}^{(i)}\big\rangle + b_j\big)\Big\}\right)^2\right]$$

$$\leq r_a^2\,\mathbb{E}_{\boldsymbol{z}}\Big[\langle \overline{\boldsymbol{w}}, \tilde{\boldsymbol{z}} - \boldsymbol{z}\rangle^2\Big]$$

$$\leq r_a^2\|\mathbf{I}_d - \hat{\boldsymbol{\Sigma}}^{-1/2}\boldsymbol{\Sigma}^{1/2}\|^2 \lesssim r_a^2 d/n',$$

where the last inequality holds with probability at least $1 - 2e^{-d}$ on the event of Lemma 24.

For the middle term, via a similar argument,

$$\mathbb{E}_{\boldsymbol{z}}\Big[(\hat{y}(\boldsymbol{z}; \overline{\boldsymbol{w}}) - \hat{y}(\boldsymbol{z}; \overline{\boldsymbol{u}}))^2\Big] \leq r_a^2\,\mathbb{E}_{\boldsymbol{z}}\Big[\langle \overline{\boldsymbol{w}} - \overline{\boldsymbol{u}}, \boldsymbol{z}\rangle^2\Big] \leq 2r_a\varepsilon.$$

In what follows, we will restrict the analysis to the case where $r_a = \tilde{\mathcal{O}}(1)$. Therefore, we have

$$\mathbb{E}_{\boldsymbol{z},y}\Big[(\hat{y}(\tilde{\boldsymbol{z}}; \overline{\boldsymbol{w}}) - y)^2\Big] \leq 3\,\mathbb{E}_{\boldsymbol{z},y}\Big[(\hat{y}(\boldsymbol{z}; \overline{\boldsymbol{u}}) - y)^2\Big] + \tilde{\mathcal{O}}(d/n' + \varepsilon).$$

**Step 2. Generalization: Bounding $R(\overline{\boldsymbol{u}})$ via $\hat{R}(\overline{\boldsymbol{u}})$.**

Define the event

$$E := \Big\{|\langle \overline{\boldsymbol{u}}, \boldsymbol{z}\rangle| \vee |\epsilon| \leq \sqrt{2\ln(nd^q)}\Big\}.$$

and similarly define $E^{(i)}$ by replacing $\boldsymbol{z}$ and $\epsilon$ with $\boldsymbol{z}^{(i)}$ and $\epsilon^{(i)}$ respectively. Via the Cauchy-Schwartz and Jensen inequalities

$$\mathbb{E}_{\boldsymbol{z},y}\Big[(\hat{y}(\boldsymbol{z}; \overline{\boldsymbol{u}}) - y)^2\Big] = \mathbb{E}_{\boldsymbol{z},y}\Big[(\hat{y}(\boldsymbol{z}; \overline{\boldsymbol{u}}) - y)^2\mathbf{1}(E)\Big] + \mathbb{E}_{\boldsymbol{z},y}\Big[(\hat{y}(\boldsymbol{z}; \overline{\boldsymbol{u}}) - y)^2\mathbf{1}(E^C)\Big]$$

$$\leq \mathbb{E}_{\boldsymbol{z},y}\Big[(\hat{y}(\boldsymbol{z}; \overline{\boldsymbol{u}}) - y)^2\mathbf{1}(E)\Big] + \sqrt{8}\big(\mathbb{E}\big[\hat{y}(\boldsymbol{z}; \overline{\boldsymbol{u}})^4\big] + \mathbb{E}\big[y^4\big]\big)^{1/2}\mathbb{P}\big(E^C\big)^{1/2}$$

Moreover, $\mathbb{E}\big[y^4\big] \lesssim 1$, $\mathbb{E}_{\boldsymbol{z},y}\big[\hat{y}(\boldsymbol{z}; \overline{\boldsymbol{u}})^4\big] \leq \tilde{\mathcal{O}}(1)$, and $\mathbb{P}\big(E^C\big) \leq 4/(nd^q)$ (via a standard sub-Gaussian tail bound). Consequently,

$$\mathbb{E}_{\boldsymbol{z},y}\Big[(\hat{y}(\boldsymbol{z}; \overline{\boldsymbol{u}}) - y)^2\Big] \leq \mathbb{E}_{\boldsymbol{z},y}\Big[(\hat{y}(\boldsymbol{z}; \overline{\boldsymbol{u}}) - y)^2\mathbf{1}(E)\Big] + \tilde{\mathcal{O}}(n^{-1}d^{-q}),$$

Let

$$\ell(\boldsymbol{z}^{(i)}, y^{(i)}; \boldsymbol{a}, \boldsymbol{b}) := \left(\sum_{j=1}^{m} a_j\phi\big(\big\langle \overline{\boldsymbol{u}}, \boldsymbol{z}^{(i)}\big\rangle + b_j\big) - y^{(i)}\right)^2\mathbf{1}(E).$$

Notice that $\overline{u}$ is fixed. Then, by a standard symmetrization argument (see e.g. [VH16, Lemma 7.4]) and Lemma 22

$$\mathbb{E}\left[\sup_{\|\boldsymbol{a}\|\leq r_a/\sqrt{m},|b_j|\leq r_b} \mathbb{E}_{\boldsymbol{z},y}[\ell(\boldsymbol{z},y;\boldsymbol{a},\boldsymbol{b})] - \frac{1}{n}\sum_{i=1}^n \ell(\boldsymbol{z}^{(i)},y^{(i)};\boldsymbol{a},\boldsymbol{b})\right] \leq 2\,\mathbb{E}\left[\hat{\mathfrak{R}}_n(\mathcal{F})\right]$$
$$\leq \tilde{\mathcal{O}}(\sqrt{1/n}).$$

where $\hat{\mathfrak{R}}_n(\mathcal{F})$. As the loss is bounded, we can apply McDiarmid's inequality to turn the above bound in expectation into a bound in probability, in particular

$$\sup_{\|\boldsymbol{a}\|\leq r_a/\sqrt{m},|b_j|\leq r_b} \mathbb{E}_{\boldsymbol{z},y}[\ell(\boldsymbol{z},y;\boldsymbol{a},\boldsymbol{b})] - \frac{1}{n}\sum_{i=1}^n \ell(\boldsymbol{z}^{(i)},y^{(i)};\boldsymbol{a},\boldsymbol{b}) \leq \tilde{\mathcal{O}}(\sqrt{d/n})$$

with probability at least $1 - 2e^{-d}$. Therefore, we conclude this step by noticing that

$$\mathbb{E}_{\boldsymbol{z},y}\left[(\hat{y}(\boldsymbol{z};\overline{u}) - y)^2\right] \leq \frac{1}{n}\sum_{i=1}^n \left(\hat{y}(\boldsymbol{z}^{(i)};\overline{u}) - y^{(i)}\right)^2 \mathbf{1}(E^{(i)}) + \tilde{\mathcal{O}}(\sqrt{d/n} + n^{-1}d^{-q})$$
$$\leq \frac{1}{n}\sum_{i=1}^n \left(\hat{y}(\boldsymbol{z}^{(i)};\overline{u}) - y^{(i)}\right)^2 + \tilde{\mathcal{O}}(\sqrt{d/n}),$$

with probability at least $1 - 2e^{-d}$.

**Step 3. Bounding the training error and finishing the proof.** This step is similar to the proof of [MHPG$^+$23, Theorem 4]. For conciseness, define

$$\hat{\mathcal{R}}(\boldsymbol{a}) := \frac{1}{n}\sum_{i=1}^n \left(\hat{y}(\boldsymbol{x}^{(i)};\boldsymbol{W},\boldsymbol{a},\boldsymbol{b}) - y^{(i)}\right)^2.$$

and $\hat{\mathcal{R}}_\lambda(\boldsymbol{a}) := \hat{\mathcal{R}}(\boldsymbol{a}) + \lambda\|\boldsymbol{a}\|^2/2$. Our goal is to choose suitable $\lambda$ such that the minimizer

$$\boldsymbol{a}^* := \arg\min_{\boldsymbol{a}\in\mathbb{R}^m} \hat{\mathcal{R}}(\boldsymbol{a}) + \lambda\|\boldsymbol{a}\|^2/2,$$

satisfies $\|\boldsymbol{a}^*\| \leq r_a/\sqrt{m}$ while the value of the above minimization problem which we denote with $\hat{\mathcal{R}}_\lambda^*$ does not significantly exceed

$$\min_{\|\boldsymbol{a}\|\leq r_a/\sqrt{m}} \hat{\mathcal{R}}(\boldsymbol{a}).$$

We argue that the suitable choice for $\lambda$ is

$$\lambda \asymp \frac{m\,\mathbb{E}[\epsilon^2] + m\varepsilon + 1}{r_a^2} = \tilde{\Theta}(m\,\mathbb{E}[\epsilon^2] + m\varepsilon + 1). \tag{E.3}$$

Let $\hat{\mathcal{R}}^*$ denote the minimizer of the regularized problem and $\tilde{\boldsymbol{a}} := \arg\min_{\|\boldsymbol{a}\|\leq r_a/\sqrt{m}} \hat{\mathcal{R}}(\boldsymbol{a})$. From Lemma 21, with a proper choice of $r_a = \tilde{\Theta}(1)$, we have

$$\hat{\mathcal{R}}(\tilde{\boldsymbol{a}}) \lesssim \mathbb{E}[\epsilon^2] + \tilde{\mathcal{O}}(1/m + \varepsilon).$$

with probability at least $1 - \mathcal{O}(d^{-q})$ over the biases and the dataset. Note that as $\boldsymbol{a}^*$ is the minimizer of $\hat{\mathcal{R}}_\lambda$, we have

$$\hat{\mathcal{R}}(\boldsymbol{a}^*) + \frac{\lambda\|\boldsymbol{a}^*\|^2}{2} \leq \hat{\mathcal{R}}(\tilde{\boldsymbol{a}}) + \frac{\lambda\|\tilde{\boldsymbol{a}}\|^2}{2},$$

and in particular

$$\frac{\lambda\|\boldsymbol{a}^*\|^2}{2} \leq \hat{\mathcal{R}}(\tilde{\boldsymbol{a}}) + \frac{\lambda\|\tilde{\boldsymbol{a}}\|^2}{2} \implies \|\boldsymbol{a}^*\| \leq \tilde{\mathcal{O}}(1/\sqrt{m}).$$

and

$$\hat{\mathcal{R}}(\boldsymbol{a}^*) \leq \hat{\mathcal{R}}(\tilde{\boldsymbol{a}}) + \frac{\lambda\|\tilde{\boldsymbol{a}}\|^2}{2} \lesssim \mathbb{E}[\epsilon]^2 + \tilde{\mathcal{O}}(1/m + \varepsilon).$$

Let $\{a^t\}_{t\geq 0}$ be the solution to the gradient flow of $a$. Then,

$$\frac{\mathrm{d}\|a^t - a^*\|^2}{\mathrm{d}t} = -2\Big\langle a^t - a^*, \nabla\hat{\mathcal{R}}_\lambda(a^t)\Big\rangle,$$

and by the first-order condition of strong convexity

$$\Big\langle a^t - a^*, \nabla\hat{\mathcal{R}}_\lambda(a^t)\Big\rangle \geq \lambda\|a^t - a^*\|^2,$$

therefore

$$\|a^{T'} - a^*\|^2 \leq e^{-2\lambda T'}\|a^0 - a^*\|^2.$$

As the training error (of the regularized problem) is $\lambda$-strongly convex in $a$, by applying the standard Polyak-Łojasiewicz condition, gradient flow for training $a$ obtains

$$\hat{\mathcal{R}}_\lambda(a^{T'}) - \hat{\mathcal{R}}^*_\lambda \leq \Big(\hat{\mathcal{R}}_\lambda(a^0) - \hat{\mathcal{R}}^*_\lambda\Big)e^{-2\lambda T'}.$$

Furthermore, since

$$\|a^*\|^2 - \|a^{T'}\|^2 \leq 2\|a^*\|\|a^{T'} - a^*\| - \|a^{T'} - a^*\|^2 \leq 2\|a^{T'} - a^*\|\|a^*\|,$$

we have

$$\hat{\mathcal{R}}(a^{T'}) - \hat{\mathcal{R}}^* \leq 2\|a^0 - a^*\|\|a^*\|e^{-\lambda T'} + \Big(\hat{\mathcal{R}}_\lambda(a^0) - \hat{\mathcal{R}}^*_\lambda\Big)e^{-2\lambda T'}.$$

Consequently, choosing

$$T' \geq \frac{\ln\left(\frac{\|a^0 - a^*\|}{\|a^*\|}\right)}{\lambda} \vee \frac{\ln\left(\frac{4\|a^0 - a^*\|\|a^*\|}{\varepsilon}\right)}{\lambda} \vee \frac{\ln\left(\frac{2(\hat{\mathcal{R}}_\lambda(a^0) - \hat{\mathcal{R}}^*_\lambda)}{\varepsilon}\right)}{2\lambda}, \tag{E.4}$$

implies

$$\hat{\mathcal{R}}(a^{T'}) \leq \hat{\mathcal{R}}^* + \varepsilon \quad\text{and}\quad \|a^{T'}\| \leq 2\|a^*\| \lesssim \tilde{\mathcal{O}}(1/\sqrt{m}).$$

Therefore

$$\hat{\mathcal{R}}(a^{T'}) \lesssim \mathbb{E}\big[\epsilon^2\big] + \tilde{\mathcal{O}}(1/m + \varepsilon).$$

Recall that

$$\hat{\mathcal{R}}(a^{T'}) = \frac{1}{n}\sum_{i=1}^{n}\Big(\hat{y}(\tilde{z}^{(i)}; \overline{w}) - y^{(i)}\Big)^2,$$

is the final training error which we also denoted by $\hat{\mathcal{R}}(\overline{w})$ earlier in this section when were not focusing on the second layer. From the previous two steps, we know how to bound $\mathcal{R}(\overline{w})$ via $\hat{R}(\overline{u})$. Thus the last step is to upper bound $\hat{R}(\overline{u})$ via $\hat{\mathcal{R}}(\overline{w})$. To that end, via Jensen's inequality

$$\begin{aligned}
\frac{1}{n}\sum_{i=1}^{n}\Big(\hat{y}(z^{(i)}; \overline{u}) - y^{(i)}\Big)^2 \leq &\frac{3}{n}\sum_{i=1}^{n}\Big(\hat{y}(\tilde{z}^{(i)}; \overline{w}) - y^{(i)}\Big)^2 \\
&+ \frac{3}{n}\sum_{i=1}^{n}\Big(\hat{y}(z^{(i)}; \overline{w}) - \hat{y}(\tilde{z}^{(i)}; \overline{w})\Big)^2 \\
&+ \frac{3}{n}\sum_{i=1}^{n}\Big(\hat{y}(z^{(i)}; \overline{w}) - \hat{y}(z^{(i)}; \overline{u})\Big)^2.
\end{aligned}$$

The first term on the RHS is $\hat{\mathcal{R}}(\overline{w})$ for which we developed a bound earlier in this step. Bounding the latter two terms can be performed similarly to the arguments in the previous sections. In particular,

$$\frac{1}{n}\sum_{i=1}^{n}\Big(\hat{y}(z^{(i)}; \overline{w}) - \hat{y}(\tilde{z}^{(i)}; \overline{w})\Big)^2 \leq r_a^2\|\mathbf{I}_d - \hat{\boldsymbol{\Sigma}}^{-1/2}\boldsymbol{\Sigma}^{1/2}\|^2\|\frac{1}{n}\sum_{i=1}^{n}z^{(i)}z^{(i)\top}\|^2 \leq \tilde{\mathcal{O}}(d/n'),$$

where the last inequality holds with probability at least $1 - 2e^{-d}$ (over the event of Lemma 24). Similarly,

$$\frac{1}{n}\sum_{i=1}^{n}\Big(\hat{y}(z^{(i)}; \overline{w}) - \hat{y}(z^{(i)}; \overline{u})\Big)^2 \leq r_a\|\overline{w} - \overline{u}\|^2 \leq \tilde{\mathcal{O}}(\varepsilon).$$

Putting the bounds back together (recall $n' \geq n \gtrsim d\varepsilon^{-2}$), we arrive at

$$\hat{R}(\overline{\bm{u}}) \lesssim \hat{\mathcal{R}}(\overline{\bm{w}}) + \tilde{\mathcal{O}}(\varepsilon + d/n') \lesssim \hat{\mathcal{R}}(\overline{\bm{w}}) + \tilde{\mathcal{O}}(\varepsilon).$$

Combining the result of this step with the two previous steps implies

$$\mathcal{R}(\overline{\bm{w}}) \lesssim \mathbb{E}\big[\epsilon^2\big] + \tilde{\mathcal{O}}(1/m + \varepsilon),$$

with probability at least $1 - \mathcal{O}(d^{-q})$ (when conditioned on $\langle \overline{\bm{w}}^0, \overline{\bm{u}} \rangle > 0$) which completes the proof of Theorem 9. $\qquad\square$

# F  Auxiliary Lemmas

In this section, we recall a number of standard lemmas which we employ in various parts of our proofs.

**Lemma 23.** [Wai19, Theorem 6.1]. *Suppose $\{\bm{x}^{(i)}\}_{i=1}^{n'} \overset{i.i.d.}{\sim} \mathcal{N}(0, \bm{\Sigma})$. Let $\hat{\bm{\Sigma}} := \frac{1}{n'} \sum_{i=1}^{n'} \bm{x}^{(i)} \bm{x}^{(i)\top}$. Then, for $n' \geq \mathrm{tr}(\bm{\Sigma})/\lambda_{\max}(\bm{\Sigma})$,*

$$\lambda_{\max}\Big(\hat{\bm{\Sigma}}\Big) \leq \lambda_{\max}(\bm{\Sigma})\left(4 + 5\sqrt{\frac{\mathrm{tr}(\bm{\Sigma})}{n'\lambda_{\max}(\bm{\Sigma})}}\right)$$

*with probability at least $1 - e^{-n'/2}$. Furthermore, for $n' \geq d$,*

$$\lambda_{\min}\Big(\hat{\bm{\Sigma}}\Big) \geq \lambda_{\min}(\bm{\Sigma})\left(\frac{1}{4} - \sqrt{\frac{\mathrm{tr}(\bm{\Sigma})}{n\lambda_{\min}(\bm{\Sigma})}}\right)$$

*with probability at least $1 - e^{-n'/8}$.*

**Lemma 24.** *Suppose $\{\bm{z}^{(i)}\}_{i=1}^{n'} \overset{i.i.d.}{\sim} \mathcal{N}(0, \bm{I}_d)$, let $\bm{x}^{(i)} := \bm{\Sigma}^{1/2} \bm{z}^{(i)}$ for some invertible $\bm{\Sigma}$, and define*

$$\hat{\bm{\Sigma}} := \frac{1}{n'} \sum_{i=1}^{n'} \bm{x}^{(i)} \bm{x}^{(i)\top}.$$

*Then*

$$\|\bm{I}_d - \hat{\bm{\Sigma}}^{1/2} \bm{\Sigma}^{-1/2}\| \vee \|\bm{I}_d - \hat{\bm{\Sigma}}^{-1/2} \bm{\Sigma}^{1/2}\| \lesssim \sqrt{\frac{d}{n'}}$$

*with probability at least $1 - 2e^{-d}$.*

**Proof.** We have

$$
\begin{aligned}
\|\bm{I}_d - \hat{\bm{\Sigma}}^{-1/2} \bm{\Sigma}^{1/2}\| &= \left\{ \lambda_{\max}\Big(\hat{\bm{\Sigma}}^{-1/2} \bm{\Sigma}^{1/2}\Big) - 1 \right\} \vee \left\{ 1 - \lambda_{\min}\Big(\hat{\bm{\Sigma}}^{-1/2} \bm{\Sigma}^{1/2}\Big) \right\} \\
&= \left\{ \lambda_{\max}\Big(\bm{\Sigma}^{1/2} \hat{\bm{\Sigma}}^{-1} \bm{\Sigma}^{1/2}\Big)^{1/2} - 1 \right\} \vee \left\{ 1 - \lambda_{\min}\Big(\bm{\Sigma}^{1/2} \hat{\bm{\Sigma}}^{-1} \bm{\Sigma}^{1/2}\Big) \right\} \\
&= \left\{ \lambda_{\min}\Big(\bm{\Sigma}^{-1/2} \hat{\bm{\Sigma}} \bm{\Sigma}^{-1/2}\Big)^{-1/2} - 1 \right\} \vee \left\{ 1 - \lambda_{\max}\Big(\bm{\Sigma}^{-1/2} \hat{\bm{\Sigma}} \bm{\Sigma}^{-1/2}\Big)^{-1/2} \right\} \\
&= \left\{ \lambda_{\min}\bigg(\frac{1}{n'} \sum_{i=1}^{n'} \bm{z}^{(i)} \bm{z}^{(i)\top}\bigg)^{-1/2} - 1 \right\} \vee \left\{ 1 - \lambda_{\max}\bigg(\frac{1}{n'} \sum_{i=1}^{n'} \bm{z}^{(i)} \bm{z}^{(i)\top}\bigg)^{-1/2} \right\}.
\end{aligned}
$$

Similarly,

$$\|\bm{I}_d - \hat{\bm{\Sigma}}^{1/2} \bm{\Sigma}^{-1/2}\| = \left\{ \lambda_{\max}\bigg(\frac{1}{n'} \sum_{i=1}^{n'} \bm{z}^{(i)} \bm{z}^{(i)\top}\bigg)^{1/2} - 1 \right\} \vee \left\{ 1 - \lambda_{\min}\bigg(\frac{1}{n'} \sum_{i=1}^{n'} \bm{z}^{(i)} \bm{z}^{(i)\top}\bigg)^{1/2} \right\}$$

Moreover, by [Wai19, Example 6.2], we have with probability at least $1 - 2e^{-d}$,

$$\lambda_{\max}\left(\frac{1}{n'}\sum_{i=1}^{n'}\boldsymbol{z}^{(i)}\boldsymbol{z}^{(i)\top}\right) \leq 1+(\sqrt{2}+1)\sqrt{\frac{d}{n'}} \quad \text{and} \quad \lambda_{\min}\left(\frac{1}{n'}\sum_{i=1}^{n'}\boldsymbol{z}^{(i)}\boldsymbol{z}^{(i)\top}\right) \geq 1-(\sqrt{2}+1)\sqrt{\frac{d}{n'}}.$$

Thus, for $n' \gtrsim d$ (with a sufficiently large absolute constant), we have

$$\|\mathbf{I}_d - \hat{\boldsymbol{\Sigma}}^{1/2}\hat{\boldsymbol{\Sigma}}^{-1/2}\| \vee \|\mathbf{I}_d - \hat{\boldsymbol{\Sigma}}^{-1/2}\boldsymbol{\Sigma}^{1/2}\| \lesssim \sqrt{\frac{d}{n'}}$$

with probability at least $1 - 2e^{-d}$. $\qquad\square$

**Lemma 25** (Chernoff's Inequality). *Suppose $X_1, \ldots, X_n$ are i.i.d. Bernoulli random variables, and further assume that $\mathbb{E}[\sum_i X_i] \leq \mu$. Then, for any $\delta \geq 1$,*

$$\mathbb{P}\left(\sum_{i=1}^{n} X_i \geq \mu(1+\delta)\right) \leq e^{-\mu\delta/3}. \tag{F.1}$$

**Proof.** The proof follows from a standard Chernoff bound. From [Ver18, Theorem 2.3.1]

$$\mathbb{P}\left(\sum_i X_i \geq \mu(1+\delta)\right) \leq e^{\mu(\delta-(1+\delta)\ln(1+\delta))},$$

(notice that the statement of [Ver18, Theorem 2.3.1] holds true even when $\mathbb{E}[\sum_i X_i] = \mu$ is replaced with $\mathbb{E}[\sum_i X_i] \leq \mu$). We conclude by remarking that $\delta - (1+\delta)\ln(1+\delta) \leq -\delta/3$ for $\delta \geq 1$. $\quad\square$

