# OpenReview forum: "Gradient-Based Feature Learning under Structured Data"
_NeurIPS.cc/2023/Conference — NeurIPS 2023 poster_

### Official Review · Reviewer_oCpw · 2023-06-27

**Soundness:** 4 excellent
**Presentation:** 3 good
**Contribution:** 3 good
**Rating:** 7
**Confidence:** 3

**Summary:**

Summary: The authors consider single index learning with a spiked input distribution, rather than the Gaussian typical in most works in this setting, showing that a modified SGD can learn where vanilla SGD fails and showing better sample complexity under the assumption that the spike is aligned with the true hidden signal.


**Strengths:**

Strengths: The question of moving beyond the Gaussian input distribution is an interesting one, and the results seem quite interesting.  In particular, the transitions in behavior depending on the alignment of the spike provide a rather full picture of the setting.


**Weaknesses:**

Weaknesses: The setting is somewhat artificial, as the spike strongly correlating with the true signal is a much easier setting than the typical single index model.  Furthermore, it’s not instantly clear how this version of SGD performs according to a somewhat silly baseline that’s aware of the possible correlation of the spike and true signal.  From the intuition of [1], most of the effort of the learning process is spent trying to find non-negligible correlation between the learned parameter and the true direction.  If you suspect the spike of your input distribution is already meaningfully correlated, you could simply initialize there.  Perhaps this is a naive idea ([1] requires the link function to be known for instance), but a little discussion of why the spiked covariance setting is not too easy would strengthen the paper I feel.

[1] Arous, Gerard Ben, Reza Gheissari, and Aukosh Jagannath. "Online stochastic gradient descent on non-convex losses from high-dimensional inference." The Journal of Machine Learning Research 22.1 (2021): 4788-4838.


**Questions:**

N/A

---

> ### Author Rebuttal · Authors · 2023-08-10
>
> We thank the reviewer for their thoughtful feedback and positive assessment.
>
> **Initialization;** We remark that our main objective was to investigate the different complexity regimes arising from structural variations in data, rather than designing optimal algorithms for the learning task at hand. Nevertheless, the reviewer's suggestion to initialize at the spike direction is in fact very useful and will be discussed in the main text; the resulting sample complexity after spectral initialization can be directly obtained from Theorems 5 and 7 as they hold for general initialization. In particular, up to polylog factors, the sample complexity from Theorem 5 (without preconditioning) would read $d^{1 + 2r_2 + ((s-1)(2r_1 - r_2) \lor 0)}$ and that from Theorem 7 (with preconditioning) would read $d^{1 + ((s-1)(2r_1 - r_2) \lor 0)}$. As expected, these complexities are strictly better than what is achieved under random initialization (Corollaries 6 and 8), and notably, the sample complexity for the preconditioned algorithm becomes linear in $d$ (up to polylog factors) as soon as $r_2 > 2r_1$.
>
> **Further discussion:** We remark that while spectral initialization does provide some benefit under the spiked model, it does not always trivialize the problem. More specifically, unless we have $r_2 > 2r_1$, the information exponent still affects the sample complexity. Furthermore, there are settings where uniform initialization has smaller sample and time complexity than initializing at the spike direction, e.g. if the spike-target alignment is worse than uniformly at random (i.e. $r_1 > 0.5$). This phenomenon can more naturally occur under a general covariance structure, where e.g. if the target direction aligns well with some other eigenvector of $\boldsymbol{\Sigma}$, Theorems 5 and 7 show that the convergence time and sample complexity of initializing from the first eigenvector can be arbitrarily large.
>
> We thank the reviewer for pointing to this and we will add a detailed discussion about this observation in the final version of the manuscript.
> ***
>
> We would be happy to clarify any concerns or answer any questions that may come up during the discussion period.

---

### Official Review · Reviewer_GGcV · 2023-07-06

**Soundness:** 3 good
**Presentation:** 2 fair
**Contribution:** 3 good
**Rating:** 6
**Confidence:** 2

**Summary:**

In this work, the authors investigate the learning of single-index models with a gradient-based algorithm. They analyze structured inputs considering a spiked matrix model for the data. They show that an appropriate normalization of the weights in the training algorithm can help recover the target direction. They study the behavior of the sample complexity to learn the target direction as a function of the magnitude of the spike in the data, and the alignment of the spike with the target direction. They identify three learning phases, characterizing the phase transitions between them.

**Strengths:**

The precise characterization of feature learning for single-index models is an important topic in machine learning theory. The analysis of structured input is a relevant step toward a better understanding of these models. The characterization of the sample complexity under the spiked model given in terms of a phase diagram is nice.

**Weaknesses:**

The main weakness of the manuscript is the writing, sometimes not entirely clear.

**Questions:**

1) The overview of the relevant literature on single-neuron training is exhaustive and nice. However, when introducing the single-neuron modification after eq. (2.3), I think it is confusing for the reader to introduce a general link $\phi$, stating assumptions on its form (e.g. Assumption 2), and then write "there is little to no hope for learning the entire single-index model unless $\phi = g$".  I would rephrase this sentence in the introduction of the single-neuron simplification, given that the following results analyze non-necessarily strictly realizable settings.
2) The authors do a good job in general in trying to explain their results. I think it would be nice to expand on Proposition 4, e.g. by explaining what is the role of Assumption 2 in a sketch of proof. On a similar note, it is not clear how far from ReLU $\phi$ can be. If an intuitive understanding can be given on the class of $\phi$ to be considered, is more than welcome.
3) In Section 4.2 extends the findings to two-layer neural networks,  in Thm 9 is written"suppose $\phi$ is ReLU and Assumption 2 holds", why there is a need to restrict strictly to a ReLU network in this case and not just assuming that Assumption 2 holds?
4) I like the illustrative power of Figure 1. The hard and easy phases are explained in simple terms on page 2, however, the intermediate phase is not introduced.

**Limitations:**

I do not see any potential negative social impact of this paper.

---

> ### Author Rebuttal · Authors · 2023-08-10
>
> We appreciate the reviewer’s thoughtful evaluation and detailed comments. Following the reviewer’s suggestions, we will improve the general readability in the final version. We address the reviewer’s questions below.
> ***
> **Q1**. We appreciate the reviewer's feedback and we will rephrase the sentence in the final version as follows.
>
> *“We emphasize that unless $\phi = g$ (i.e. the link function is known), the first stage of training only recovers the relevant direction $\boldsymbol{u}$ and is not able to approximate the non-linear link function $g$; thus, a second stage of training with $m > 1$ is required to learn the unknown $g$.”*
> ***
>
> **Q2**. We provide a technical overview in Appendix A and we will move this section to the main text in the final version.
> To provide a quick intuition between the connection of Assumption 2 and Proposition 4, consider the following ODE (see (A.2) for derivation), which follows from Lemma 3. The below inequality can be used to describe the evolution of the alignment between $\overline{\boldsymbol{w}}^t$ and $\overline{\boldsymbol{u}}$
>
> $$\frac{\mathrm{d}\langle \overline{\boldsymbol{w}}^t, \overline{\boldsymbol{u}}\rangle}{\mathrm{d}t} = A_t\mathbb{E}\left[\phi'(\langle \overline{\boldsymbol{w}}^t, \boldsymbol{z}\rangle)g'(\langle\overline{\boldsymbol{u}}, \boldsymbol{z}\rangle )\right] = -A_t\zeta_{\phi,g}(\langle \overline{\boldsymbol{w}}^t, \overline{\boldsymbol{u}}\rangle) \geq cA_t\langle \overline{\boldsymbol{w}}^t, \overline{\boldsymbol{u}}\rangle^{s-1},$$
> where $A_t \geq \lambda_{\text{min}}(\boldsymbol{\Sigma})(1 - \langle \overline{\boldsymbol{w}}^t, \overline{\boldsymbol{u}}\rangle^2)$ and we used Assumption 2 in the last step.
> Integrating this equation will yield the statement of Proposition 4. See Appendix D.1 for a precise derivation.
>
> Regarding the class of admissible activations, the population limit (Proposition 4) can cover any weakly differentiable activation $\phi$ with $\phi,\phi' \in L^2(\gamma)$ (pointed out in line 140). Our finite-sample analysis (Theorems 5 and 7) however is more specific and can accommodate the following activations: 1-ReLU (non-smooth) and 2- Lipschitz activations with a Lipschitz derivative (smooth). Examples of the second class include popular activations like the sigmoid and tanh. We will further clarify this in the final version.
> ***
>
> **Q3**. For Theorem 9, in addition to recovering the relevant direction, we need to approximate the function $g$ using a linear combination of the activation $\phi$. The piecewise-linear structure of ReLU makes the approximation analysis easier, which is why we (along with several recent works in the literature [DLS22,BBSS22,MHPG+23]) focus on function approximation via ReLU units. However, in principle, Theorem 9 can be restated for any activation which is capable of universal function approximation.
> ***
>
> **Q4**. The intermediate regime interpolates between the hard and easy regimes, in which larger $r_2$ and smaller $r_1$ continuously improve the sample complexity. This is in contrast to the hard regime where the complexity is not affected by either $r_1$ or $r_2$, and the easy regime where only $r_2$ determines the sample complexity. We will add a discussion of this regime to the final version of the manuscript.
> ***
>
> We would be happy to clarify any concerns or answer any questions that may come up during the discussion period.
>
> [BBSS22] Alberto Bietti, Joan Bruna, Clayton Sanford, and Min Jae Song, “Learning single-index models with shallow neural networks”, NeurIPS 2022.
>
> [DLS22] Alexandru Damian, Jason Lee, and Mahdi Soltanolkotabi, “Neural Networks can Learn Representations with Gradient Descent”, COLT 2022.
>
> [MHPG+23] Alireza Mousavi-Hosseini, Sejun Park, Manuela Girotti, Ioannis Mitliagkas, and Murat A Erdogdu, “Neural networks efficiently learn low-dimensional representations with SGD”, ICLR 2023.

---

> > ### Comment · Reviewer_GGcV · 2023-08-20
> > **Thank you**
> >
> > I sincerely thank the authors for the rebuttal.
> >
> >
> > After reading the authors’ response and the other reviewers’ comments, I would like to keep my score to a weak accept one.

---

### Official Review · Reviewer_XbBV · 2023-07-11

**Soundness:** 3 good
**Presentation:** 3 good
**Contribution:** 2 fair
**Rating:** 5
**Confidence:** 3

**Summary:**

This paper investigates the learning dynamics of single index models when the input data has a structured covariance. It demonstrates that the commonly used spherical gradient flow fails to learn the target direction even when the spike and target directions are identical. However, the paper introduces an appropriate weight normalization technique that overcomes this limitation and successfully recovers the target direction. It further reveals that leveraging the alignment between the covariance structure and the target direction improves the sample complexity compared to isotropic cases and outperforms lower bounds for rotationally-invariant kernel methods. The paper also suggests future directions, such as studying multi-index models, understanding the limitations from a Correlational Statistical Query (CSQ) perspective, and exploring the effects of different initializations in training networks with multiple neurons.

**Strengths:**

${\bf Originality}$: This paper introduces a new investigation into learning single-index models with a spiked covariance structure, expanding upon prior research that primarily focused on isotropic data. This unique focus on the impact of additional structure in the covariance matrix sets it apart from existing works. Furthermore, the introduction of weight normalization techniques and the exploration of their effects in anisotropic scenarios demonstrate originality in addressing the limitations of conventional spherical gradient dynamics.

$\bf Quality$: This paper demonstrates a high level of quality in terms of its theoretical analysis, rigorous mathematical proofs, and well-supported findings. The authors delve into the dynamics of gradient flow, leveraging mathematical techniques to derive insights into the behavior of the learning algorithm. The use of appropriate assumptions and the incorporation of empirical covariance preconditioning showcase the paper's thoughtful methodology and attention to detail.

$\bf Clarity$: The paper is written in a clear and accessible manner, making complex concepts and mathematical formulations understandable to the reader. The abstract, introduction, and conclusion concisely summarize the objectives, methods, and key findings of the study. The organization of the paper and the logical flow of ideas contribute to its overall clarity, enhancing the reader's comprehension of the research.

$\bf Significance$: The paper holds good implications for the understanding of learning dynamics in the presence of structured covariance matrices. By uncovering phenomena where conventional approaches fail, such as the inability of spherical gradient dynamics to recover the target direction, the paper sheds light on the limitations and challenges associated with anisotropic data. Moreover, the proposed weight normalization techniques, insights into improved sample complexity, and the comparison to rotationally invariant kernel methods highlight the practical relevance and potential impact of the findings.


**Weaknesses:**

${\bf (1)}\ \textbf{The training procedure considered in the paper is not practical:}$ The authors focus on a two-step training procedure that deviates from the standard gradient descent (GD) commonly used in practice. It is challenging to assess the extent to which the results obtained from this training procedure translate to impacts on standard training algorithms.

${\bf (2)} \ \textbf{Differences from the existing works:}$ Some existing works, such as Refs [BBSS22] [BAGJ21], have established some results on learning the single-index model using neural networks. It seems that the authors don't clarify the difference of this work from the existing ones.

${\bf (3)}\ \textbf{Practical implications and applications:}$ Although the authors have successfully established strong theoretical results for learning single-index models using two-layer neural networks, the practical implications of this work on real-world applications of deep learning remain somewhat unclear. Further clarification regarding the practical relevance and potential impact of the findings would greatly benefit readers seeking to understand the practical implications of the research in the field of deep learning.

**Questions:**

$\bf Q1.$ What happens if assuming $g$ and $\phi$ to be differentiable? Can you get stronger results?

$\bf Q2.$ Is it possible to extend the current analysis to the multiple-index models? What are the main technical difficulties?

---

> ### Author Rebuttal · Authors · 2023-08-10
>
> We thank the reviewer for their valuable feedback and positive evaluation. We address the reviewer's concerns below.
> ***
> **Impractical training procedure**:
> While we agree with the reviewer's statement, we emphasize that two-step procedures have become mainstream in the theory of neural networks, mainly because they allow for relatively more in-depth analysis (e.g. [DLS22,BES+22,BBSS22,MHPG+23,ABAM23]). Developing theory when both layers are trained simultaneously is certainly interesting and more practical; yet, existing works in this setting (e.g. large-width mean-field regime) are not able to characterize the generalization behavior to the extent we did in our paper. However, extending our results to such regimes is an interesting direction for future research.
> ***
>
> **Difference from prior works**:
> Our paper distinguishes itself from prior work by focusing on structured input covariance rather than isotropic, which allows us to obtain a sample complexity that, unlike prior works [BAGJ21,BES+22,BBSS22], can be *independent of information exponent* and almost linear in the input dimension $d$ under a spiked model.
>
> More generally, by focusing on the spiked model, we reveal the existence of a *three-stage phase transition* based on the spike-target alignment and the spike magnitude, with each stage representing a separate regime of statistical complexity of learning the target. Furthermore, we establish how preconditioning can help generalization, an observation absent in the non-linear feature learning literature due to concentrating on isotropic data.
>
> Note that we also establish the *necessity of weight normalization* when learning under such covariance structures, which demonstrates that the covariance-agnostic training procedures used in prior work will provably fail in this setting. We discuss these differences in the paragraph starting at line 170 and throughout the introduction (lines 55 to 78), and we will further emphasize the distinctions with the existing works in the revision.
> ***
>
> **Practical implications**: While our primary objective in this paper was entirely theoretical, there are still immediate practical implications. First, note that our results can be interpreted as “*neural networks benefit from additional structure in the input data*”. This is particularly interesting given that, in practice, principal directions of the input covariance contain non-trivial information about the target [HTFF09]; thus, *data whitening* as pre-processing may hurt generalization. Second, our work proves the necessity of proper weight normalization when the data is anisotropic, which provides theoretical justification for weight normalization practices used in deep learning.
> ***
>
> **Differentiable $\phi$ and $g$**: For recovering the target direction, smoothness does not affect *supervised sample complexity*, which is the main quantity of interest. However, smooth activations allow for a better *unsupervised sample complexity* (number of samples used to estimate the covariance), as discussed in line 252.
>
> In section 4.2, we simply use ReLU since its piecewise linearity simplifies the function approximation needed to learn the link function $g$. In principle, Theorem 9 can be restated for any activation that is a universal function approximator, without any change in the sample complexity.
>
> As for the link function $g$, further differentiability will not provide any benefits in theory since the main property of $g$ that determines the sample complexity is its information exponent.
> ***
>
> **Extending the analysis to multiple-index models**: This is a very interesting direction for future work. We believe that in principle, our results should generalize to $k$-index models with $k = \mathcal{O}(1)$ with roughly the same sample complexity. To learn such a model, one needs to show that training the first layer recovers the entire span of $k$ target directions (prior work has shown convergence to this subspace, but recovering the entire subspace is more challenging [MHPG+23]). Even in the simpler isotropic case, [ABAM23] only provides general conjectures in this setting, and their theorems are valid only for specific examples of multiple-index targets. Therefore, while conceptually simple, such generalizations can be highly non-trivial from a technical perspective, mainly due to the complex interactions that arise between the neurons.
> ***
>
> We would be happy to clarify any concerns or answer any questions that may come up during the discussion period.
>
> [ABAM23] Emmanuel Abbe, Enric Boix-Adsera, and Theodor Misiakiewicz, “Sgd learning on neural networks: leap complexity and saddle-to-saddle dynamics”, COLT 2023.
>
> [BAGJ21] Gerard Ben Arous, Reza Gheissari, and Aukosh Jagannath, “Online stochastic gradient descent on non-convex losses from high-dimensional inference”, JMLR 2021.
>
> [BBSS22] Alberto Bietti, Joan Bruna, Clayton Sanford, and Min Jae Song, “Learning single-index models with shallow neural networks”, NeurIPS 2022.
>
> [BES+22] Jimmy Ba, Murat A Erdogdu, Taiji Suzuki, Zhichao Wang, Denny Wu, and Greg Yang, “High-dimensional Asymptotics of Feature Learning: How One Gradient Step Improves the Representation”, NeurIPS 2022.
>
> [DLS22] Alexandru Damian, Jason Lee, and Mahdi Soltanolkotabi, “Neural Networks can Learn Representations with Gradient Descent”, COLT 2022.
>
> [HTFF09] Trevor Hastie, Robert Tibshirani, Jerome H Friedman, and Jerome H Friedman, “The elements of statistical learning: data mining, inference, and prediction”, vol. 2, Springer 2009.
>
> [MHPG+23] Alireza Mousavi-Hosseini, Sejun Park, Manuela Girotti, Ioannis Mitliagkas, and Murat A Erdogdu, “Neural networks efficiently learn low-dimensional representations with SGD”, ICLR 2023.

---

> > ### Comment · Reviewer_XbBV · 2023-08-17
> >
> > Thank you for your detailed rebuttal. I have carefully read through your responses to the comments and truly appreciate the time and effort you have put into addressing each concern. I keep my score unchanged and leave it for the AC for decision.

---

### Decision · Program_Chairs · 2023-09-21

**Decision:**

Accept (poster)

**Comment:**

The paper analyzes neural network learning in the single index model, with structured data. In this setup, the observed label y is a function of a one dimensional projection <u,x> of the features x. Previous works assumed an isotropic Gaussian distribution for x. This submission considers the situation in which x follows a Gaussian distribution with spiked covariance. It performs a comprehensive analysis of this setup, identifying phase transitions in terms of (i) the alignment between u and the covariance spike and (ii) the information exponent of the link function. An interesting byproduct of this analysis is the following observation: in the spiked model, SGD does not accurately learn u, but a normalized variant (somewhat reminiscent of batch normalization) does.

Reviewers positively evaluated the manuscript’s technical quality, noting that extending results on the single index models to more realistic (and structured) distributions is an important direction for theory. The paper’s phase transition results give a comprehensive picture of this model. Concerns included the realism of the model, as well as certain artificial aspects of the algorithm (this is a two-stage theoretical algorithm, not SGD or batch normalized SGD). The review consensus is that the paper’s merits (high technical quality, important direction for theory) outweigh these concerns; the AC concurs, and recommends acceptance.